# MMMG: A Comprehensive and Reliable Benchmark for Multitask Multimodal Generation

## Abstract

Automatically evaluating multimodal generation presents a significant challenge, as automated metrics often struggle to align reliably with human evaluation, especially for complex tasks that involve multiple modalities. To address this, we present MMMG, a comprehensive and human-aligned benchmark for multimodal generation across 4 modality combinations (image, audio, interleaved text and image, interleaved text and audio), with a focus on tasks that present significant challenges for generation models, while still enabling reliable automatic evaluation through a combination of models and programs. MMMG encompasses 55 tasks (including 31 newly developed ones), each with a carefully designed evaluation pipeline, and 1248 instructions to systematically assess reasoning, controllability, and other key capabilities of multimodal generation models. Extensive validation demonstrates that MMMG is highly aligned with human evaluation, achieving an average agreement of 94.4%. Benchmarking results on 29 multimodal generation models reveal that even though the state-of-the-art model, GPT IMAGE, achieves 70.7% accuracy for image generation, it falls short on multimodal reasoning and interleaved generation. Furthermore, results suggest considerable improvement space in audio generation, highlighting an important direction for future research.

## 1 Introduction

As investments in multimodal generative AI grow, current models are rapidly advancing their capabilities in generating text (Achiam et al., 2023), images (Podell et al., 2024), audio (Evans et al., 2025), and their interleaved combinations (Chen et al., 2025d; Wang et al., 2024). However, rigorous and reproducible evaluation of multimodal generation lags behind, raising a critical question: how can we accurately and effectively assess the capabilities of these models?

Human evaluations (Chiang et al., 2024; Saharia et al., 2022; Liu et al., 2025), while considered the gold standard, are prohibitively expensive for comprehensive assessment at scale. Moreover, inherent subjectivity makes it difficult to systematically identify specific model weaknesses, limiting targeted improvements. As an alternative, existing automated evaluation approaches face two main limitations. First, it is hard to align automatic evaluation metrics well with human judges. Most multimodal generation benchmarks (Xia et al., 2025; Chen et al., 2024b; 2025a) rely on multimodal language models as judges (MLM-as-a-judge) (Hu et al., 2023; Chen et al., 2024a) without carefully validating their reliability, potentially causing misalignment with human judgment (Chen et al., 2024a; Pu et al., 2025). Second, most benchmarks focus solely on single modalities (Ji et al., 2024; Ghosh et al., 2023; Xie et al., 2025b), failing to capture the rich interleaved multimodal content (vision, language, speech/audio) that characterizes real-world tasks such as cross-modal reasoning (Hu et al., 2024).

To address these gaps, we introduce MMMG, a new benchmark containing tasks that meet two criteria: (1) tasks that are *verifiable* as defined in IF-Eval (Zhou et al., 2023), where outputs can be objectively verified by programs through straightforward checks (e.g., checking if a speech transcript begins with a keyword by comparing the first word with the keyword), and (2) tasks with significant *generation-evaluation gaps*, where the generation step is challenging due to complex constraints, yet the evaluation step remains simple (e.g., generating an image of a snowman without a carrot nose can

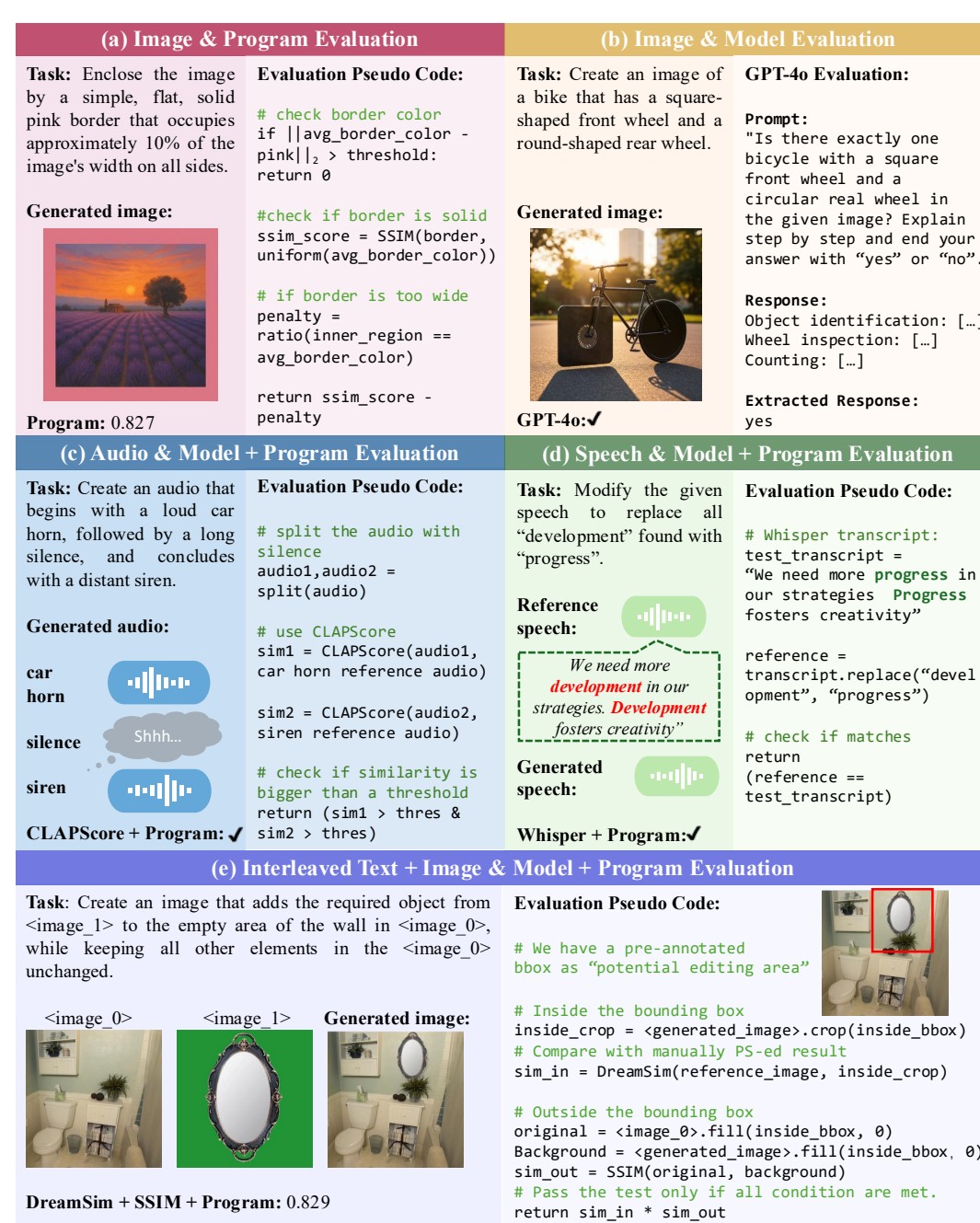

Figure 1: Examples of tasks and their evaluation in MMMG. For each task, we develop an evaluation metric using programs, models or their combinations. The tasks are either verifiable by programs or with big generation-evaluation gaps: generation is challenging while automatic evaluations can easily align with human judgment. We show pseudo-code to demonstrate the evaluation process.

be challenging due to spurious correlation (Ye et al., 2024), but verifying the absence of the carrot nose can be achieved accurately by prompting a VLM). Example tasks can be found in Figure 1.

MMMG includes 55 tasks (31 are newly developed) and 1248 instructions across 4 modality combinations—text, image, audio, and interleaved modalities—as depicted in Table 1. By categorizing tasks based on assessed capabilities, MMMG enables fine-grained analysis of model performance and targeted identification of weaknesses.

To validate the human alignment of MMMG, we conduct human evaluation across 41 tasks—938 instructions and 2556 evaluation questions—with each question assessed by three independent

| Task | Subtask | Description | Input | Output | # Inst. | Evaluation |
|------|---------|-------------|-------|--------|---------|------------|
| Object Generation | Inclusion | Include one or two unrelated objects in the scene. | T | img | 40 | VLM |
| | Exclusion | Exclude one related object from the scene. | T | img | 40 | VLM |
| | Count | Generate exactly N objects. | T | img | 40 | VLM |
| | Attribution | Generate an object with uncommon attributes. | T | img | 40 | VLM |
| | Knowledge | Reason the answer object to a multi-hop question. | T | img | 40 | VLM |
| | Commonsense | Reason the answer object/scene by commonsense. | T | img | 40 | VLM |
| Relation Control | Comparison | Generate two objects with uncommon relations. | T | img | 20 | VLM |
| | Universal | Generate objects with all identical/different attributes. | T | img | 20 | VLM |
| | Relative Spatial | Generate two objects with given relative spacial relation. | T | img | 20 | VLM |
| | Absolute Spatial | Generate one/two objects in asked absolute image quarter. | T | img | 20 | VLM |
| Image Format | Border Fill | Surround the image with pure, solid and colored border. | T | img | 15 | Program + SSIM |
| | Region Fill | Fill the given region with pure and solid color. | T | img | 15 | Program + SSIM |
| Text Rendering | Single | Render English text on one object. | T | img | 20 | VLM |
| | Double | Render two English texts on two objects. | T | img | 20 | VLM |
| | Multi-Lingual | Render one Chinese text on one object. | T | img | 20 | VLM |
| Image Editing | Object Adding | Add a new object by textual/visual prompting. | T, img | img | 20 | VLM + SSIM |
| | Object Removing | Remove an existing object by textual/visual prompting. | T, img | img | 20 | VLM + SSIM |
| | Object Replacing | Replace an existing object by textual/visual prompting. | T, img | img | 20 | VLM + SSIM |
| | Object Altering | Change the attributes of objects by textual/visual prompting. | T, img | img | 20 | VLM + SSIM |
| | Text Adding | Add text/long sentence by textual/visual prompting. | T, img | img | 20 | VLM + SSIM |
| | Text Altering | Remove/Modify/Translate text by textual/visual prompting. | T, img | img | 20 | VLM + SSIM |
| | Interleaved Adding | Add an external image object to the original image. | T, imgs | img | 20 | DreamSim + SSIM |
| | Interleaved Altering | Change the color of an object with external color reference. | T, imgs | img | 20 | DreamSim + SSIM |
| Image Consistency | Semantic | Generate multiple images in semantic order. | T | imgs | 20 | VLM |
| | Composition | Gradually add individual objects in the given order. | T | imgs | 20 | VLM |
| | Decomposition | Gradually remove object combination in the given order. | T, img | imgs | 20 | VLM |
| | Multi-View | Generate multiple views of the reference scene. | T, img | imgs | 20 | SSIM |
| | Multi-Angle | Generate multiple views of the reference object. | T, img | imgs | 20 | SSIM |
| Image-Text Coherence | Self Count | Count objects in the self-generated image. | T | T, img | 20 | VLM |
| | Self Color | Name object colors in the self-generated image. | T | T, img | 20 | VLM |
| | Self Size | Compare object sizes in the self-generated image. | T | T, img | 20 | VLM |
| | Self Relative Spatial | Decide relative spacial relation in the generated image. | T | T, img | 20 | VLM |
| | Self Absolute Spatial | Decide absolute spacial relation in the generated image. | T | T, img | 20 | VLM |
| | Self OCR | Recognize the text in the generated image. | T | T, img | 20 | VLM |
| Interleaved Reasoning | Math | Solve the IQ-test puzzles. | T, img | T, img | 20 | VLM |
| | Code | Read SVG codes and render the SVG image. | T | T, img | 20 | VLM + DreamSim |
| Sound Generation | Begin-End | Begin/End the audio with the given sound effect. | T | audio | 20 | CLAPScore |
| | Positional Inclusion | Include one sound effect at a relative audio position. | T | audio | 20 | CLAPScore |
| | Silence | Generate two ordered sound effects separated by silence. | T | audio | 20 | CLAPScore |
| | Knowledge | Reason the answer sound to a multi-hop question. | T | audio | 18 | CLAPScore |
| Music Generation | Instrument Inclusion | Generate music with the given instrument. | T | audio | 20 | CLAPScore |
| | Instrument Exclusion | Generate music without the given instrument. | T | audio | 20 | CLAPScore |
| | Genre | Generate music with the given genre. | T | audio | 20 | CLAPScore |
| | Tempo | Generate music with the given tempo. | T | audio | 20 | Program |
| | Intensity | Generate music with fade in/out at the beginning/end. | T | audio | 20 | Program |
| Interleaved Speech Generation | Voice Attribution | Generate an en. speech with required voice attributes. | T | audio | 20 | Whisper+W2V+Program |
| | Voice Replication | Generate an en. speech replicating the reference voice. | T, audio | audio | 20 | Whisper + WavLM |
| | Multi-Lingual | Generate a zh. speech with required voice attributes. | T | audio | 20 | Whisper+W2V+Program |
| | Transcript Generation | Generating an speech with textual constraints for transcripts. | T | audio | 20 | Whisper + Program |
| | Transcript Editing | Editing an speech with textual constraints for transcripts. | T, audio | audio | 20 | Whisper + Program |
| | Conversation | Generate a conversation with given speaker order. | T | audios | 20 | Whisper + WavLM |
| | Speech Translate | Directly translate multi-lingual speech into English speech. | T, audio | audio | 40 | Whisper + Program |
| | Speech Retrival | Retrieve the key speech segment in extremely long speech. | T, audio | audio | 40 | Whisper + Program |
| Modality Order Control | Image-Text | Generate interleaved image-text content in given order. | T, img | T, imgs | 20 | Program |
| | Audio-Text | Generate interleaved audio-text content in given order. | T, audio | T, audios | 20 | Program |

Table 1: Detailed task definition and metadata for MMMG. T denotes text modality, img for image modality, imgs for multiple images, audio for audio and audios for multiple audios. We evaluate each task with the method that yields the highest human agreement. green background indicates new tasks.

annotators and aggregated by majority vote. MMMG achieves an average human agreement of 94.4% with average inter-annotator agreement being 97.1%. Modality-specific agreements achieve 93.5% for image, 92.3% for audio, 96.6% for interleaved image-text, and 91.0% for interleaved audio-text,

| Dataset | # Samples | # Tasks | Generation Modality | | | | Evaluation | | | | Tested Capability | | |
|---|---|---|---|---|---|---|---|---|---|---|---|---|---|
| | | | 🖼 | 🔊 | $\mathbb{T}$ + 🖼 | $\mathbb{T}$ + 🔊 | human | mllm | score | code | gen | edit | reason |
| GenEval (Ghosh et al., 2023) | 553 | 6 | ✔ | ✗ | ✗ | ✗ | ✗ | ✗ | ✔ | ✔ | ✔ | ✗ | ✗ |
| DrawBench (Saharia et al., 2022) | 200 | 11 | ✔ | ✗ | ✗ | ✗ | ✔ | ✗ | ✗ | ✗ | ✔ | ✗ | ✗ |
| GenAI-Bench (Li et al., 2024) | 1,600 | 8 | ✔ | ✗ | ✗ | ✗ | ✔ | ✗ | ✗ | ✗ | ✔ | ✗ | ✗ |
| AudioTime (Xie et al., 2024) | 500 | 4 | ✗ | ✔ | ✗ | ✗ | ✗ | ✗ | ? | ✔ | ✔ | ✗ | ✗ |
| MusicEval (Liu et al., 2025) | 384 | 1 | ✗ | ✔ | ✗ | ✗ | ✔ | ✗ | ✔ | ✗ | ✔ | ✗ | ✗ |
| CommonVoice (Ardila et al., 2020) | 58,250 | 1 | ✗ | ✔ | ✗ | ✗ | ✗ | ✗ | ✔ | ✗ | ✔ | ✗ | ✗ |
| MMIE$_{\text{MMG}}$ (Xia et al., 2025) | 16,487 | 7 | ✗ | ✗ | ✔ | ✗ | ✗ | ? | ✗ | ✗ | ✔ | ✗ | ✗ |
| CoMM (Chen et al., 2024b) | 227,000 | 4 | ✗ | ✗ | ✔ | ✗ | ✗ | ? | ✔ | ✗ | ✔ | ✗ | ✗ |
| ISG-Bench (Chen et al., 2025a) | 1,150 | 21 | ✗ | ✗ | ✔ | ✗ | ✗ | ? | ✔ | ✗ | ✔ | ✔ | ✗ |
| MixEval-X$_{\text{MMG}}$ (Ni et al., 2025) | 600 | 3 | ✔ | ✔ | ✗ | ✗ | ✔ | ? | ✗ | ✗ | ✔ | ✔ | ✗ |
| Eval-Anything (Ji et al., 2024) | 500 | 6 | ✔ | ✔ | ✔ | ✗ | ✔ | ? | ✗ | ✗ | ✔ | ✗ | ✗ |
| MMMG (Ours) | 1248 | 55 | ✔ | ✔ | ✔ | ✔ | ✗ | ✔ | ✔ | ✔ | ✔ | ✔ | ✔ |

Table 2: Comprehensiveness of MMMG, compared with other multimodal generation benchmarks. 🖼, 🔊, $\mathbb{T}$ + 🖼, $\mathbb{T}$ + 🔊 represent image, audio, interleaved image-text, and interleaved audio-text generation, respectively. "score" stands for embedding-based / rule-based similarity score, "code" for programmatically verification, and "reason" for multi-step reasoning. ? represents low human alignment or no human experiments. MMMG exceeds other benchmarks in the number of covered tasks and modalities while providing more reliable evaluation.

with relative improvements over prior best results by 12.7% for image, and 34.5% for interleaved image-text evaluation (Ghosh et al., 2023; Chen et al., 2025a).

We benchmark 29 open and proprietary multimodal generation models using the optimal evaluation methods identified in human studies. Partial results are shown in Figure 2; the rest are in Appendix D.2. We find that modality-unified autoregressive models (ARMs) surpass diffusion models in image generation, with GPT IMAGE (OpenAI, 2025) achieving the best accuracy of 70.7%. This indicates ARMs trained on extensive language-image datasets have stronger linguistic capabilities, enabling better instruction following and improved alignment with user intent. However, GPT IMAGE still falls short in interleaved text-image reasoning tasks for math and code, achieving only 10.1% accuracy and 3D scene transformation at 31.8%. Our qualitative error analysis reveals that another ARM, GEMINI 2 IMAGE, tends to tangle multiple images in generation, hindering accurate image-sequence and image-text pair generation. Additionally, MMMG reveals greater improvement potentials for audio generation compared to image, with top-performing models achieving accuracies of 48.7% for sound and 46.3% for music generation. Overall, MMMG is the *first* benchmark while ensuring reliability, provides the most comprehensive multimodal model ranking and fine-grained capability analysis.

## 2 RELATED WORK

**Interleaved Multimodal Generation.** Interleaved multimodal generation involves generating coherent content across multiple modalities simultaneously, such as visual storytelling (Huang et al., 2016; Wen et al., 2023), reference-based image editing (Chen et al., 2025c), and voice chatbots (Chu et al., 2024). Effective models must understand multimodal inputs and produce aligned outputs across modalities. Current approaches include (1) LLM backbones with specialized decoders (Chen et al., 2025d; Xie et al., 2025a), which leverage dedicated components to render visual or audio outputs; (2) modality-unified autoregressive models (Chern et al., 2024; Hurst et al., 2024; Wang et al., 2024), processing text, visual, and acoustic tokens within a single sequence model, enabling native generation of interleaved content; and (3) agent-based methods (Chen et al., 2025a), using a "Plan-Execute-Refine" pipeline with modality-specific tools. Despite significant advances, evaluation frameworks for interleaved multimodal generation remain underdeveloped, particularly in accurately and automatically assessing cross-modal consistency, and instruction-following capabilities.

**Multimodal Generation Evaluation.** Evaluating image, audio and their interleaved generation presents unique challenges that have been addressed through several approaches, each with notable limitations, including (1) using specialized visual or audio models (Ghosh et al., 2023; Xie et al., 2025b), which struggle to generalize beyond their training data (Ming et al., 2022); (2) directly employing MLMs as evaluators (Xia et al., 2025; Chen et al., 2024b; 2025a), which often misalign with human judgments (Chen et al., 2024a); and (3) for image evaluation particularly, leveraging visual question answering (VQA) to assess specific aspects of generated content (Hu et al., 2023;

Lin et al., 2024), which declines significantly in accuracy when facing complex evaluation scenarios that require nuanced reasoning (Chen et al., 2025a). To address these limitations, previous research incorporates extensive human preference data to enhance MLM accuracy (Xiong et al., 2024; Yao et al., 2025). Our work is an orthogonal approach that carefully designs evaluation instructions to leverage current MLM strengths while mitigating their limitations, enabling reliable multimodal evaluation without extra training or finetuning. Table 2 compares MMMG with existing benchmarks.

## 3 MMMG BENCHMARK CONSTRUCTION

Our goal is to build a multimodal generation benchmark that (1) covers a wide range of modalities and their combinations (image, audio, interleaved text and image, interleaved text and audio) with diverse tasks spanning different model capabilities. For each task, (2) we also ensure reliable automated evaluation that aligns well with humans. In this section, we first discuss our data and instruction construction in detail (§3.1), and then introduce the evaluation methods we built for each task (§3.2).

### 3.1 DATA CURATION

To guarantee high-quality instructions and reliable evaluation, we design a systematic data curation pipeline consisting of three key stages.

**Task Creation.** We begin by creating an initial pool of 78 candidate task templates. These tasks span various modality combinations and each task aims to evaluate a single multimodal generation capability. The complete list of 76 tasks can be found in Appendix B.2. For each task, we conduct a rigorous feasibility assessment to ensure there is at least one reliable evaluation method available— either programmatic verification or a literature-supported, highly human-aligned evaluation method. Based on this process, we narrow our task pool down to 60 tasks.

**Instruction Synthesis and Validation.** We employ a human-in-the-loop approach to synthesize high-quality instructions for each task. Inspired by Self-Instruct (Wang et al., 2023), we prompt GPT-4O (Hurst et al., 2024) with the task template and quality-controlled criteria to generate 10 candidate instructions per task. We then go through a two-stage selection process:

- **Quality Filtering.** Initially, we remove instructions that are ambiguous (instructions with unclear or multiple interpretations), unrealistic (instructions that describe improbable or nonsensical scenarios), or redundant (instructions that closely resemble previously accepted examples). For instance, unrealistic instruction "*Generate an image of a forest without any trees*" is discarded because it is semantically contradictory and unlikely to occur in actual user queries.

- **Verifiability Assessment.** After quality filtering, we sample generated outputs and verify if at least one evaluation methods yield high alignment with human judgments, which avoids cases where models fail to accurately evaluate out-of-distribution samples. For example, GPT-4O can accurately count fewer than 10 objects but is prone to errors counting more than 10.

We then generate another 10 candidate instructions and repeat the generation and validation process continues until we gather approximately 40 high-quality instructions per task. Statistically, only 10% of generated instructions pass examination, highlighting the high standard of data selection.

**Postprocessing.** For final quality control, we perform a task-level postprocessing step to further refine our benchmark. This involves two procedures: (1) Task filtering: we recruit two independent annotators to judge if each task is realistic. We eliminate six tasks that at least one annotator judges to be unrealistic. (2) Instruction paraphrasing: To ensure linguistic diversity and prevent models from memorizing specific instruction patterns, we paraphrase all remaining instructions. Each paraphrased instruction is examined manually to verify that it is equivalent to the original instruction semantically.

To this end, we ultimately collect a total of 1248 instructions across 55 tasks spanning 4 modality combinations. This systematic approach ensures that MMMG provides a comprehensive, fine-grained, and reliable evaluation framework for assessing multimodal generation capabilities. The detailed definitions and metadata of each task in MMMG can be found in Table 1.

## 3.2 EVALUATION METHOD

We report the evaluation method used for each task in Table 1. For more details about implementation, please refer to Appendix C.3.

**VLM.** We employ vision language models (VLMs) for most reference-free image evaluation tasks. We do not use object detection or OCR models because VLMs demonstrate superior performance in out-of-domain scenarios. A common practice to boost VLM-as-a-judge is visual question answering (VQA), where models generate verification questions and answer the questions based on images to determine if images follow given instructions. However, we find that automatically generated question-answer pairs like those in TIFA (Hu et al., 2023) often misalign with human judgment on challenging tasks. Therefore, we manually design visual questions for each instruction based on these important principles as shown in Figure 1(b):

- Chain-of-thought prompting significantly improves VLM performance on boolean questions. Specifically, instructing models not to output yes/no at the beginning of their responses substantially reduces hallucination which echoes findings in Zhang et al. (2024).

- Multiple-choice format can boost VLM's performance on object counting and spatial relationship reasoning. We hypothesize that multiple-choice questions effectively reduce the output space, thereby simplifying these tasks. For example, including an option like "*E. More than 6*" in object counting questions can prevent miscounting errors in scenarios with numerous objects.

- Adding negative prompts helps alleviate visual hallucination. For instance, VLMs can easily overlook a constraint such as "*one basketball with a cube shape*," whereas "*one basketball with a cube shape instead of a sphere*" forces the VLM to reject a spherical basketball.

**Image Similarity.** For reference-based image evaluation tasks requiring perceptual similarity, we employ DreamSim (Fu et al., 2023). When exact matching is necessary, we use SSIM (Wang et al., 2004). For image editing tasks, we implement a dual approach: DreamSim/VQA evaluates the edited region, while SSIM assesses the unmodified areas outside it, ensuring that local editing instructions are precisely followed as shown in Figure 1(e).

**Audio Similarity.** Research indicates that current audio language models (ALMs) cannot reliably analyze sound or music clips (Sakshi et al., 2025). Therefore, we select ESC-50 (Piczak, 2015), OpenMIC-2018 (Humphrey et al., 2018) and GTZAN (Sturm, 2012) as reference datasets for sound and music evaluation, and compute the average top-10 CLAP cosine similarity (Wu et al., 2023) with reference audio as shown in Figure 1(c).

**Audio Model.** For specialized audio analysis, we employ several targeted models. WAVLM (Chen et al., 2022) is employed for speaker similarity verification. For speech transcription, we use Whisper (Radford et al., 2023) as shown in Figure 1(d). Gender classification in speech leverages a finetuned WAV2VEC checkpoint (Fiury, 2023). For music tempo computation, we employ BEATTHIS (Foscarin et al., 2024) for beat tracking and the beats statistics are used for music tempo computation.

**Program.** For programmatic verification, we utilize PIL for image analysis as shown in Figure 1(a), Librosa (McFee et al., 2015) and Praat (Boersma & Van Heuven, 2001) for audio pitch, intensity, and speed analysis. For textual constraint verification, we follow the implementation of IF-Eval (Zhou et al., 2023). We use word accuracy (WAcc) to evaluate textual similarity for visual text rendering and text-to-speech tasks which requires exact matching.

**Scoring.** Each generation receives either a binary classification or an accuracy score representing the exact level of how each instruction being followed. We convert binary classification to numerical scores (0.0 for incorrect, 1.0 for correct) and then macro-average all task scores following previous work setup (Ghosh et al., 2023). Despite employing different evaluation protocols, MMMG guarantees implicit unification across protocols by high human alignment. For example, even though CLAPScore gives scores between [0, 1], task-specific thresholds are selected to best align with human judgment. This makes CLAPScore's sensitivity equivalent to VLM binary judgments because both are calibrated against the same unified human judgment standard.

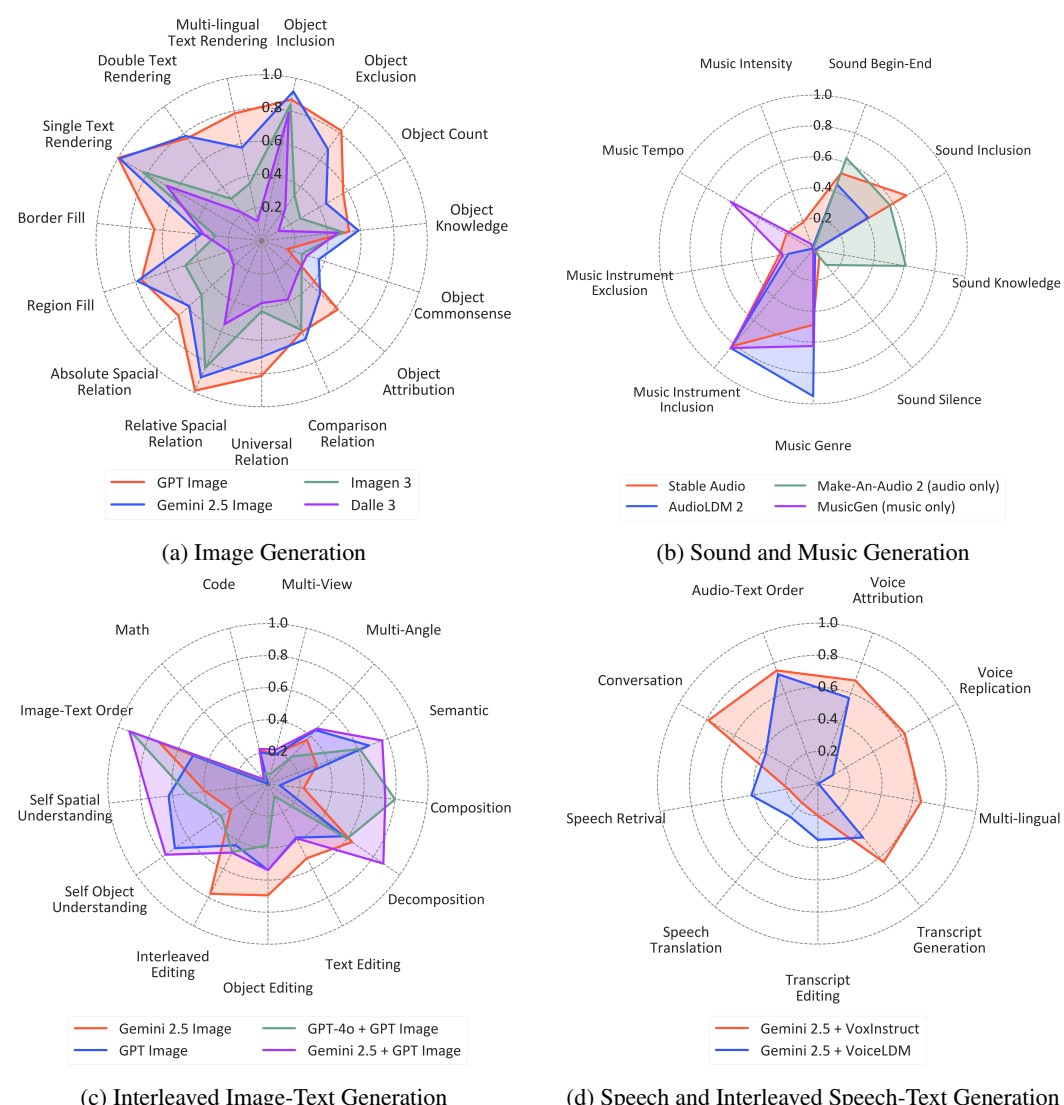

(a) Image Generation

(b) Sound and Music Generation

(c) Interleaved Image-Text Generation

(d) Speech and Interleaved Speech-Text Generation

Figure 2: Benchmark results of multimodal generation models on MMMG covering four modality combinations. Please refer to Table 1 for more detailed category information. We aggregate some sub-tasks for interleaved image-text generation. GPT IMAGE beats all other models on most image generation tasks, and strongly competes other baselines in generating consistent image sequences and coherent interleaved image-text contents.

## 4 RESULTS AND ANALYSIS

In this section, we first report our human alignment experiment results in §4.1, and then the benchmarking results evaluated by the most human-aligned metrics in §4.2. We also report the correlation between MMMG with real-world human preference leaderboard in §4.3.

### 4.1 ALIGNMENT WITH HUMAN JUDGES

We conduct human evaluations on 938 instructions evaluated by models. For each instruction, we randomly select two models from all models evaluated on this instruction and obtain one generation per model. Each generation is evaluated by two independent annotators, randomly selected from our pool of 20 graduate student annotators. To standardize the evaluation process and reduce subjective bias, we design specific multiple-choice questions for each instruction exemplified in Appendix C.5, thereby constraining annotators' responses to a fixed set of choices and ensuring high inter-annotator agreement. In cases of disagreement, a third annotator determines the final annotation. In total, human

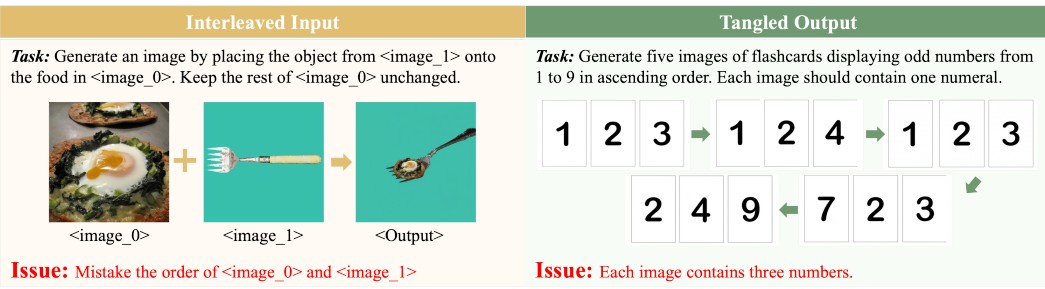

Figure 3: Two prevalent failure cases observed in interleaved image-text generation tasks for GEMINI 2 IMAGE: (1) models fail to accurately interpret the order of images in interleaved inputs; and (2) models frequently blend multiple images together, possibly due to limitations in encoding multiple images with continuous latent image representations.

studies involve 2556 evaluation questions and collect 5112 annotations. For verifiable instructions, human alignment validation is unnecessary as these tasks are designed for objective programmatic verification. Human-model and inter-human agreement measures can be found in Table 8.

MMMG demonstrates high human alignment, with average best human-model agreement for image, audio, interleaved image-text, and interleaved audio-text being 0.935, 0.923, 0.966 and 0.910 respectively, calculated by selecting the method achieving the highest agreement per task and averaging across tasks. The average inter-annotator agreement remains as high as 0.971 with the worst case being 0.917. MMMG also outperforms previous best benchmark alignment significantly: agreement on image generation surpasses GenEval (0.830) by 12.7%, and Pearson correlation on interleaved image-text generation surpasses ISG-Bench (0.718) by 34.5%. Experiments show that while GPT-4O remains the most human-aligned image evaluation model with an average agreement of 0.935, GEMINI 2.5 shows superior performance on spatial relationships and editing evaluation. Though open-source models like QWEN2.5-VL still have a gap with proprietary models, its judging accuracy already surpassing previous SOTA benchmarks, suggesting the high reliability of MMMG. For audio evaluation, even though $CLAPScore_{text}$ yields a satisfactory agreement of 0.923, it relies highly on the quality of reference audio, thus making it challenging for out-of-domain audio evaluation.

## 4.2 BENCHMARKING RESULTS

We benchmark models with the most aligned evaluation methods for each task. Selected model performances are illustrated in Figure 2, with complete evaluation results provided in Appendix D.2.

**Image Generation.** ARMs outperform diffusion models significantly, with GPT IMAGE and GEMINI 2.5 IMAGE achieving accuracies of 0.707 and 0.654 respectively, ranking 1st and 2nd. This indicates that ARMs with stronger linguistic capabilities can better follow instructions. However, models struggle notably when generating objects with uncommon attributes or unusual relationships, showing average accuracies of only 0.340 and 0.416 respectively. This underscores the vulnerability of image generation models to out-of-domain instructions. Despite top models (GPT IMAGE) achieving 0.531 accuracy on knowledge reasoning tasks, they fail drastically on commonsense reasoning with an accuracy of only 0.163, suggesting their performance may rely more on memorization than genuine reasoning ability.

**Interleaved Image-Text Generation.** Ensembling GEMINI 2.5 improves GPT IMAGE's accuracy by 21.1%, indicating that agent-based models outperform unified ARMs via stronger planning capabilities and clear modality separation, which leads to better image consistency and coherence. Still, some tasks pose considerable challenges, with the best-performing combination (GEMINI 2.5 + GPT IMAGE) achieving limited accuracies of 0.131 on math and coding reasoning, 0.341 on 3D scene transformations, and 0.383 on text editing. Error analysis on one of the ARM, GEMINI 2 IMAGE, reveals that it suffers from two primary failure modes: (1) misinterpreting image order in interleaved inputs, and (2) tangle multiple images in output due to continuous latent representations, as shown in Figure 3. However, GEMINI 2.5 IMAGE as an ARM perform best at image editing tasks with an accuracy of 0.672, which may indicate that ARMs preserve input image information better due to reduced information loss from image-to-text transitions.

**Sound and Music Generation.** Only MAKE-AN-AUDIO 2, which leverages LLMs for instruction parsing, shows competence in sound reasoning. Other models exhibit significant reasoning limitations, achieving low accuracies of 0.233 for instrument exclusion and 0.175 for sound knowledge reasoning. Volume control is also poor, with silence generation and intensity control reaching just 0.048 and 0.063 accuracy, respectively. Only MUSICGEN effectively handles tempo control, while only STABLE AUDIO and AUDIOLDM 2 support both sound and music generation, demonstrating that audio generation models generally remain domain-constrained.

**Speech and Interleaved Speech-Text Generation.** SPIRIT LM, the only inherently interleaved speech-text model fails entirely on most speech generation tasks. Agent-based models struggle with tasks that require simultaneous speech understanding and generation, reaching average accuracies of 0.275 for speech editing, 0.212 for speech translation, and 0.304 for speech retrieval. Ablation studies reveal that LLM backbones perform perfectly on speech transcripts, indicating failures stem from error accumulation in speech processing rather than reasoning deficits.

### 4.3 CORRELATION WITH REAL-WORLD LEADERBOARD

We compare the correlation of the MMMG score with the Chatbot Arena (Chiang et al., 2024) score on the text-to-image task. We take the Arena Score for 9 image generation models under the "User Prompts Only" category as a gold reference. We report the Pearson correlation and Spearman's rank correlation coefficient between gold arena scores and scores produced by evaluating on different benchmarks in Table 3. We compare with GenEval, Draw-Bench, and GenAI-Bench. We employ VQAScore (Lin et al., 2024) to replace human evaluation on DrawBench and GenAI-Bench; due to budgetary limitations, we randomly sample 400 out of 1600 instructions for GenAI-Bench.

MMMG provides reliable model rankings with a Spearman correlation coefficient of 0.561, sig-

| Model | Arena | GenEval | Draw | GenAI | MMMG |
|---|---|---|---|---|---|
| IMAGEN 3 | 1064 | 0.707 | 0.861 | 0.793 | 0.474 |
| RECRAFT V3 | 1018 | 0.732 | 0.826 | 0.817 | 0.441 |
| LUMA PHOTON | 997 | 0.738 | 0.766 | 0.804 | 0.587 |
| FLUX 1.1 PRO | 992 | 0.588 | 0.725 | 0.736 | 0.431 |
| IDEOGRAM 2 | 1011 | 0.615 | 0.757 | 0.782 | 0.508 |
| DALLE 3 | 978 | 0.627 | 0.809 | 0.811 | 0.352 |
| SD 3.5 | 911 | 0.591 | 0.711 | 0.715 | 0.332 |
| GEMINI 2 IMAGE | 996 | 0.669 | 0.765 | 0.783 | 0.592 |
| GPT IMAGE | 1126 | 0.808 | 0.793 | 0.824 | 0.707 |
| Spearman | | 0.460 | 0.444 | 0.418 | **0.561** |

Table 3: Correlation of automated image generation benchmarks with Chatbot Arena. Arena, Draw, GenAI represent Chatbot Arena, DrawBench, and GenAI-Bench. MMMG achieves the highest correlation with Chatbot Arena. This indicates even though our instructions are synthetic, the evaluation results are still highly human-aligned.

nificantly outperforming baseline benchmarks. This indicates that despite that synthetic instructions may not fully align with real-world queries, MMMG achieves higher alignment with human preferences. Such results suggest that evaluator alignment (i.e., the reliability of the evaluation method) may outweigh instruction distribution alignment (i.e., the extent to which benchmark tasks reflect real-world task distributions) for accurate model assessment. Moreover, MMMG demonstrates superior differentiation capabilities among evaluated models. The performance gap of 0.375 between the highest- and lowest-ranked models is much larger than the next-best baseline (GenEval), which has only a gap of 0.217. This larger range underscores MMMG's enhanced ability to distinguish among models, particularly for differentiating performance among top-tier models.

Due to the lack of real-world human preference leaderboards like Chatbot Arena for other modalities, we leave human preference correlation studies for other modalities as future work.

## 5 CONCLUSION

In this work, we introduce MMMG, a comprehensive automated evaluation suite for multitask multimodal generation, addressing critical limitations of existing benchmarks. We collect 1248 high-quality instructions spanning 55 diverse tasks involving text, image, audio, and interleaved content. Extensive human validation demonstrates that MMMG correlates better with human judgments compared to previous benchmarks. Benchmarking results highlight ongoing challenges in multimodal reasoning, interleaved generation, and audio generation. The fine-grained nature of MMMG enables detailed capability analysis, providing valuable insights for targeted multimodal improvements. Beyond serving as a leaderboard, we hope MMMG inspires scalable collection of verifiable validation signals for future multimodal generation training.

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

## A  LIMITATIONS AND ETHICS STATEMENT

### A.1  LIMITATIONS

While MMMG constitutes a significant advancement in automated multimodal generation evaluation, we acknowledge several limitations inherent to our methodology and scope.

**Limited Task Coverage.**  MMMG does not exhaustively cover all potential tasks within multi-modal generation, particularly in the domains of interleaved image-text generation and sound/music generation. This limitation primarily arises from current inadequacies in available evaluation methods or models, which fail to yield sufficiently human-aligned results on numerous widely-used tasks. Such gaps in coverage may introduce biases into our model rankings, potentially misaligning evaluation results with actual user experiences. To mitigate this, we intend to dynamically expand and update our benchmark tasks in real-time as more powerful and reliable evaluation models become available. We also include tasks that we considered commonly used but abandoned due to infeasible evaluation in Appendix B.2.

**Dependence on Proprietary Models.**  Our evaluation relies on proprietary models (e.g., GPT-4O, GEMINI 2.5). The substantial performance gap between proprietary and open-source models makes reliance on proprietary models necessary for achieving highly accurate and human-aligned evaluations across diverse tasks. Unfortunately, current open-source alternatives often lack sufficient accuracy on certain complex tasks, rendering them unsuitable as reliable evaluators. Consequently, this dependence limits broad reproducibility and access within the academic community, highlighting the urgent need for improved and accessible open-source evaluation models.

### A.2  ETHICS STATEMENT

MMMG is designed as a general-purpose benchmark for evaluating multimodal generation capa-bilities and deliberately excludes high-stakes applications, e.g. medical image generation or other safety-critical domains that would require specialized evaluation protocols, domain expertise, and more rigorous validation procedures. For researchers and practitioners deploying multimodal genera-tion models in high-stakes scenarios, we emphasize that MMMG's evaluation framework should not be considered sufficient (even with reported SOTA human agreement on general domains) without additional domain-specific validation, expert review, and safety protocols appropriate to the specific application context.

### A.3  REPRODUCIBILITY STATEMENT

We provide comprehensive details to ensure full reproducibility of MMMG's benchmark construction and evaluation. Appendix B.1 documents all data sources with specific dataset versions and access links. Appendix C.1 specifies all 29 evaluated generation models with exact checkpoint identifiers, along with 3 VLMs and 4 specialized audio models used for evaluation. Detailed evaluation protocols are provided in Appendix C.3, including task-specific VLM prompts, programmatic verification procedures using PIL, Librosa, and Praat, etc., and agent-based generation system prompts (Tables 6 and 7). Human annotation procedures are documented in Appendix C.5 with example annotation interfaces (Figure 4) and task-specific annotation questions. Computational requirements and costs are detailed in Appendix B.4, including GPU specifications, runtime estimates and API costs. All benchmark data, evaluation code, and model outputs are publicly available at this anonymous link.

### A.4  USE OF LARGE LANGUAGE MODELS (LLMS) STATEMENT

For paper writing, we used LLMs (specifically CLAUDE 4) solely for language polishing and improving clarity after drafting the complete manuscript ourselves. All factual claims, experimental results, analyses, and conclusions were written by the authors and carefully verified for accuracy before and after any LLM-assisted editing. For data collection, we employed GPT-4O to synthesize part of the candidate instructions as described in Section B.1. These LLM-generated instructions underwent rigorous quality filtering and verifiability assessment by human annotators as stated in Section 3.1. Additionally, we occasionally used LLMs to assist with debugging experimental code.

# B    DETAILED DATASET INFORMATION

## B.1    DATA SOURCE

- **Object Reasoning.** We sample from HotpotQA (Yang et al., 2018) through the official website and WISE Niu et al. (2025) through the official website. We take the QA pairs where the answers are individual objects and can be directly transformed into image generation instructions or the answers are nations and can be transformed into the national flags or animals generation instructions.

- **Image Editing.** We sample images from EmuEdit (Sheynin et al., 2024) through the "facebook/emu_edit_test_set" checkpoint on Huggingface (Wolf et al., 2019) for object adding, removing, modifying, local, color and text editing tasks. We modify the instructions to make sure they are clear, unambiguous and more challenging. We also sample object images from COCO (Lin et al., 2014) through the official website and use PhotoShop to combine with the scene images in EmuEdit to form golden reference images. We sample scene images from CLEVR (Johnson et al., 2017) through the official website for the interleaved color modifying task, since modifying color for pure-colored geometries is much more unambiguous than regular objects. We also use PhotoShop to generate the golden reference images.

- **3D transformation.** We sample instructions and golden reference images from ISG-Bench (Chen et al., 2025a) through the official website. We polish the instructions to make sure they are clear and unambiguous.

- **Math.** We sample images from MM-IQ (Cai et al., 2025) through the "huanqia/MM-IQ" checkpoint on Huggingface. We manually edit the images to transform the multiple-choice questions into free-form generation questions. We have 2 annotators to check if the free-form questions can only have one possible answer without alternatives.

- **Code.** We sample SVG codes from StarVector (Rodriguez et al., 2023) through the "starvector/text2svg-stack" checkpoint on Huggingface. and transform the original image-to-text instructions into interleaved reasoning instructions. We samples SVG codes with a length between 1000-1500 characters to control difficulty.

- **Sound Generation.** We make sure all the target sounds fall in the 50 categories in ESC-50 (Piczak, 2015) through the "ashraq/esc50" checkpoint on Huggingface so that $CLAPScore_{audio}$ can have reference audios to compare with.

- **Instrument Generation.** We make sure all the target instruments fall in the 20 categories in OpenMIC-2018 (Humphrey et al., 2018) through the official website so that $CLAPScore_{audio}$ can have reference audios to compare with.

- **Speech.** For speech replication task, we samples speaker voices from LibriSpeech (Panayotov et al., 2015) ASR corpus through the official website and use them as reference speeches for voice replication tasks. For speech translation task, we sample original speeches and human-annotated translation from CVSS dataset (Jia et al., 2022). For speech retrieval tasks, we sample from TED-LIUM 2 (Rousseau et al., 2014) for the original extremely long speech contexts.

The remaining tasks are generated from GPT-4O with manually designed templates.

## B.2    EXCLUDED TASKS

We present the remaining 23 tasks we considered from our initial task set in Table 4. We exclude "Format Color", "Format Symmetric", "Speech Encoding" tasks since they are not commonly seen in real user queries and "Image-to-Sound" and "Sound-to-Image" tasks are excluded because no models today can support these modalities. Other tasks are excluded because we could not find any reliable evaluation methods for those tasks.

## B.3    DATASET STATISTICS

We present some important statistics of MMMG in Table 5.

## B.4    COMPUTATION STATISTICS

| Task | Example | Input | Output |
|------|---------|-------|--------|
| Table Generation | Create a 2x2 table image. In the first column, place the text 'apple' in the top cell and 'pear' in the bottom cell. In the second column, place an image of an apple in the top cell and an image of a pear in the bottom cell. | 𝕋 | 🖼 |
| Figure Generation | Create a histogram to visualize the given data. <data> | 𝕋 | 🖼 |
| Format Color | Create a watermelon farm using only varying shades of red. | 𝕋 | 🖼 |
| Format Symmetric | Generate an image of a futuristic cityscape. The image must be axisymmetric along the vertical center line. | 𝕋 | 🖼 |
| Art Style | Create a painting of a dandelion sea in Impressionist style. | 𝕋 | 🖼 |
| Photography | Create q zoomed out photo of a small bag of coffee beans from below. | 𝕋 | 🖼 |
| Scene Editing | Make the weather in <image_0> sunny. | 𝕋, 🖼 | 🖼 |
| Sound Count | Generate an audio of exactly three door knocks. | 𝕋 | 🔊 |
| Sound Order | Generate an audio of a can being opened followed by a sipping sound. | 𝕋 | 🔊 |
| Sound Duration | Generate audio of a car horn lasting for 3 seconds. | 𝕋 | 🔊 |
| Speech Emotion | Generate an audio of a woman sorrowfully saying, "What a life." | 𝕋 | 🔊 |
| Speech Accent | Generate an audio of a man speaking in Indian accent, "What a beautiful day!" | 𝕋 | 🔊 |
| Speech Background | Generate an audio of a man speaking in noisy train station distantly, "I am really busy." | 𝕋 | 🔊 |
| Speech Stress | Generate an audio of a man saying, "Give me money now!" with stress on word "now". | 𝕋 | 🔊 |
| Music Emotion | Generate a vibrant, pulsating disco drum track. | 𝕋 | 🔊 |
| Music Lyrics | Create a flute melody with the lyrics, <lyrics>. | 𝕋 | 🔊 |
| Singer Attribution | Generate a jazz piece accompanied by lyrics "<lyrics>", featuring a tenor singer performing in Bel Canto style. | 𝕋 | 🔊 |
| Lyrics Editing | Replace the lyrics in <audio_0> with <lyrics>, keeping the original melody unchanged. | 𝕋, 🔊 | 🔊 |
| Transition Visualization | Generate three images showing the transition process from <image_0> to <image_1>. | 𝕋, 🖼 | 🖼 |
| Future Prediction | Generate three images showing the future events after <image_0>. | 𝕋, 🖼 | 🖼 |
| Speech Encoding | Generate a speech about sustainable development, and provide the speech transcript encoded in Base64. | 𝕋 | 𝕋, 🔊 |
| Image-to-Sound | Create a music predominately featuring the instrument shown in <image_0>. | 𝕋, 🖼 | 🔊 |
| Sound-to-Image | Draw an image showing the animal that is mostly likely to make the sound in <audio_0>. | 𝕋, 🔊 | 🖼 |

Table 4: Tasks that are not included in MMMG. 𝕋 denotes text modality, 🖼 for image modality, 🖼 for multiple images, 🔊 for audio and 🔊 for multiple audios. We hope to incorporate these tasks when reliable evaluation methods are available.

The evaluation pipeline for MMMG requires at least a single NVIDIA A10 GPU for open-source models, and APIs from OpenAI and Gemini for proprietary models. In our experiments, we used a single NVIDIA A40 GPU. On average, the evaluation runtime for each task is approximately 4 minutes, incurring a API cost of about $1.1 for a sample size of 4. For the generation phase, runtime significantly varies depending on the model itself. The most time-consuming model tested is YUE, which runs on a single NVIDIA H100 GPU. On average, YUE takes around 3 hours to complete generation per task.

| Statistics | Number |
|------------|--------|
| Total number of modality combinations | 4 |
| Total number of tasks | 55 |
| - I : A : I-T : A-T | 15 : 12 : 22 : 6 |
| Total number of questions | 1248 |
| - I : A : I-T : A-T | 410 : 238 : 440 : 160 |
| Total number of images | 522 |
| Total number of audios | 90 |
| Average length of instructions | 242.5 |

Table 5: Statistics of MMMG. I, A, I-T, A-T stands for image, audio, interleaved image-text and interleaved audio-text generations respectively.

## B.5 DESIGN PRINCIPLE

For single-modality generation tasks (image, audio, speech), we follow established task collections from prior work that have identified critical capabilities including object composition, spatial reasoning, attribute control, text rendering and instruction following, etc. (Ghosh et al., 2023; Li et al., 2024). However, for interleaved multimodal generation, no systematic guidelines exist for determining which tasks are most important for comprehensive evaluation. To address this gap, we systematically identify three fundamental capabilities for interleaved multimodal generation based on analysis of prior work and real-world applications (Chen et al., 2025a; Xia et al., 2025; Deng et al., 2025): (1) **Input preservation and understanding**: The ability to accurately process and retain information from multimodal inputs, including understanding relationships between provided images, text, and their combinations. (2) **Modality sequence consistency**: The ability to generate coherent

sequences of outputs (e.g., multiple images or speech segments) that maintain temporal, spatial or semantic consistency. (3) **Cross-modal coherence**: The ability to generate outputs where different modalities (text and images, text and audio) are semantically aligned and mutually supportive and also keep modality number and order correct.

These capabilities are assessed through carefully selected proxy tasks. For instance, image editing tasks serve as diagnostic measures for input preservation and understanding. A model that cannot accurately modify specific objects while preserving unchanged regions likely lacks the fine-grained multimodal comprehension necessary for more complex interleaved generation tasks. This design is supported by recent findings that during interleaved multimodal pre-training, models develop basic multimodal understanding and generation abilities before complex editing and reasoning capabilities emerge (Deng et al., 2025), establishing a developmental hierarchy where foundational skills are prerequisites for higher-order reasoning.

Our empirical results validate this hierarchical assumption: current models already struggle significantly with basic object-focused editing tasks (Section 4.2), while complex reasoning tasks in math and code show near-zero performance across most models. This pattern suggests that current limitations in complex compositional reasoning may stem from deficiencies in foundational capabilities. As more advanced models emerge with improved capabilities, we plan to expand MMMG's task scope to include increasingly complex compositional reasoning tasks, enabling deeper investigation of the relationships between foundational and advanced capabilities.

## C  DETAILED EXPERIMENT SETUP

### C.1  MODEL DETAILS

**Generation.**  We evaluate 29 multimodal generation models specified in Appendix C.1. To encourage diversity, we only incorporate the latest model of a series. Even though our benchmark supports comprehensive and cross-modality evaluation, current multimodal generation models have very restricted output modalities. Thus, we categorize these models by their supported output modalities into image, interleaved image-text, sound-music, and interleaved speech-text generation.

- **Image Generation.** We include GPT IMAGE (OpenAI, 2025), through the "gpt-image-1" checkpoint on OpenAI API; IMAGEN 3 (Baldridge et al., 2024), through the "imagen-3.0-generate-002" checkpoint on Gemini API; RECRAFT V3 (AI, 2024c), through the "recraftv3" checkpoint on Recraft API; LUMA PHOTON (AI, 2024b), through the "luma/photon" checkpoint on Replicate API; FLUX 1.1 PRO (Labs, 2024), through the "black-forest-labs/flux-1.1-pro" checkpoint on Replicate API; IDEOGRAM 2 (AI, 2024a), through the "ideogram-ai/ideogram-v2" checkpoint on Replicate API; DALLE 3 (Betker et al., 2023), through the "dall-e-3" checkpoint on OpenAI API; STABLE DIFFUSION 3.5 Rombach et al. (2022), through the "stabilityai/stable-diffusion-3.5-large" checkpoint on Huggingface; JANUS PRO (Chen et al., 2025d), through the official implementation; BLIP-3O (Chen et al., 2025b), through the official implementation.

- **Interleaved Image-Text Generation.** We include SEED-LLAMA (Ge et al., 2024), through the official implementation; ANOLE (Chern et al., 2024), through the official implementation on Github; GEMINI 2 IMAGE (Team et al., 2023), through the "gemini-2.0-flash-preview-image-generation" checkpoint on Gemini API; GEMINI 2.5 IMAGE (Team et al., 2023), through the "gemini-2.5-flash-image-preview" checkpoint on Gemini API; and GPT IMAGE (Team et al., 2023), through the "gpt-image-1" checkpoint on OpenAI API for image generation and the "gpt-4.1" with "image_generation" tool on OpenAI API for interleaved image-text generation. We also implement two agents models composing of a MLM and an image generation model: GPT-4O + GPT IMAGE and GEMINI 2.5 + GPT IMAGE. GEMINI 2.5 is through the "gemini-2.5-pro-preview-03-25" checkpoint on Gemini API and GPT-4O is through the "gpt-4o-2024-08-06" checkpoint on Openai API.

- **Sound and Music Generation.** We include STABLE AUDIO (Evans et al., 2025), through the "stabilityai/stable-audio-open-1.0" checkpoint on Huggingface, and AUDIOLDM 2 (Liu et al., 2024), through the "cvssp/audioldm2-large" checkpoint on Huggingface, capable of generating both sound and music. We also include sound generation models: AUDIOGEN (Kreuk et al., 2022), through the official implementation; MAKE-AN-AUDIO 2 (Huang et al., 2023), through the official implementation; and TANGO 2 (Majumder et al., 2024), through the "declare-lab/tango2-full"

checkpoint on Huggingface. We also include music generation models: MUSICGEN (Copet et al., 2023), through the "facebook/musicgen-large" checkpoint on Huggingface; TANGO MUSIC (Kong et al., 2024), through the "declare-lab/tango-music-af-ft-mc" checkpoint on Huggingface; and YUE (Yuan et al., 2025), through the official implementation.

- **Interleaved Speech-Text Generation.** We include SPIRIT LM (Nguyen et al., 2025), through the official implementation. We also implement two agents models composing of a MLM and a voice synthesizing model: GEMINI 2.5 + VOXINSTRUCT (Zhou et al., 2024) and GEMINI 2.5 + VOICELDM (Lee et al., 2024). VOXINSTRUCTION is through the official implementation and VOICELDM is through the official implementation.

**Evaluation.** We compare several evaluation methods. For image generation, we include GPT-4O, GEMINI 2.5, and QWEN2.5-VL-72B-INSTRUCT (Bai et al., 2025) to perform VQA for evaluation. CLIPScore (Hessel et al., 2021) is found as less aligned with human judgment in previous studies (Hu et al., 2023), thus not included. For sound and music evaluation, we include CLAPScore$_{audio}$, CLAPScore$_{text}$, and employing GEMINI 2.5 for acoustic question answering (AQA). CLAPScore$_{audio}$ computes the CLAP cosine similarity with reference audio, while CLAPScore$_{text}$ computes the similarity with reference audio captions. We pick the the optimal thresholds separately per dataset.

The checkpoints of the evaluation models we use are: GPT-4O, through the "chatgpt-4o-latest" checkpoint on OpenAI API; GEMINI 2.5, through the "gemini-2.5-pro" checkpoint on Gemini API; and QWEN2.5-VL, through the "Qwen/Qwen2.5-VL-72B-Instruct" checkpoint on Huggingface. For audio models, we employ CLAP, through the "laion/clap-htsat-unfused" checkpoint on Huggingface; WHISPER, through the "openai/whisper-large-v3" checkpoint on Huggingface and a finetuned Chinese speech-to-text checkpoint "BELLE-2/Belle-whisper-large-v3-zh" on Huggingface; WAVLM, through the "microsoft/wavlm-base-sv" checkpoint on Huggingface; and "Wav2Vec", through the "alefiury/wav2vec2-large-xlsr-53-gender-recognition-librispeech" checkpoint on Huggingface.

## C.2 GENERATION DETAILS

Following the experimental setup in Ghosh et al. (2023), we sample 4 generations for every instruction in our benchmark. We employ a temperature of 0 and a retry count of 4 for MLMs and sampling steps of 200 for diffusion models. We keep other parameters, such as guidance scale, as default values. For non-agent models, we directly provide instructions to the model. For agent-based models, we prepend a system prompt to the instructions. This system prompt explicitly instructs the model to generate outputs following a structured, function-call-based approach. When the model needs visual or auditory outputs, it generates placeholders formatted as function calls within the text. Each placeholder clearly specifies the generation instructions and any necessary references to prior outputs or provided multimedia in user's instructions. For each placeholder, we extract the function call, which are then fed into specialized image or audio generation models. To correctly handle references to previously generated media, we employ topological sorting. This ensures media outputs are generated in a sequence by dependencies, and circular dependencies are identified and reported as errors. Detailed system prompt for interleaved image-text agent is in Table 6 and interleaved audio-text agent is in Table 7.

## C.3 EVALUATION DETAILS

**Prompts for VLMs**

- **Object Count.** "*How many [object] are there in the given image? Choose from the options: A. Less than 3 or the image is blank B. 3 C. 4 D. 5 E. 6 F. More than 6. Respond only with the option letter (A, B, C, D, E or F). Do not provide any explanation, reasoning, or additional information.*" Multiple choice questions can boost VLM's performance on object count tasks. We employ this prompt for object count and self count tasks.

- **Absolute Spacial Relationship.** "*The [object] is located in which section of the image? Choose from the options: A. bottom left B. bottom right C. up left D. up right E. none of the above (positioned in a more central way) Explain step by step and end your answer with Answer: [only an optional letter].*" Multiple choice questions can boost VLM's performance on spacial reasoning

You are a multimodal assistant capable of generating both text and images. When visual content would enhance your response or is specifically requested, you can generate or edit images through advanced diffusion models.

To generate or edit an image:

1. Identify when visual content would be beneficial or requested.

2. Insert an image generation/editing placeholder using the following format:

   ```
   <image_start><image_prompt="Detailed
   image generation or editing prompt
   here."><image_end>
   ```

3. The post-processing system replaces this placeholder with an image created or edited based on your instructions.

4. Naturally incorporate references to the generated or edited image in your ongoing conversation.

When crafting image prompts, follow these guidelines:

**For image prompts:**

- Provide detailed, specific descriptions (15-30 words) for optimal results.

- Include artistic styles (photorealistic, cartoon, watercolor, etc.) or style transfers.

- Specify key objects and their attributes (colors, textures, etc.), or modifications.

- Detail composition elements (spatial relationships, perspective, lighting, etc.), or compositional changes.

- Ensure instructions are clear and concise.

**For image references:**

Three reference types are available:

1. Image generation (no reference):

   ```
   
   ```

2. Editing user-provided images:

   Format: `` where `i` is the index of the provided image (indices starting at 0).
   Example: `` references the first provided image.
   Multiple images example: `` references the first and third provided images.

3. Editing previously generated images:

   Format: ``, where `N` is the sequential number of previously generated images (starting from 0).
   Example: `` references the fourth generated image.
   Multiple images example: `` references the first and third generated images.

**Important:** Use only one reference type within each placeholder. Different reference types may be used across multiple placeholders.

Provide concise and direct responses following user instructions precisely. Always maintain the exact placeholder format for proper parsing, ensuring that both images and text appear in the required order. Do not omit any necessary text following image placeholders.

Table 6: System prompt for interleaved image-text agent.

tasks. We employ this prompt for absolute spatial relationship and self absolute spatial relationship recognizing tasks.

- **Left-Right Spacial Relationship.** "*Looking at the 2D composition of the image, what is the horizontal alignment relationship between the [object1] and the [object2]? Choose from the*

You are a multimodal assistant capable of generating both text and audio. When audio content would enhance your response or is specifically requested, you can generate audio through text-to-audio models.
To generate audio:

1. Identify when audio content would be beneficial or requested.

2. Insert an audio generation placeholder using the format:

    ```
    <audio_start><audio_type="sound" OR "speech"
    OR "music"><audio_text="Text to be spoken
    here."><audio_style="Descriptive text here." OR
    audio reference ID><audio_end>
    ```

3. The post-processing system replaces this placeholder with generated audio based on your specifications.

4. Naturally incorporate references to the generated audio in your ongoing conversation.

When crafting audio prompts, follow these guidelines:
**Audio Type:**

- Must be exactly one of: `"sound"`, `"speech"`, or `"music"`.
- `"speech"`: For human speech.
- `"sound"`: For environmental sounds or effects.
- `"music"`: For musical compositions or instrumental pieces.

**Audio Text:**

- For `"speech"`: Provide the exact transcript.
- For `"sound"` or `"music"`: Leave as empty string (`""`).
- Keep speech concise (typically under 50 words).

**Audio Style:**

1. **Descriptive Text:**
   - For `"speech"`: Specify voice characteristics (gender, emotion, pace, pitch, accent).
   - For `"sound"`: Specify sound source, environment, qualities.
   - For `"music"`: Specify genre, mood, tempo, instruments.

2. **Reference Audio:**
   - For consistency, particularly with speech:
     - Previously generated audio: `<audio_style=#N>` (N is sequential number starting at 0).
     - User-provided audio: `<audio_style=N>` (N is sequential number of provided audio starting at 0).
   - Important: Only reference audio that itself does not reference previous audio to avoid circular references.

Provide concise, direct responses precisely following user instructions. In multi-speaker scenarios, maintain consistent and distinctive voice characteristics for each speaker. Always maintain the exact placeholder format for correct parsing

Table 7: System prompt for interleaved audio-text agent.

*options: A. the [object1] is obviously to the left of the [object2]. B. the [object1] is obviously to the right of the [object2]. C. the [object1] is neither obviously to the right nor left of the [object2]. Explain step by step and end your answer with Answer: [only an optional letter].*" VLMs tend to be confused by perspective relationship, thus we ask VLMs to focus on 2D composition. We employ this prompt for relative spatial relationship and self relative spatial relationship recognizing tasks.

- **Up-Down Spacial Relationship.** "*Looking at the 2D composition of the image, what is the vertical alignment relationship between the [object1] and the [object2]? Choose from the options: A. the [object1] is obviously positioned higher than the [object2]. B. the [object1] is obviously positioned lower than the [object2]. C. the [object1] is neither obviously positioned higher nor lower than the [object2]. Explain step by step and end your answer with Answer: [only an optional letter].*" We employ this prompt for relative spatial relationship and self relative spatial relationship recognizing tasks.

- **OCR English.** "*### Instruction: Recognize all the major texts (ignore small texts on the edge) ONLY on [object]. Only recognize texts in Latin alphabet characters (a-z, A-Z). Do not correct the text if it is misspelled, nonsense or wrong, output the most direct recognition result. Do not call any function. ### Output format: Output an executable Python list of all recognized texts from top to down, from left to right, e.g. ["Hello World", "Good morning"]. Output an empty list if the there is no text on [object] or the image is blank.*" We employ this prompt for single and double text rendering and self OCR tasks.

- **OCR Chinese.** "*### Instruction: You are a conservative text recognition model. Your task is to recognize all the major Chinese characters in the given image. If the Chinese characters in the image are wrongly written or distorted, you should return an empty string. Do not call any function. ### Output format: Only a string of all recognized characters from top to down, from left to right. Do not add quotations.*" We employ this prompt for multi-lingual text rendering task. Since VLMs tend to recognize Chinese characters incorrectly or identify fake characters, we employ two separate VLMs and use the intersection of their recognition results to improve accuracy.

- **Text Pattern Verifying (Math)** "*Below are two descriptions of the same geometric pattern, one is ground-truth and the other is model-generated. Your task is to judge if the generated description is accurate. Analyze step by step and end your answer with "Yes" or "No". Here are some criteria: 1. The model-generated pattern must state the pattern clearly without ambiguity. For example, a 3\*3 grid of circles with some circles filled is ambiguous. 2. Make sure the overall structure, the position and situation of each element are accurate. Specifically, the situation of each element can include: filled (black, grey, filled with black or any equivalent words), unfilled (white, hollow, empty or any equivalent words), missing (the position is empty or missing). If the situation is not specified in the ground-truth, the element can take any situation of the right shape. 3. If the ground-truth describes a coordinate system, the x-axis will increase from left to right while y-axis will increase from top to down. For example, for a 3\*3 grid, the (3,2) coordinate is the middle-right element.*" We employ this prompt for math task.

- **Image Verifying (Math)** "*Your task is to judge if the given image accurately follows the ground-truth pattern. Analyze step by step and end your answer with "Yes" or "No". Here are some criteria: 1. Make sure the overall structure, the position and situation of each element are accurate. Specifically, the situation of each element can include: filled (black, grey, filled with black or any equivalents), unfilled (white, hollow, empty or any equivalents), missing (the position is empty, missing or any equivalents). If the situation is not specified in the ground-truth, the element can take any situation of the right shape. 2. If the ground-truth describes a coordinate system, the x-axis will increase from left to right while y-axis will increase from top to down. For example, for a 3\*3 grid, the (3,2) coordinate is the middle-right element. 3. If the given image contains multiple patterns (e.g. multiple grids) or question mark, the given image doesn't follow the ground-truth pattern.*" We employ this prompt for math task.

- **Object Existing.** "*Is/Are there [detailed object description] in the given image? Explain step by step and end your answer with "Yes" or "No". Answer "No" if the image is blank.*" We design detailed object description for each instruction manually, include object number, object attributes and undesired negative attributes, etc.. We employ this prompts for all image tasks unmentioned above. For spatial relation tasks, we first exam if the object number is accurate by object existing prompt and then check spatial relationship by corresponding prompts.

**Program Verifying**

- **Solid Color Fill.** The evaluation procedure starts by cropping the targeted region from the image and calculating its average RGB value. The average RGB value is compared with a standard reference color; if the relative deviation exceeds 15%, indicating significant color discrepancy, the

evaluation returns zero. Next, structural consistency is assessed by computing the SSIM between the targeted region and an artificially generated solid region filled with the calculated average RGB color, confirming color uniformity. Finally, the procedure examines over-fill by evaluating the margin area surrounding the targeted region and computing the proportion of pixels matching the region's average RGB color. The ratio as penalty is subtracted from the SSIM score.

- **Image Editing.** The evaluation for image editing begins by manually labeling a potential editing area within each image. Then crop the edited area from the generated image and compare against the corresponding area in a reference image or assessed via a VLM. Additionally, regions outside this area are compared with corresponding original outside area using SSIM to detect unintended changes. The final score is the product of these two comparisons, reflecting editing accuracy and preservation of original content.

- **Sound Generation.** For begin-end tasks, clip the first or last 4 seconds of audio directly. For positional inclusion tasks, crop the corresponding fraction of the audio. For silence detection tasks, utilize the `librosa.effects.split` function to segment audio based on silence intervals and then verify if each section contains target sound through CLAPScore$_{\text{udio}}$.

- **Music Generation.** For tempo evaluation, use BEATTHIS to extract beat tracks and calculate Beats Per Minute (BPM). For intensity evaluation, analyze the initial and final 4 seconds of the music, plotting the energy spectrum through `librosa.feature.rms` and computing its slope and goodness of fit. Only audio segments demonstrating clear upward or downward trends in energy pass the intensity evaluation.

- **Speech Generation.** For pitch evaluation, calculate the average energy of each pitch through `parselmouth.Sound.to_pitch` and select the pitch with the highest average energy through `parselmouth.Sound.to_intensity` as the speech pitch. For speed evaluation, transcribe English audio using WHISPER and compute words per minute (WPM); for Chinese audio, compute characters per minute (CPM). For textual constraints, normalize transcripts using WHISPER's tokenizer (removing punctuation, case sensitivity, etc.) and evaluate with the tools of IFEval. For speech translation task, we calculate BLEU (Papineni et al., 2002) and for other speech tasks, we calculate Word Accuracy.

### C.4 ANNOTATION INTERFACE

We design task-specific annotation interfaces by Gradio (Abid et al., 2019), each including reference images or audio, model's generated outputs, judgment instructions, and judgment criteria. We preprocess some generated outputs to assist annotators in their judgments. For example, we provide cropped images within editing area for image editing tasks and clipped audio segments at the beginning or end for audio begin-end tasks. Judgments are typically collected through multiple-choice radio buttons to ensure high inter-annotator agreement. However, for OCR tasks specifically, annotators type the recognized text directly. An example of annotation interface is in Figure 4.

### C.5 ANNOTATION QUESTIONS

For image and interleaved image-text evaluation tasks, we employ the same questions as the prompts used for VLMs. We paraphrase the questions to make them more annotator friendly and add judging criteria to reduce the ambiguity of the questions. For audio and interleaved audio-text evaluation tasks, we design new annotations questions as follow:

**Music Instrument.** "*What is the dominant instrument played the given audio? Reminder: 1. Failed generation should be considered as none of the above. 2. Choose multiple labels only when you are unsure or the given audio clearly have different types of instruments.*" We employ this question for instrument inclusion and exclusion tasks.

**Music Genre.** "*What is the dominant genre played the given audio? Reminder: 1. Failed generation should be considered as none of the above. 2. Choose multiple labels only when you are unsure or the given audio can fall into different types. 3. Only choose a genre when it is very obvious/typical.*" We employ this question for instrument inclusion and exclusion tasks.

flute examples ◀

saxophone examples ◀

guitar examples ◀

drums examples ◀

accordion examples ◀

♫ Response ⬇

0:00                                                                     0:10

◁)) [1x]                    ◀◀ ▶ ▶▶

Evaluation

What is the dominant instrument played the given audio?
Reminder:
1. Failed generation should be considered as none of the above.
2. Choose multiple labels only when you are unsure or the given audio can fall into different types.

Judgement

☐ flute    ☐ saxophone    ☐ guitar    ☐ drums    ☐ accordion    ☐ None of the above

Next

Figure 4: Human annotation interface for instrument inclusion task. Typically, an inference will include reference audios/images, model's generation, evaluation instruction, evaluation criteria and judgment radio boxes and next/previous button.

**Sound Inclusion.** *"Is the given audio about [sound]? Reminder: 1. Chose yes when [sound] is the main sound existing in the audio. 2. [sound] should be common real-world sound without distortion."* We employ this question for all sound generation tasks.

**Speaker Similarity.** *"Are the speeches coming from the same speaker? Reminder: 1. Little speaker voice difference can be tolerated, but overall, there should be no major difference."* We employ this question for voice replication and conversation tasks.

**Speaker Gender** *"What is the gender of the speaker in the given speech? Reminder: 1. Choose none of above when the voice sounds like electronic synthesizer sound or it is hard to categorize into binary genders. 2. Do not consider speech quality (clarity and fluency, etc.) when judging gender."* We employ this question for voice attribution and multi-lingual speech tasks.

# D  EXPERIMENT RESULTS (CONT.)

## D.1  CORRELATION WITH HUMAN ANNOTATION

| Task | GPT-4O | | GEMINI 2.5 | | QWEN2.5-VL | | IAA | |
|------|--------|--------|------------|--------|------------|--------|-------|-------|
| | agree | corr | agree | corr | agree | corr | agree | corr |
| Object Inclusion | **0.925** | **0.776** | 0.900 | 0.715 | 0.888 | 0.657 | 1.000 | 1.000 |
| Object Exclusion | **0.963** | **0.924** | 0.913 | 0.823 | 0.925 | 0.855 | 1.000 | 1.000 |
| Object Count | 0.875 | 0.709 | 0.963 | **0.912** | **0.900** | 0.763 | 0.975 | 0.943 |
| Object Knowledge | **0.963** | **0.925** | **0.963** | **0.925** | 0.938 | 0.875 | 1.000 | 1.000 |
| Object Commonsense | 0.913 | 0.787 | 0.888 | 0.696 | **0.938** | **0.822** | 1.000 | 1.000 |
| Object Attribution | **0.938** | **0.848** | **0.938** | 0.835 | 0.900 | 0.758 | 1.000 | 1.000 |
| Compassion Relation | **0.925** | **0.850** | 0.875 | 0.741 | 0.850 | 0.699 | 0.950 | 0.896 |
| Universal Relation | **0.975** | **0.951** | 0.900 | 0.818 | 0.925 | 0.860 | 0.975 | 0.951 |
| Relative Spatial | **0.925** | **0.819** | 0.825 | 0.640 | 0.800 | 0.572 | 0.950 | 0.875 |
| Absolute Spatial | 0.825 | 0.641 | **0.925** | **0.839** | 0.839 | 0.775 | 0.983 | 0.960 |
| Text Rendering (TR) | 0.991 | 0.994 | **0.992** | **1.000** | 0.967 | 0.942 | 1.000 | 1.000 |
| Double TR | **0.841** | **0.906** | 0.646 | 0.662 | 0.574 | 0.471 | 0.938 | 0.938 |
| Multi-lingual TR | **0.889** | **0.989** | **0.889** | 0.968 | 0.800 | 0.875 | 1.000 | 1.000 |
| Semantic | **0.958** | **0.910** | 0.946 | 0.890 | 0.940 | 0.869 | 0.982 | 0.961 |
| Composition | **0.971** | **0.930** | 0.942 | 0.847 | 0.920 | 0.793 | 0.978 | 0.944 |
| Decomposition | **0.971** | **0.941** | **0.971** | **0.941** | 0.949 | 0.900 | 0.978 | 0.956 |
| Text Adding | **0.969** | **0.996** | 0.750 | 0.710 | 0.927 | 0.980 | 0.950 | 0.950 |
| Text Altering | 0.950 | 1.000 | 0.950 | 0.976 | **0.975** | **1.000** | 0.950 | 0.950 |
| Object Adding | **0.975** | **0.912** | 0.925 | 0.728 | 0.825 | 0.428 | 1.000 | 1.000 |
| Object Removing | 0.975 | 0.933 | 0.975 | 0.933 | **1.000** | **1.000** | 1.000 | 1.000 |
| Object Replacing | 0.925 | 0.819 | **0.975** | **0.941** | 0.900 | 0.762 | 0.925 | 0.819 |
| Object Altering | **0.975** | **0.951** | 0.875 | 0.747 | 0.850 | 0.704 | 0.925 | 0.819 |
| Self Count | **0.975** | **0.950** | 0.950 | 0.899 | 0.525 | 0.006 | 1.000 | 1.000 |
| Self Color | **0.950** | 0.881 | **0.950** | **0.883** | 0.642 | 0.363 | 0.983 | 0.960 |
| Self Size | **0.892** | **0.788** | 0.867 | 0.735 | 0.917 | 0.838 | 0.967 | 0.933 |
| Self OCR | **0.906** | **0.909** | 0.806 | 0.790 | 0.748 | 0.690 | 1.000 | 1.000 |
| Self Relative Spatial | 0.838 | 0.669 | **0.950** | **0.896** | 0.813 | 0.605 | 0.963 | 0.923 |
| Self Absolute Spatial | 0.913 | 0.821 | **0.950** | **0.897** | 0.963 | 0.923 | 0.975 | 0.948 |
| Math | 0.950 | 0.436 | **1.000** | **1.000** | 0.950 | -0.026 | 0.988 | 0.703 |
| Code | **1.000** | **1.000** | **1.000** | **1.000** | **1.000** | **1.000** | 1.000 | 1.000 |
| Average | **0.935** | **0.865** | 0.913 | 0.846 | 0.867 | 0.718 | 0.980 | 0.950 |

| Task | CLAPScore$_{audio}$ | | CLAPScore$_{text}$ | | GEMINI 2.5 | | IAA | |
|------|---------------------|--------|--------------------|--------|------------|--------|-------|-------|
| | agree | corr | agree | corr | agree | corr | agree | corr |
| Sound Begin-End | **0.925** | **0.951** | 0.825 | 0.687 | 0.625 | 0.204 | 0.967 | 0.933 |
| Sound Inclusion | **0.850** | **0.711** | 0.800 | 0.564 | 0.650 | 0.207 | 0.925 | 0.856 |
| Sound Knowledge | **0.944** | **0.817** | 0.861 | 0.534 | 0.639 | 0.439 | 0.917 | 0.720 |
| Sound Silence | **0.975** | **0.946** | 0.975 | 0.946 | 0.950 | 0.690 | 1.000 | 1.000 |
| Instrument Inclusion | **0.967** | **0.894** | 0.967 | 0.894 | 0.967 | 0.894 | 1.000 | 1.000 |
| Instrument Exclusion | **0.893** | **0.663** | 0.325 | 0.189 | 0.825 | 0.378 | 0.929 | 0.782 |
| Music Genre | **0.900** | **0.764** | 0.775 | 0.435 | 0.750 | 0.355 | 0.929 | 0.782 |
| Average | **0.923** | **0.821** | 0.790 | 0.607 | 0.772 | 0.452 | 0.956 | 0.882 |

| Task | WavLM | | Wav2Vec | | IAA | |
|------|-------|--------|---------|--------|-------|-------|
| | agree | corr | agree | corr | agree | corr |
| Voice Attribution | - | - | **0.949** | **0.826** | 0.950 | 0.844 |
| Voice Replication | **0.875** | **0.731** | - | - | 0.925 | 0.843 |
| Speech Multi-lingual | - | - | **0.966** | **0.876** | 0.925 | 0.856 |
| Conversation | **0.850** | **0.630** | - | - | 0.925 | 0.819 |
| Average | **0.863** | **0.681** | **0.957** | **0.851** | 0.931 | 0.841 |

Table 8: Agreement and Pearson correlation of MMMG evaluation with human annotations. "IAA" stands for inter-annotator agreement, "agree" stands for agreement and "corr" stands for Pearson correlation. We report Word Accuracy for text rendering, text editing and OCR tasks. Best results are in **bold**. MMMG achieves an average best human agreement of 0.944 with average inter-annotator agreement being 0.971. GPT-4O is the most human-aligned image evaluation model while CLAPScore$_{audio}$ is the most human-aligned audio evaluation method.

We report the agreement and Pearson correlation of MMMG with human annotation per task in Table 8. We exclude DreamSim and Whisper as they are widely recognized as established "silver" standards (Huang et al., 2025; Mehrish et al., 2023).

## D.2 FULL BENCHMARKING RESULTS

Evaluation results of 29 multimodal generation models on 55 tasks are listed in Table 9, Table 10, Table 11 and Table 12, categorized by modalities. We report the following additional findings:

- Although image generation models generally maintain consistent rankings across various tasks, certain models exhibit notable weaknesses in specific areas. For instance, FLUX 1.1 PRO performs particularly poorly when tasked with including unrelated objects in a scene, whereas IMAGEN 3 struggles significantly with text rendering. These observations underscore the effectiveness of MMMG in pinpointing specific model weakness.

- When comparing different interleaved image-text agent models, GEMINI 2.5 demonstrates superior planning capabilities over GPT-4O, resulting in a 51.8% performance improvement with the image generator GPT IMAGE.

- Unified understanding-generation models such as JANUS (Chen et al., 2025d) are excluded from our interleaved image-text evaluation due to their requirement for manual modality selection, limiting their capability for automated, interleaved generation tasks. We also notice that models like ANOLE and SEED-LLAMA trained only on individual image generation and image understanding tasks can't follow instructions at all for interleaved image-text input. This highlight the importance of collecting more comprehensive image-text interleaved dataset for training.

- The natural speech-text interleaved model SPIRIT LM rarely scores above zero on evaluated tasks, suggesting it lacks adequate instruction tuning and consequently struggles to follow instructions effectively. Models like GPT-4O-AUDIO and QWEN2.5-OMNI (Xu et al., 2025) doesn't support customized speaker voice, thus can not be evaluated. Models like YUE, which are designed for text-to-song generation, may face challenges when are required to generate pure music.

| Task | IMAGEN 3 | RECRAFT v3 | LUMA PHOTON | FLUX 1.1 PRO | IDEO-GRAM 2 | DALLE 3 | SD 3.5 | GEMINI 2 IMAGE | GPT IMAGE | BILP-o3 | JANUS PRO | GEMINI 2.5 IMAGE |
|---|---|---|---|---|---|---|---|---|---|---|---|---|
| Object Inclusion | 0.838 | 0.688 | 0.831 | 0.444 | 0.863 | 0.788 | 0.544 | 0.844 | 0.869 | 0.475 | 0.706 | **0.919** |
| Object Exclusion | 0.338 | 0.300 | 0.425 | 0.325 | 0.469 | 0.244 | 0.013 | 0.281 | **0.819** | 0.138 | 0.063 | 0.681 |
| Object Count | 0.269 | 0.319 | 0.369 | 0.375 | 0.319 | 0.119 | 0.256 | 0.356 | **0.569** | 0.138 | 0.263 | 0.450 |
| Object Knowledge | 0.494 | 0.481 | 0.656 | 0.325 | 0.419 | 0.463 | 0.150 | **0.706** | 0.531 | 0.238 | 0.081 | 0.588 |
| Object Commonsense | 0.256 | 0.306 | 0.288 | 0.206 | 0.288 | 0.288 | 0.306 | 0.275 | 0.163 | 0.194 | 0.175 | **0.363** |
| Object Attribution | 0.325 | 0.206 | 0.319 | 0.256 | 0.275 | 0.294 | 0.244 | 0.375 | **0.619** | 0.331 | 0.363 | 0.475 |
| Comparison Relation | 0.588 | 0.288 | 0.488 | 0.375 | 0.475 | 0.388 | 0.150 | 0.450 | 0.600 | 0.338 | 0.200 | **0.650** |
| Universal Relation | 0.425 | 0.538 | 0.638 | 0.463 | 0.500 | 0.375 | 0.350 | 0.450 | **0.813** | 0.238 | 0.325 | 0.700 |
| Relative Spatial | 0.838 | 0.625 | 0.875 | 0.663 | 0.738 | 0.550 | 0.575 | 0.750 | **0.988** | 0.563 | 0.588 | 0.900 |
| Absolute Spatial | 0.488 | 0.388 | 0.700 | 0.488 | 0.450 | 0.225 | 0.338 | **0.700** | 0.675 | 0.363 | 0.563 | 0.588 |
| Region Fill | 0.484 | 0.236 | 0.628 | 0.442 | 0.375 | 0.207 | 0.320 | 0.683 | 0.762 | 0.550 | 0.415 | **0.788** |
| Border Fill | 0.279 | 0.353 | 0.528 | 0.349 | 0.273 | 0.350 | 0.267 | 0.450 | **0.651** | 0.520 | 0.363 | 0.368 |
| Single TR | 0.827 | 0.994 | 0.936 | 0.901 | 0.995 | 0.661 | 0.811 | 0.997 | **1.000** | 0.141 | 0.544 | 0.990 |
| Double TR | 0.313 | 0.422 | 0.686 | 0.528 | 0.701 | 0.215 | 0.325 | 0.745 | 0.763 | 0.008 | 0.006 | **0.780** |
| Multi-lingual TR | 0.351 | 0.471 | 0.440 | 0.326 | 0.483 | 0.120 | 0.330 | **0.817** | 0.784 | 0.000 | 0.261 | 0.573 |
| Average | 0.474 | 0.441 | 0.587 | 0.431 | 0.508 | 0.352 | 0.332 | 0.592 | **0.707** | 0.282 | 0.328 | 0.654 |

Table 9: Benchmarking results of 10 models on 15 image generation tasks. Best results are in **bold**. GPT-4O significantly outperforms other image generation models.

## D.3 ANALYSIS

**Interleaved System Prompt.** To investigate whether autoregressive models' capabilities in generating the desired number and order of modalities can be improved, we conducted experiments with GEMINI 2 IMAGE using the planning system prompt detailed in Table 17. The experimental results, summarized in Table 18, indicate that incorporating system prompts emphasizing modality count and order does not consistently lead to positive outcomes. Generally, adding a system prompt negatively impacts image generation quality, as the models shift their focus away from optimizing visual quality. Conversely, image editing tasks benefit from the addition of system prompts since without such prompts, models frequently generate multiple images unnecessarily. Nonetheless, system prompts do not effectively support generating sequential images or integrated image-text pairs, because models continue to intermix multiple images during generation, as illustrated in Figure 3.

**Variance Control.** To validate the evaluation robustness of MMMG, we present the 95% confidence intervals for each task in Table 13, Table 14, Table 15 and Table 16. A sample size of 4 can

| Task | SEED LLAMA | ANOLE | GPT-4O + GPT IMAGE | GEMINI 2.5 + GPT IMAGE | GEMINI 2 IMAGE | GPT IMAGE | GEMINI 2.5 IMAGE |
|---|---|---|---|---|---|---|---|
| Semantic Consistency | 0.000 | 0.000 | 0.613 | **0.763** | 0.013 | 0.675 | 0.325 |
| Multi-angle Consistency | 0.000 | 0.000 | 0.230 | **0.461** | 0.352 | 0.448 | 0.367 |
| Multi-view Consistency | 0.000 | 0.000 | 0.064 | **0.221** | 0.143 | 0.188 | 0.191 |
| Composition Consistency | 0.000 | 0.000 | **0.800** | 0.738 | 0.000 | 0.075 | 0.225 |
| Decomposition Consistency | 0.000 | 0.000 | 0.600 | **0.875** | 0.013 | 0.575 | 0.638 |
| Self Count | 0.000 | 0.038 | 0.100 | **0.850** | 0.213 | 0.763 | 0.000 |
| Self Color Recognition | 0.000 | 0.000 | 0.663 | 0.700 | 0.000 | **0.713** | 0.088 |
| Self Size Recognition | 0.000 | 0.000 | 0.338 | 0.600 | 0.263 | **0.675** | 0.463 |
| Self Text Recognition | 0.000 | 0.000 | 0.312 | **0.958** | 0.101 | 0.674 | 0.569 |
| Self Relative Spatial | 0.000 | 0.000 | 0.538 | **0.725** | 0.250 | 0.425 | 0.363 |
| Self Absolute Spatial | 0.000 | 0.000 | 0.475 | 0.775 | 0.100 | **0.825** | 0.425 |
| Text-image Order Control | 0.150 | 0.100 | 0.913 | **0.925** | 0.725 | 0.500 | 0.725 |
| Interleaved Adding | 0.154 | 0.052 | 0.394 | 0.394 | 0.545 | 0.370 | **0.680** |
| interleaved Altering | 0.179 | 0.033 | 0.566 | 0.573 | 0.609 | 0.495 | **0.868** |
| Text Adding | 0.000 | 0.000 | 0.097 | 0.410 | 0.444 | 0.421 | **0.501** |
| Text Altering | 0.046 | 0.100 | 0.077 | 0.356 | 0.182 | 0.335 | **0.545** |
| Object Adding | 0.165 | 0.190 | 0.470 | 0.631 | 0.748 | 0.635 | **0.828** |
| Object Removing | 0.350 | 0.175 | 0.415 | 0.540 | 0.605 | 0.508 | **0.727** |
| Object Replacing | 0.109 | 0.121 | 0.453 | 0.627 | 0.487 | 0.650 | **0.722** |
| Object Altering | 0.142 | 0.000 | 0.192 | 0.352 | 0.316 | 0.360 | **0.503** |
| Interleaved Math | 0.000 | 0.000 | 0.025 | **0.038** | 0.000 | 0.000 | 0.000 |
| Interleaved Code | 0.000 | 0.000 | 0.071 | **0.224** | 0.136 | 0.202 | 0.215 |
| Average | 0.059 | 0.037 | 0.382 | **0.579** | 0.284 | 0.478 | 0.453 |

Table 10: Benchmarking results of 6 models on 22 image-text interleaved generation tasks. Best results are in **bold**. Agent model GEMINI 2.5 PRO + GPT IMAGE is the best combination for consistent image sequence and coherent image-text pair generation. GEMINI 2.5 IMAGE as a modality-unified autoregressive model, performs best at image editing tasks.

| Task | STABLE AUDIO | AUDIO LDM 2 | AUDIOGEN | MAKE-AN -AUDIO 2 | TANGO 2 | MUSICGEN | TANGO MUSIC | YUE |
|---|---|---|---|---|---|---|---|---|
| Sound Begin-End | 0.525 | 0.450 | 0.475 | **0.631** | 0.525 | - | - | - |
| Sound Inclusion | **0.700** | 0.413 | 0.450 | 0.575 | 0.513 | - | - | - |
| Sound Reasoning | 0.014 | 0.014 | 0.042 | **0.611** | 0.194 | - | - | - |
| Sound Silence | 0.063 | 0.019 | 0.019 | **0.131** | 0.006 | - | - | - |
| Instrument Inclusion | 0.817 | 0.833 | - | - | - | 0.833 | **0.950** | 0.600 |
| Instrument Exclusion | 0.225 | 0.163 | - | - | - | 0.200 | 0.050 | **0.525** |
| Music Genre | 0.488 | 0.950 | - | - | - | 0.625 | **0.925** | 0.000 |
| Music Tempo | 0.200 | 0.010 | - | - | - | **0.620** | 0.080 | 0.040 |
| Music Intensity | **0.188** | 0.013 | - | - | - | 0.038 | 0.075 | 0.000 |
| Average | 0.358 | 0.318 | 0.246 | **0.487** | 0.310 | **0.463** | 0.416 | 0.233 |

Table 11: Benchmarking results of 8 models on 9 sound and music generation tasks. MAKE-AN-AUDIO 2 is the best audio generation model and the only model that can perform sound reasoning task; MUSICGEN is the best music generation model and the only model that can have tempo control.

| Task | GEMINI 2.5 + VOXINSTRUCT | GEMINI 2.5 + VOICELDM | SPIRIT LM |
|---|---|---|---|
| Voice Attribution | **0.684** | 0.568 | 0.000 |
| Voice Replication | **0.625** | 0.109 | 0.002 |
| Speech Multi-lingual | **0.654** | - | - |
| Transcript Generation | **0.638** | 0.438 | 0.200 |
| Transcript Editing | 0.200 | **0.350** | 0.000 |
| Conversation Generation | **0.788** | 0.375 | 0.000 |
| Speech Translation | 0.155 | **0.269** | 0.000 |
| Speech Retrieval | 0.188 | **0.421** | 0.007 |
| Audio-Text Order Control | **0.520** | 0.362 | 0.023 |
| Average | **0.620** | 0.427 | 0.034 |

Table 12: Benchmarking results of 3 models on 9 speech-text interleaved generation tasks. Best results are in **bold**. Natural speech-text interleaved model SPIRIT LM does not have instruction following capability and get zero for most tasks. VOXINSTRUCT is the best multi-functional speech synthesizer.

substantially reduce variance, with the average relative confidence interval of all tasks and models being 4.05%. While some individual models show higher variance on certain tasks, this reflects the inherent robustness differences of the models themselves rather than evaluation instability.

| Task | IMAGEN 3 | RECRAFT v3 | LUMA PHOTON | FLUX 1.1 PRO | IDEO-GRAM 2 | DALLE 3 | SD 3.5 | GEMINI 2 IMAGE | GPT IMAGE | BILP-o3 | JANUS PRO | Average |
|---|---|---|---|---|---|---|---|---|---|---|---|---|
| Object Inclusion | 4.24 | 3.16 | 5.05 | 5.43 | 4.24 | 4.24 | 5.79 | 3.08 | 3.67 | 4.90 | 3.08 | 4.26 |
| Object Exclusion | 4.24 | 5.29 | 7.21 | 5.29 | 6.44 | 2.35 | 6.44 | 1.22 | 4.24 | 1.41 |  | 4.24 |
| Object Count | 3.08 | 2.35 | 4.18 | 5.29 | 5.05 | 6.75 | 1.23 | 5.43 | **5.43** | 5.10 | 3.16 | 4.28 |
| Object Knowledge | 1.23 | 3.08 | 2.35 | 2.00 | 3.67 | 5.10 | 7.21 | 1.23 | 5.05 | 5.83 | 2.35 | 3.35 |
| Object Commonsense | 2.35 | 6.44 | 4.69 | 2.35 | 1.41 | 4.69 | 3.67 | 2.00 | 4.24 | 4.64 | 6.93 | 3.95 |
| Object Attribution | 3.46 | 3.08 | 5.05 | 5.79 | 7.75 | 1.22 | 4.47 | 3.08 | 2.35 | 2.45 |  | 3.85 |
| Comparison Relation | 2.45 | 2.45 | 10.10 | 6.33 | 8.49 | 8.37 | **6.93** | 6.93 | 4.00 | 8.37 | 8.00 | **6.85** |
| Universal Relation | 2.83 | 7.35 | 6.17 | 2.45 | **12.00** | 12.96 | 5.66 | 6.93 | 2.45 | 6.17 | **9.38** | 6.76 |
| Relative Spatial | 4.69 | 6.33 | 6.33 | **10.87** | 4.69 | 6.93 | 2.83 | 6.93 | 2.45 | 10.10 | 8.37 | 6.41 |
| Absolute Spatial | 6.17 | 4.69 | 6.93 | 4.69 | 5.66 | 12.96 | 2.45 | 6.93 | 4.90 | **10.87** | 6.17 | 6.58 |
| Region Fill | 4.56 | 0.92 | 5.26 | 7.49 | 5.31 | 7.17 | 5.06 | 4.09 | 3.28 | 2.04 | 4.34 | 4.50 |
| Border Fill | 3.26 | 7.54 | 1.36 | 8.39 | 6.51 | 5.52 | 2.70 | **8.19** | 4.07 | 5.58 | 1.62 | 4.98 |
| Single TR | 5.36 | 1.23 | 6.13 | 4.48 | 0.98 | 8.81 | 7.46 | 0.61 | 0.00 | 3.76 | 5.66 | 4.04 |
| Double TR | 2.49 | **11.66** | 5.48 | 9.87 | 5.98 | 6.20 | 4.18 | 1.91 | 0.81 | 0.30 | 0.68 | 4.50 |
| Multi-lingual TR | **6.69** | 2.52 | 1.47 | 1.98 | 1.12 | 2.82 | 5.71 | 3.94 | 1.23 | 0.00 | 8.20 | 3.24 |
| Average | 3.81 | 4.54 | 5.18 | 5.51 | 5.29 | **6.41** | 4.47 | 4.61 | 3.06 | 4.95 | 4.79 | 4.78 |

Table 13: 95% relative confidence intervals of 9 models on 15 image generation tasks. The numbers are in percentile. Highest CIs are in **bold**.

| Task | SEED LLAMA | ANOLE | GPT-4o + GPT IMAGE | GEMINI 2.5 + GPT IMAGE | GEMINI 2 IMAGE | GPT IMAGE | Average |
|---|---|---|---|---|---|---|---|
| Semantic Consistency | 0.00 | 0.00 | 7.35 | 2.45 | 2.45 | 6.33 | 3.10 |
| Multi-angle Consistency | 0.00 | 0.00 | 2.54 | 0.55 | 4.30 | 2.69 | 1.68 |
| Multi-view Consistency | 0.00 | 0.00 | 1.03 | 0.20 | 2.01 | 0.78 | 0.67 |
| Composition Consistency | 0.00 | 0.00 | 6.93 | **9.28** | 0.00 | 6.33 | 3.76 |
| Decomposition Consistency | 0.00 | 0.00 | 6.93 | 2.83 | 2.45 | 6.33 | 3.09 |
| Self Count | 0.00 | 2.45 | 6.93 | 6.93 | 4.69 | 8.37 | 4.89 |
| Self Color Recognition | 0.00 | 0.00 | 2.45 | 5.66 | 0.00 | 4.69 | 2.13 |
| Self Size Recognition | 0.00 | 0.00 | 15.17 | 6.93 | 6.17 | 6.33 | 5.77 |
| Self Text Recognition | 0.00 | 0.00 | 2.13 | 0.56 | 3.02 | **10.68** | 2.73 |
| Self Relative Spatial | 0.00 | 0.00 | **16.19** | 4.90 | 4.00 | 6.33 | **5.24** |
| Self Absolute Spatial | 0.00 | 0.00 | 11.66 | 2.83 | 4.00 | 4.90 | 3.90 |
| Text-image Order Control | 0.00 | **4.00** | 4.69 | 2.83 | 6.33 | 0.00 | 2.97 |
| Interleaved Adding | 0.08 | 0.71 | 0.75 | 0.93 | 2.56 | 1.84 | 1.14 |
| interleaved Altering | 0.07 | 1.04 | 1.18 | 1.60 | 3.33 | 1.29 | 1.42 |
| Text Adding | 0.00 | 0.00 | 1.57 | 1.46 | 0.91 | 1.32 | 0.88 |
| Text Altering | 1.60 | 0.00 | 2.74 | 0.70 | 5.25 | 3.40 | 2.28 |
| Object Adding | 1.38 | 3.61 | 7.25 | 0.38 | 3.50 | 4.36 | 3.41 |
| Object Removing | 1.27 | 3.44 | 4.97 | 1.40 | 5.46 | 2.42 | 3.16 |
| Object Replacing | 2.85 | 3.46 | 3.23 | 4.00 | **9.85** | 2.73 | 4.35 |
| Object Altering | **5.85** | 0.00 | 4.50 | 7.94 | 1.23 | 4.56 | 4.01 |
| Interleaved Math | 0.00 | 0.00 | 4.90 | 2.45 | 0.00 | 0.00 | 1.23 |
| Interleaved Code | 0.00 | 0.00 | 3.22 | 4.81 | 3.23 | 4.85 | 2.68 |
| Average | 0.60 | 0.85 | **5.38** | 3.25 | 3.40 | 4.11 | 2.93 |

Table 14: 95 relative confidence intervals of 6 models on 22 image-text interleaved generation tasks. The numbers are in percentile. Highest CIs are in **bold**.

| Task | STABLE AUDIO | AUDIO LDM 2 | AUDIOGEN | MAKE-AN -AUDIO 2 | TANGO 2 | MUSICGEN | TANGO MUSIC | YUE | Average |
|---|---|---|---|---|---|---|---|---|---|
| Sound Begin-End | 5.66 | 2.83 | **13.12** | 8.95 | 5.25 | - | - | - | **7.16** |
| Sound Inclusion | **10.59** | 8.37 | 5.66 | 2.45 | 5.13 | - | - | - | 6.44 |
| Sound Reasoning | 2.72 | 2.72 | 2.72 | 5.44 | 1.94 | - | - | - | 3.11 |
| Sound Silence | 2.45 | 3.67 | 1.23 | 1.23 | 0.63 | - | - | - | 1.84 |
| Instrument Inclusion | 6.26 | 3.77 | - | 0.00 | - | 3.77 | 3.27 | 0.00 | 2.84 |
| Instrument Exclusion | 2.83 | 7.35 | - | 0.00 | - | 5.66 | 5.66 | **2.83** | 4.05 |
| Music Genre | 9.28 | 5.66 | - | 0.00 | - | **6.33** | 9.38 | 0.00 | 5.11 |
| Music Tempo | 3.20 | 1.96 | - | 0.00 | - | 5.06 | 6.40 | 0.00 | 2.77 |
| Music Intensity | 7.35 | 2.45 | - | 0.00 | - | 4.69 | 6.33 | 0.00 | 3.47 |
| Average | 5.59 | 4.31 | 5.68 | 2.01 | 3.24 | 5.10 | **6.21** | 0.57 | 4.09 |

Table 15: 95 relative confidence intervals of 8 models on 9 sound and music generation tasks. The numbers are in percentile. Highest CIs are in **bold**.

Importantly, when averaged across all models, the maximum 95% CI is only 10.12% of all tasks and 6.41% of all models. The, demonstrating that MMMG provides statistically robust evaluation across the full spectrum of capabilities.

| Task | GEMINI 2.5 + VOXINSTRUCT | GEMINI 2.5 + VOICELDM | SPIRIT LM | Average |
|------|------|------|------|------|
| Voice Attribution | 4.14 | 6.08 | 0.01 | 5.11 |
| Voice Replication | 5.91 | 2.80 | **0.13** | 4.35 |
| Speech Multi-lingual | 4.06 | 0.17 | 0.00 | 2.12 |
| Transcript Generation | 7.35 | **12.89** | 0.00 | **10.12** |
| Transcript Editing | 5.66 | 12.00 | 0.00 | 8.83 |
| Conversation Generation | **7.35** | 8.49 | 0.00 | 7.92 |
| Speech Translation | 0.51 | 1.16 | 0.00 | 0.84 |
| Speech Retrieval | 3.37 | 2.38 | 0.00 | 2.87 |
| Audio-Text Order Control | **6.93** | 6.33 | 0.00 | 6.63 |
| Average | 4.79 | 5.74 | 0.02 | 5.27 |

Table 16: 95 relative confidence intervals of 3 models on 9 speech-text interleaved generation tasks. The numbers are in percentile. Highest CIs are in **bold**.

You are a multimodal assistant capable of generating interleaved text and images based on user instructions.

• Follow the required modality structure and number in user's instruction exactly, especially when multiple images are implied or requested.

• Generate separate images for each described part, do not combine multiple concepts into one image unless told to.

• Interleave images and text in the order described.

Your goal is to match the user's intent with exact number and sequence of image and text.

Table 17: System prompt used to make GEMINI IMAGE output correct modality order and number.

| Task | GEMINI IMAGE w/ prompt | GEMINI IMAGE w/o prompt |
|------|------|------|
| Semantic Consistency | **0.263** | 0.013 |
| Multi-Angel Consistency | 0.135 | **0.352** |
| Multi-View Consistency | 0.094 | **0.143** |
| Compose Consistency | **0.013** | 0.000 |
| Decompose Consistency | 0.000 | **0.013** |
| Interleaved Object Adding | 0.399 | **0.545** |
| Interleaved Color Modifying | 0.486 | **0.609** |
| Text Editing | **0.423** | 0.283 |
| Object Adding | 0.622 | **0.748** |
| Object Removing | 0.485 | **0.605** |
| Object Modifying | 0.468 | **0.487** |
| Self Count | **0.275** | 0.213 |
| Self Color | **0.113** | 0.000 |
| Self Size | 0.188 | **0.263** |
| Self OCR | **0.335** | 0.101 |
| Self Relative Spatial | 0.138 | **0.250** |
| Self Absolute Spatial | **0.175** | 0.100 |
| Interleaved Math | **0.000** | **0.000** |
| Interleaved Code | 0.110 | **0.136** |
| Image-Text Order | **0.725** | **0.725** |
| Average | 0.273 | **0.279** |

| Task | GEMINI IMAGE w/ prompt | GEMINI IMAGE w/o prompt |
|------|------|------|
| Object Inclusion | **0.888** | 0.875 |
| Object Exclusion | **0.400** | 0.313 |
| Object Count | **0.500** | 0.450 |
| Object Reasoning | 0.813 | 0.825 |
| Object Attribution | **0.475** | **0.475** |
| Comparison Relation | **0.475** | 0.450 |
| Universal Relation | **0.488** | 0.450 |
| Relative Spacial Relation | **0.850** | 0.750 |
| Absolute Spacial Relation | **0.738** | 0.700 |
| Region Fill | 0.585 | **0.683** |
| Border Fill | **0.459** | 0.450 |
| Single Text Rendering | 0.945 | **0.997** |
| Double Text Rendering | **0.800** | 0.745 |
| Multi-lingual Text Rendering | 0.691 | **0.817** |
| Average | **0.650** | 0.641 |

Table 18: Comparison of GEMINI IMAGE performance with and without system prompt on image generation (right) and interleaved image-text generation (left) tasks. Best results are in **bold**. System prompt does not always have positive impact.

# E  EXAMPLES AND FAILURE ANALYSIS

We provide examples of each task from Figure 5 to Figure 53.

## Object Inclusion

**Instruction**: Generate an image of a crowded beach. Please include a single snowman in the image.

---

**Good Example:** FLUX 1.1 PRO

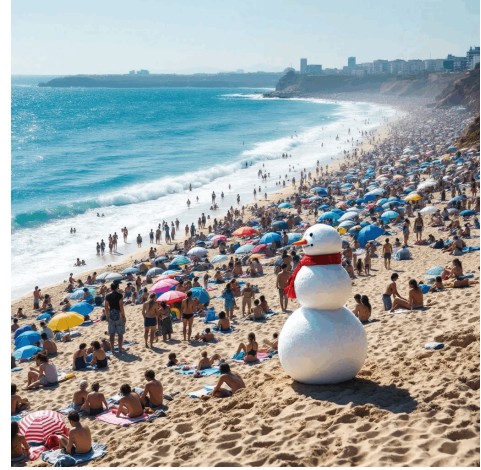

**Evaluation:** ✔

**Bad Example:** IMAGEN 3

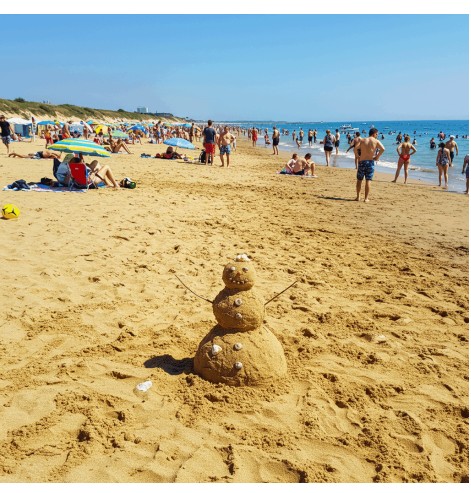

**Evaluation:** ✘
**Analysis**: It's not a snowman but a "sand-man", affected by its context of a beach.

Figure 5: Examples for the task: Object Inclusion

## Object Exclusion

**Instruction**: Generate an image of a birthday party. Do not include cakes in the image.

---

**Good Example:** IDEOGRAM 2

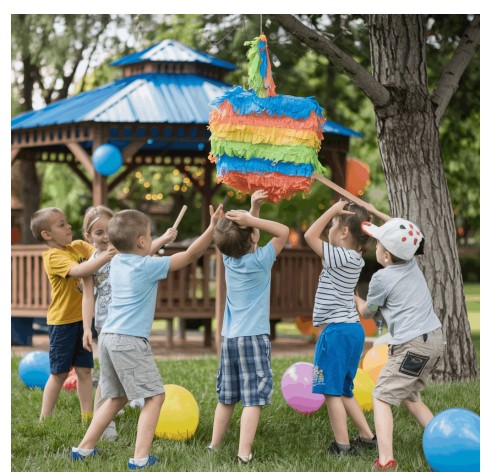

**Evaluation:** ✔

**Bad Example:** DALLE 3

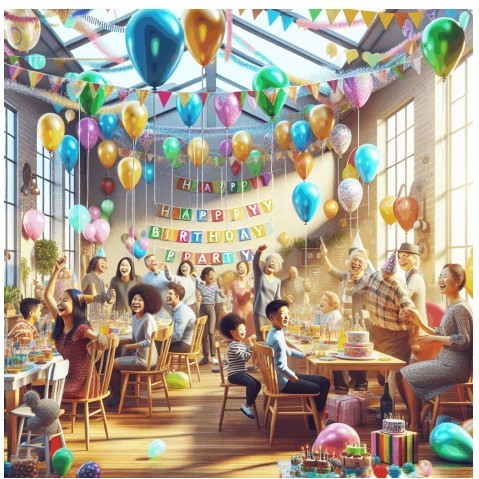

**Evaluation:** ✘
**Analysis**: It fails to exclude cakes in the context of a birthday party.

Figure 6: Examples for the task: Object Exclusion

Object Count

**Instruction**: Generate an image of a race track with 3 clearly visible race cars, evenly spaced and not overlapping. The race cars should be of regular size and common shapes.

**Good Example:** RECRAFT V3

**Bad Example:** SD 3.5

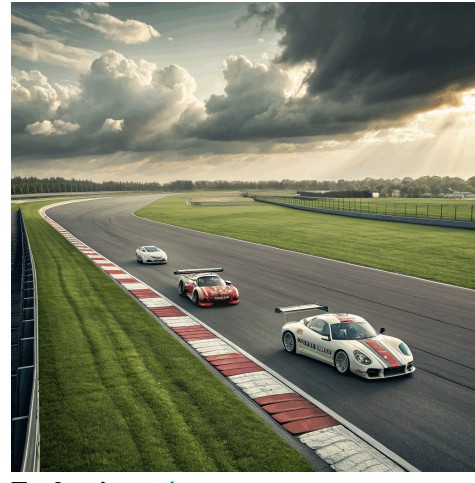
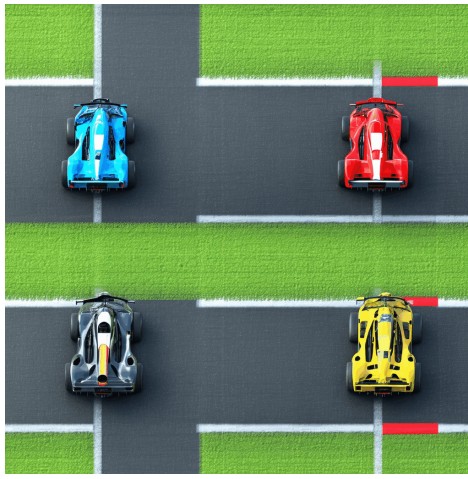

**Evaluation:** ✔

**Evaluation:** ✘
**Analysis**: Neither the number of tracks nor the number fo cars is correct.

Figure 7: Examples for the task: Object Count

Object Reasoning

**Instruction**: Generate an image of a single fruit named after a flightless bird native to New Zealand. Do not add additional elements or details in the background.

**Good Example:** IMAGEN 3

**Bad Example:** SD 3.5

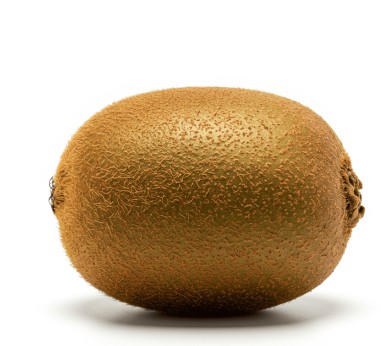
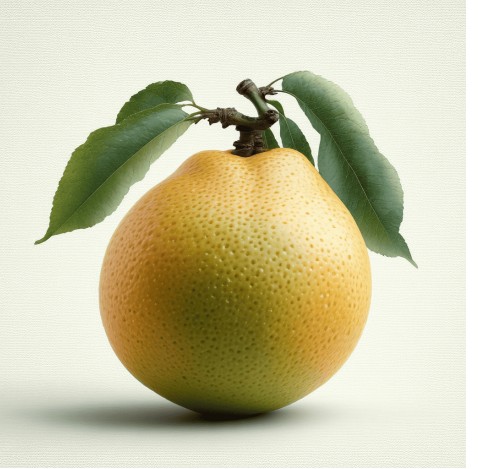

**Evaluation:** ✔

**Evaluation:** ✘
**Analysis**: Wrong object.

Figure 8: Examples for the task: Object Reasoning

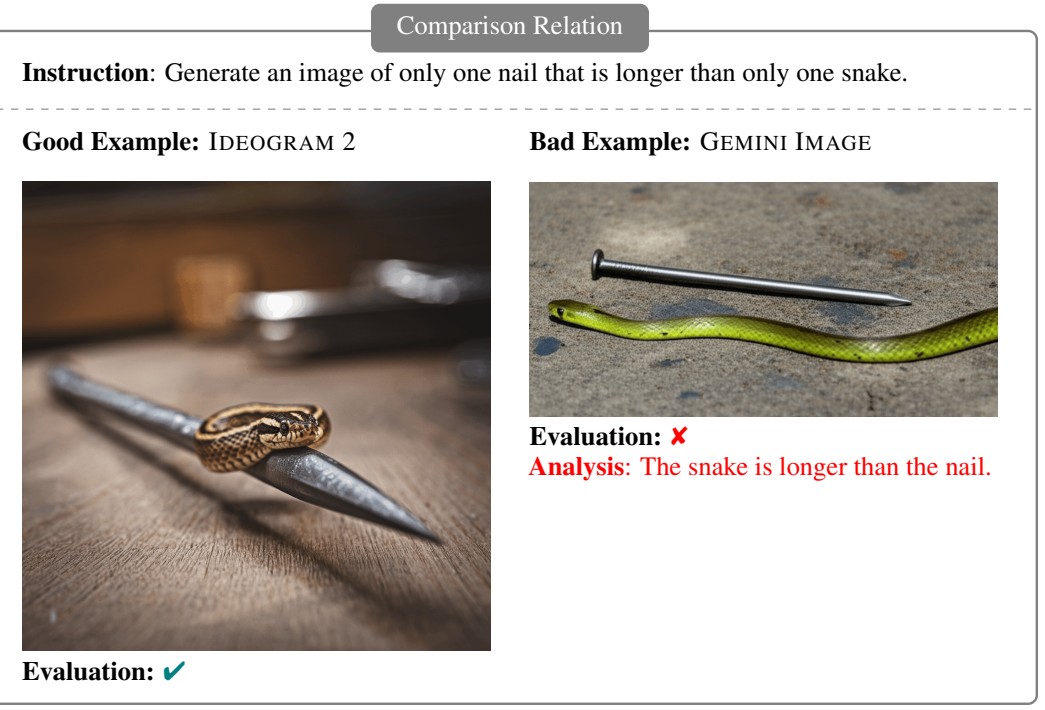

Figure 9: Examples for the task: Object Attribution

Figure 10: Examples for the task: Comparison Relation

Universal Relation

**Instruction**: Generate an image of a toy box where all toys are animals except one car.

**Good Example:** GPT IMAGE

**Evaluation:** ✔

**Bad Example:** RECRAFT V3

**Evaluation:** ✗
**Analysis**: Have two cars instead of one.

**Bad Example:** FLUX 1.1 PRO

**Evaluation:** ✗
**Analysis**: Have building blocks which are not allowed in the instruction.

Figure 11: Examples for the task: Universal Relation

**Absolute Spatial Relation**

**Instruction**: Generate an image of a countryside porch, with a single rocking chair at the bottom left quarter of the image and a single lantern at the up right quarter of the image.

**Good Example:** GEMINI IMAGE

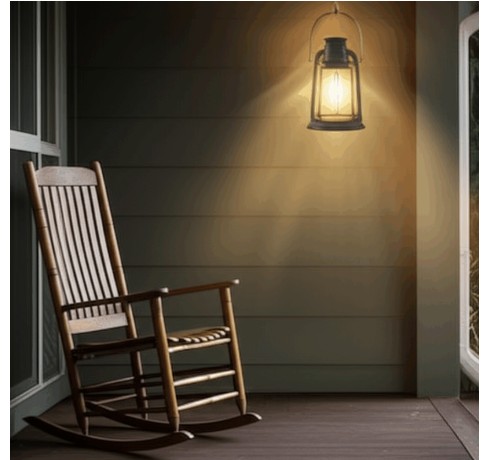

**Evaluation:** ✔

**Bad Example:** DALLE 3

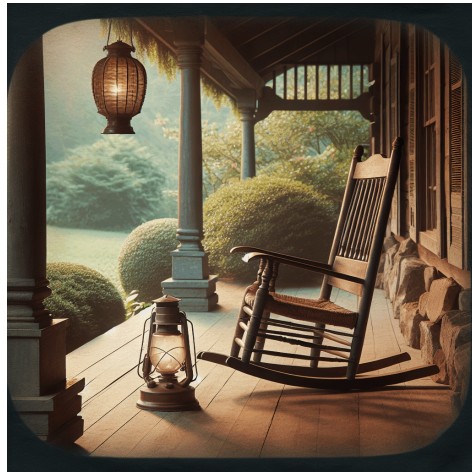

**Evaluation:** ✗

**Analysis**: The number of lanterns is two. Generating wrong number of objects is a prevalent failure mode for all image generation tasks other than obejct count task.

**Bad Example:** SD 3.5

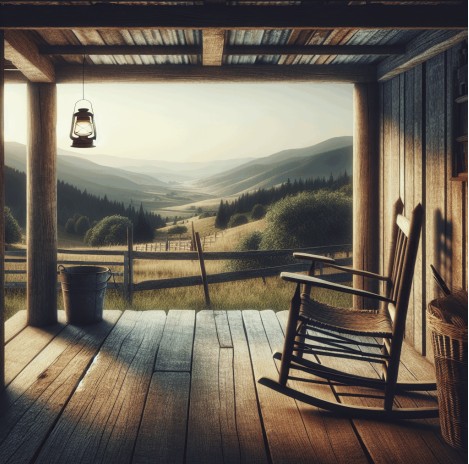

**Evaluation:** ✗

**Analysis**: The latten is at the left-up quarter and the chair at bottom-right.

Figure 12: Examples for the task: Absolute Spatial Relation

---

**Relative Spatial Relation**

**Instruction**: Generate an image about a quiet library. Please include a single wooden chair and and a single lamp. Make sure that the wooden chair is to the right of the lamp.

---

**Good Example:** DALLE 3

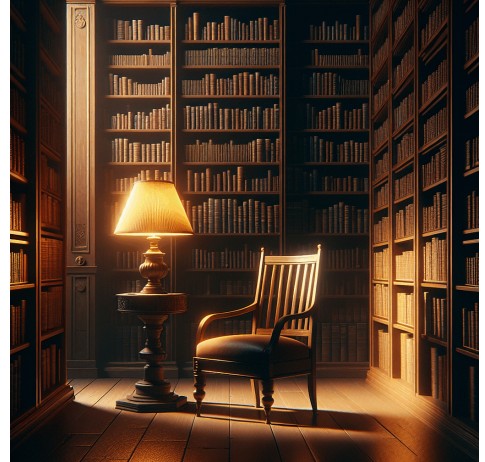

**Evaluation:** ✔

**Bad Example:** SD 3.5

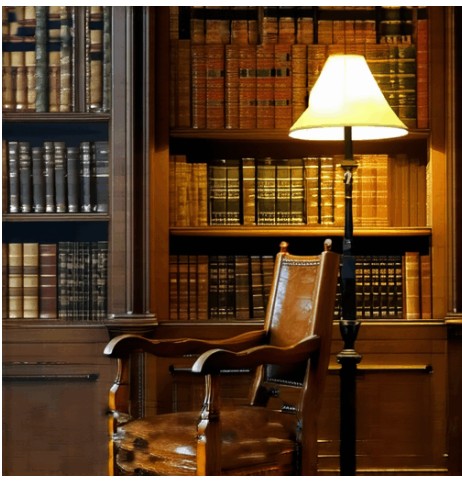

**Evaluation:** ✘
**Analysis**: The wooden chair is to the left of the lamb.

Figure 13: Examples for the task: Relative Spatial Relation

---

**Region Fill**

**Instruction**: Generate an image about a bustling city street at dusk. The left half should be pure green.

---

**Good Example:** GPT IMAGE

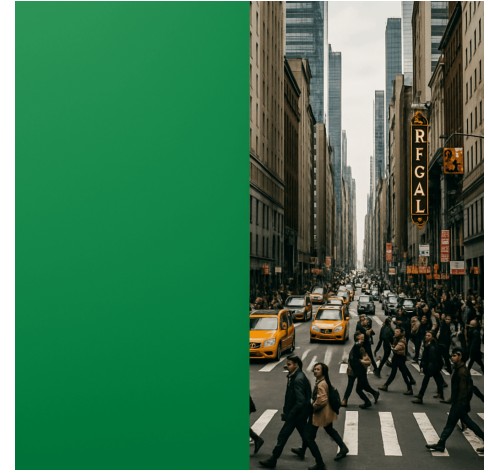

**Evaluation:** 0.892 (similarity score)

**Bad Example:** SD 3.5

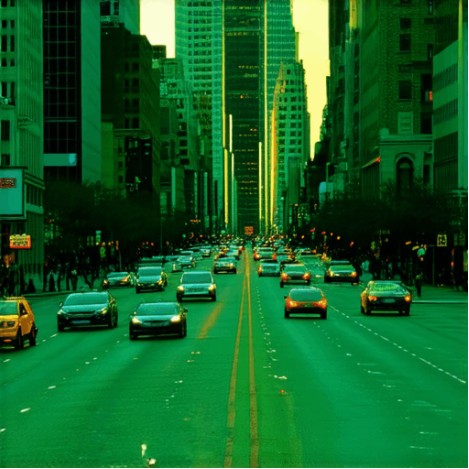

**Evaluation:** 0.327 (similarity score)
**Analysis**: The whole image is green.

Figure 14: Examples for the task: Region Fill

## Border Fill

**Instruction**: Generate a serene mountain landscape at sunset. The entire image should be surrounded by a simple and flat, solid and green border and approximately 10% of the image width on all sides.

**Good Example:** GPT IMAGE

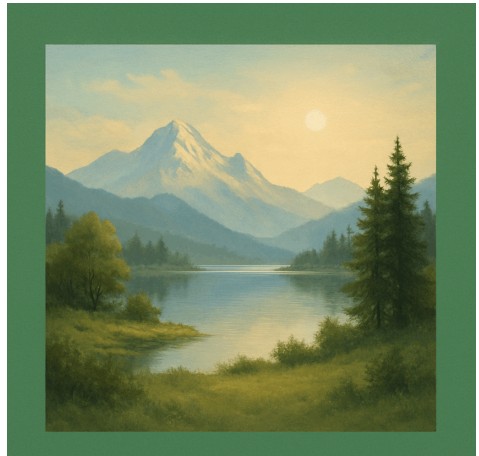

**Evaluation:** 0.859 (similarity score)

**Bad Example:** GEMINI IMAGE

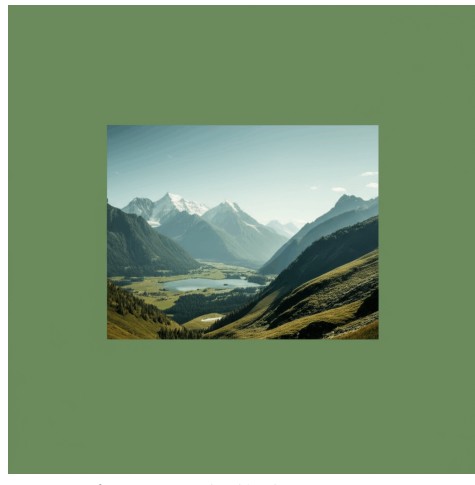

**Evaluation:** 0.0 (similarity score)
**Analysis**: The border is too wide
**Bad Example:** IDEOGRAM 2

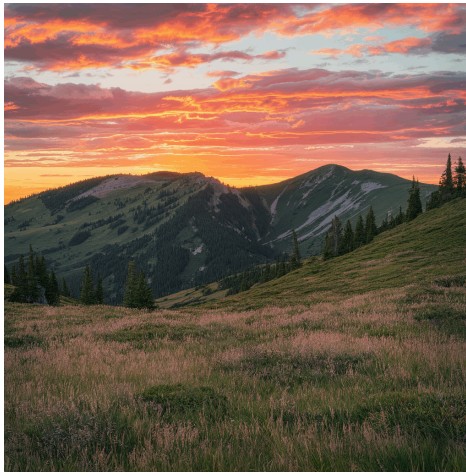

**Evaluation:** 0.0 (similarity score)
**Analysis**: There is no border.

Figure 15: Examples for the task: Border Fill

## Single Text Rendering

**Instruction**: Generate an image of a sign and the only text on it is "Love The World Forever". The text should be written in a clear, standard, easily readable typeface without any artistic distortions.

**Good Example:** RECRAFT V3

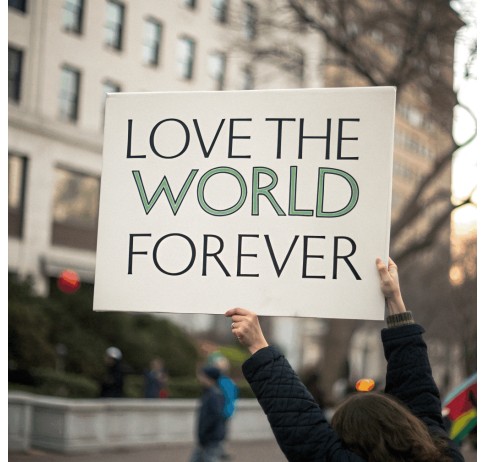

**Bad Example:** DALLE 3

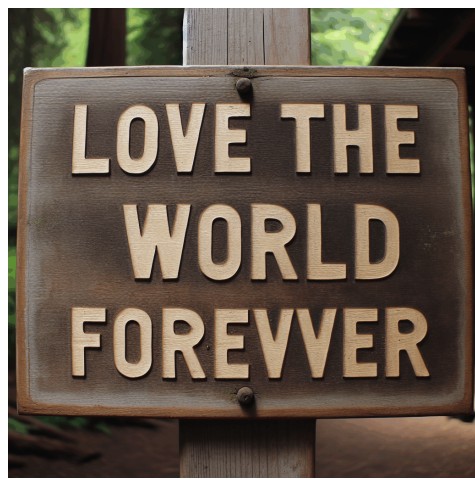

**Evaluation:** ✔

**Evaluation:** ✘
**Analysis**: The text rendered ("love the world forevver") is wrong.
**Bad Example:** IDEOGRAM 2

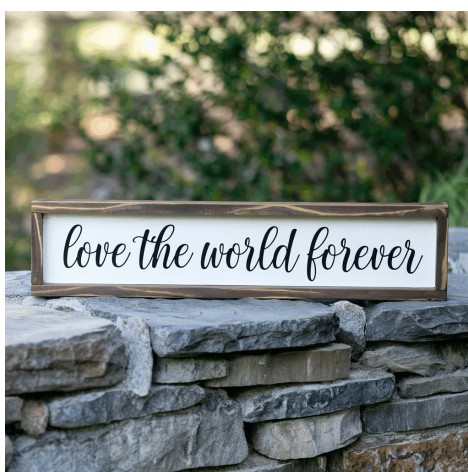

**Evaluation:** ✘
**Analysis**: Artistic distortion makes it hard to recognize.

Figure 16: Examples for the task: Single Text Rendering

## Double Text Rendering

**Instruction**: Generate an image of exactly one sticky note on the left side of a white desk and exactly one coffee mug on the right. The sticky note should have "Stay bold, chase dreams" written on it while the coffee mug should have "Coffee fuels infinite morning possibilities" printed on it. The text should be written in a clear, standard, easily readable typeface without any artistic distortions. The objects should be clearly visible and avoid overlapping.

**Good Example:** GPT IMAGE

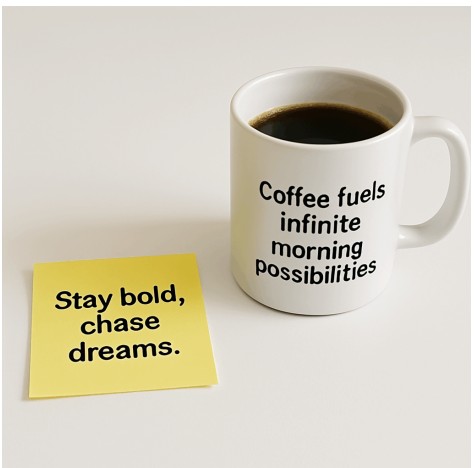

**Evaluation:** ✔

**Bad Example:** IMAGEN 3

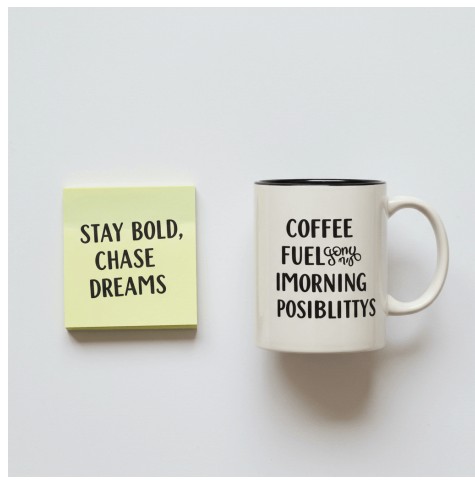

**Evaluation:** ✘
**Analysis**: The text rendered on the coffee mug is wrong.

**Bad Example:** RECRAFT V3

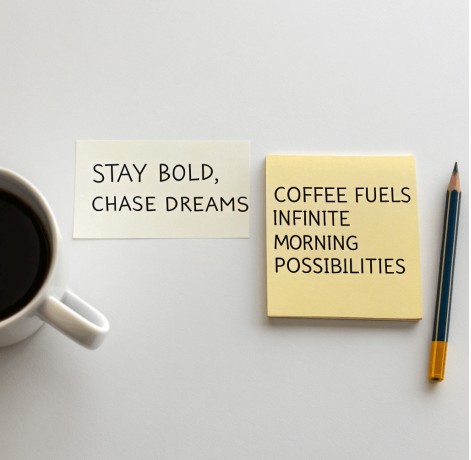

**Evaluation:** ✘
**Analysis**: The text is not rendered on the coffee mug.

**Bad Example:** RECRAFT V3

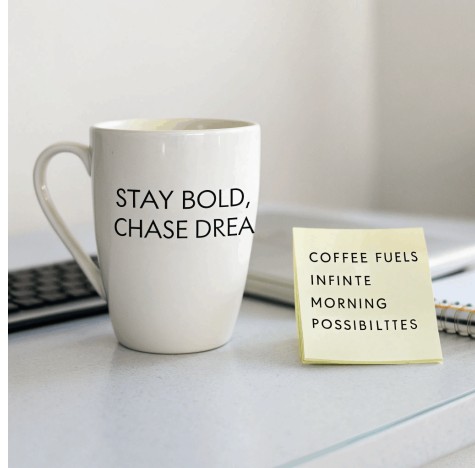

**Evaluation:** ✘
**Analysis**: The coffee mug is to the left of the sticky note.

Figure 17: Examples for the task: Double Text Rendering

Multi-Lingual Text Rendering

**Instruction**: Generate an image of a mug and the only text on it is "美好". The text should be written in a clear, standard, easily readable typeface without any artistic distortions.

**Good Example:** GPT IMAGE          **Bad Example:** RECRAFT V3

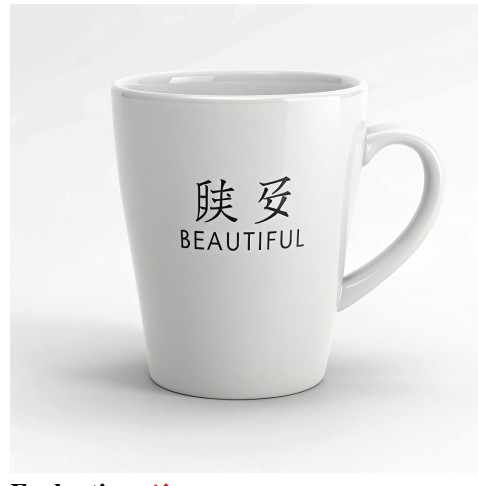

**Evaluation:** ✔

**Evaluation:** ✘
**Analysis**: Distorted Chinese-like characters. In fact, all other models fails in generating Chinese except GPT IMAGE and GEMINI IMAGE.

Figure 18: Examples for the task: Multi-Lingual Text Rendering

---

**Semantic Consistency**

**Instruction**: Create four images of traditional dice faces, each displaying a single face with an increasing number of pips, starting from two up to the highest number.

- - - - - - - - - - - - - - - - - - - - - - - - - - - - - - - - - - - - - - - - - - -

**Good Example:** GPT-4O + GPT IMAGE
**Response:** Here are the images of traditional dice faces, each displaying a single face with an increasing number of pips:

1. Two pips:

2. Three pips:

3. Four pips:

2. Five pips:

**Evaluation:** ✔

**Bad Example:** GEMINI IMAGE
**Response**: I will generate four images showing individual faces of a standard six-sided die. The first image will show the '2' face, the second will show the '3' face, the third will show the '4' face, and the final image will show the '5' face. Each face will be depicted with the traditional arrangement of white pips on a black background.

**Evaluation:** ✗
**Analysis**: Generate only one image; pip numbers are inconsistent with textual response; the last face is a number rather than pips.

Figure 19: Examples for the task: Semantic Consistency

---

### Multi-Angle Consistency

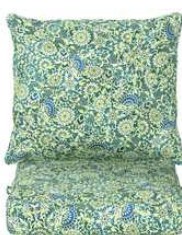

**Instruction**: Using the provided image as the reference angle, create four additional images of the object from these specific angles: 30 degrees to the right, 60 degrees to the right, 30 degrees to the left, and 60 degrees to the left.

---

Unfortunately all the models we evaluate perform poorly on this task. Here we will demonstrate our gold reference images.

**Reference Answer:**

1. 30 degrees to the right:

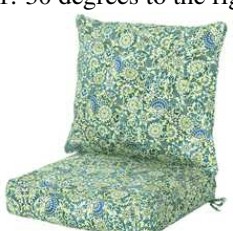

2. 60 degrees to the right:

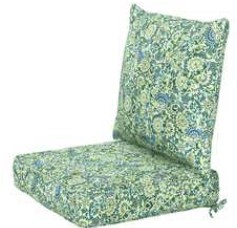

3. 30 degrees to the left:

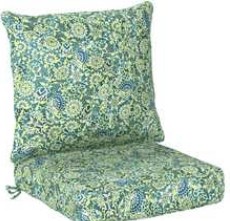

4. 60 degrees to the left:

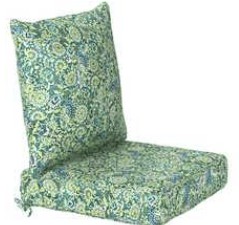

**Bad Example:** GEMINI IMAGE
**Response**: Generating an image of the floral patterned seat cushion rotated 30 degrees to the right, showing more of its right side and slightly obscuring the left.

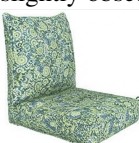

Generating an image of the floral patterned seat cushion rotated 60 degrees to the right, revealing much of its right side and significantly obscuring the left.

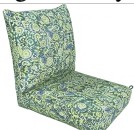

Generating an image of the floral patterned seat cushion rotated 30 degrees to the left, showing more of its left side and slightly obscuring the right.

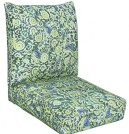

Generating an image of the floral patterned seat cushion rotated 60 degrees to the left, revealing much of its left side and significantly obscuring the right.

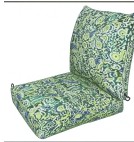

**Evaluation:** [0.184, 0.401, 0.321, 0.320] (image similarity)
**Analysis**: Fail to rotate the object correctly at the specific angles.

Figure 20: Examples for the task: Multi-Angle Consistency

---

**Multi-View Consistency**

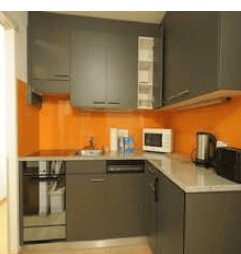

**Instruction**: Using the provided image as a reference for the frontal view, create four additional images from the following angles, in order: 15 degrees to the right, 60 degrees to the left, 30 degrees to the right, and 45 degrees to the left.

---

Unfortunately all the models we evaluate perform poorly on this task. Here we will demonstrate our gold reference images.

**Reference Answer:**

1. 15 degrees to the right:

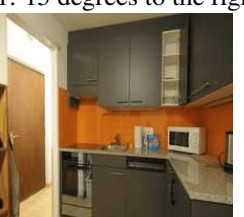

2. 60 degrees to the left:

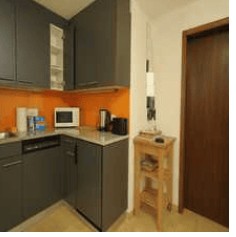

3. 30 degrees to the right:

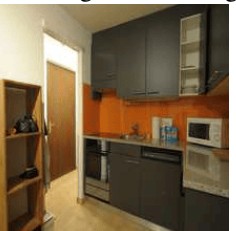

4. 45 degrees to the left:

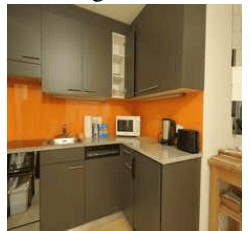

**Bad Example:** GEMINI 2.5 + GPT IMAGE

**Response**: Okay, here are the four images of the kitchen from the specified angles, using your provided image as the frontal view reference.

First, here is the kitchen viewed from 15 degrees to the right:

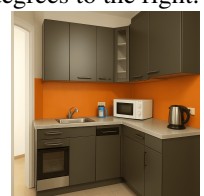

Next, the kitchen viewed from 60 degrees to the left:

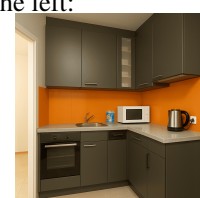

Then, the kitchen viewed from 30 degrees to the right:

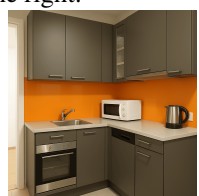

And finally, the kitchen viewed from 45 degrees to the left:

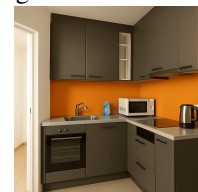

**Evaluation:** [0.302, 0.304, 0.299, 0.351] (image similarity)

**Analysis**: Fail to rotate the scene correctly at the specific angles

Figure 21: Examples for the task: Multi-View Consistency

## Composition Consistency

**Instruction**: Create three images that sequentially display the addition of an orange, a banana, and a kiwi into an initially empty fruit bowl. Ensure that previously added fruits remain in the bowl.

**Good Example:** GPT-4O + GPT IMAGE
**Response**: Here are the images showing the sequential addition of fruits into the bowl:
1. Empty Fruit Bowl with an Orange Added:

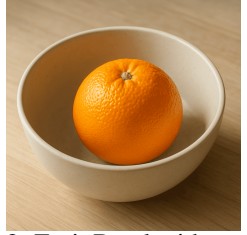

2. Fruit Bowl with an Orange and a Banana Added:

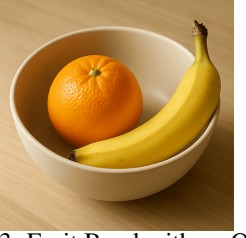

3. Fruit Bowl with an Orange, a Banana, and a Kiwi Added:

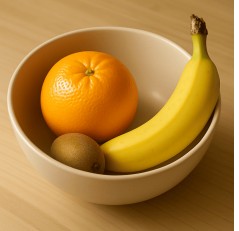

These images will illustrate the gradual addition of each fruit into the bowl.
**Evaluation:** ✔

**Bad Example:** GEMINI IMAGE
**Response**: Image 1: An empty, clear glass fruit bowl sits on a light wooden surface, ready to be filled.

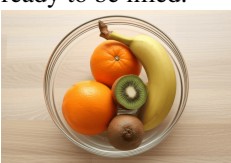

Image 2: A vibrant orange has been placed in the clear glass fruit bowl, resting slightly off-center. The light wooden surface is visible beneath and around the bowl.

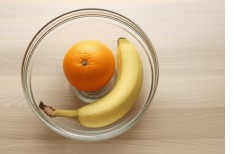

Image 3: A bright yellow banana has been added to the bowl, curving gently alongside the orange. Both fruits are now visible inside the clear glass bowl on the light wooden surface.

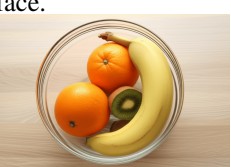

Image 4: A fuzzy brown kiwi has been placed in the bowl with the orange and banana. The three fruits now sit together in the clear glass bowl on the light wooden surface.

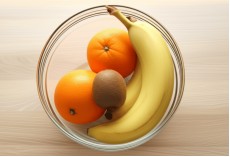

**Evaluation:** ✗
**Analysis**: Fail to follow the required order of adding objects and generate incorrect number of images.

Figure 22: Examples for the task: Composition Consistency

## Decomposition Consistency

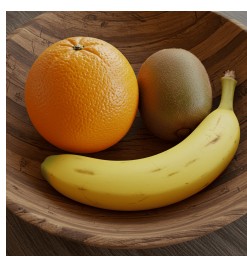

**Instruction**: Create three images that sequentially show the result after removing the kiwi, then the banana, and finally the orange from the original image. Do not replace the removed objects.

**Good Example:** GEMINI 2.5+GPT IMAGE
**Response**: Okay, here are the images showing the sequential removal of the fruits.
First, here is the image with the kiwi removed:

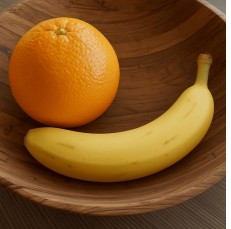

Next, here is the image with the banana also removed:

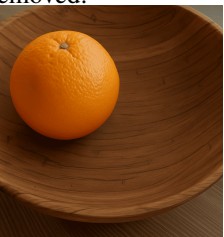

Finally, here is the image with the orange also removed, leaving an empty bowl:

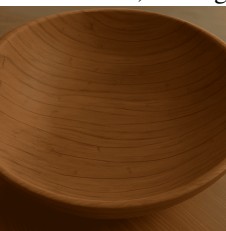

**Evaluation:** ✔

**Bad Example:** GEMINI IMAGE
**Response**: I will generate three images. The first image will show the wooden bowl with the orange and banana, after the kiwi has been removed.

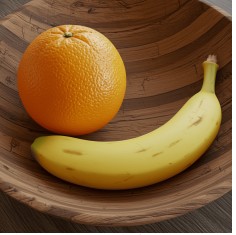

The second image will show the wooden bowl with only the orange, after the banana has been removed from the previous image.

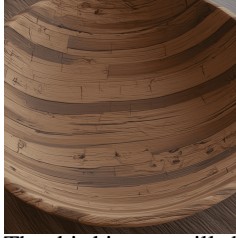

The third image will show the empty wooden bowl, after the orange has been removed from the previous image.

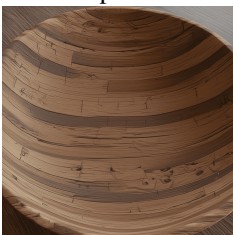

**Evaluation:** ✗
**Analysis**: Fail to follow the sequence of removing objects in the second image.

Figure 23: Examples for the task: Decomposition Consistency

Interleaved Image Editing - Object Adding

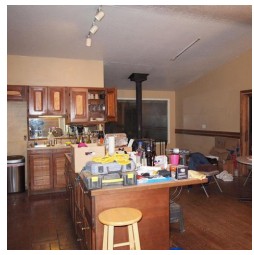 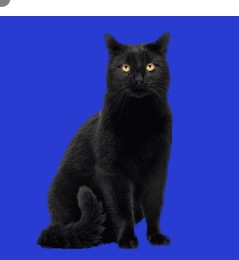

**Instruction**: Create an image displaying the result after placing the object <image_0> from <image_1>.

<image_0>        <image_1>

**Good Example:** GEMINI IMAGE

**Bad Example:** GEMINI 2.5 + GPT IMAGE

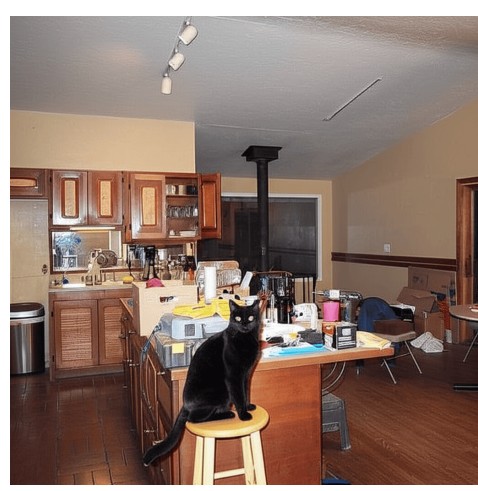 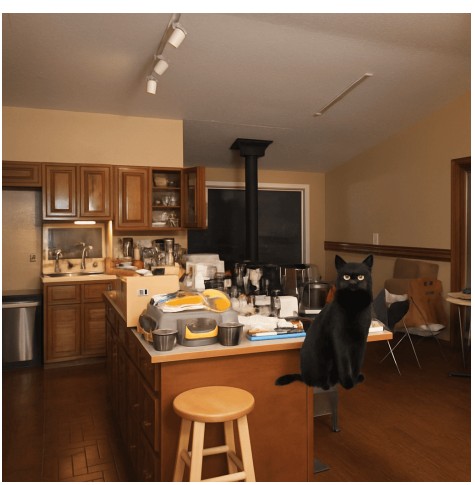

**Evaluation:** 0.770 (image similarity)

**Evaluation:** 0.342 (image similarity)
**Analysis**: The position of the cat is wrong and unreasonable. This is a common failure pattern among all the models.

Figure 24: Examples for the task: Interleaved Image Editing - Object Adding

**Interleaved Image Editing - Color Modifying**

**Instruction**: Generate an image that shows the result after changing the color of the largest ball in <image_0> to <image_1>.

<image_0>                <image_1>

**Good Example:** GPT-4O + GPT IMAGE

**Bad Example:** GEMINI 2.5 + GPT IMAGE

**Evaluation:** 0.722 (image similarity)

**Evaluation:** 0.483 (image similarity)
**Analysis**: The colors of background and some other objects are changed.

**Bad Example:** GEMINI IMAGE

**Evaluation:** 0.608 (image similarity)
**Analysis**: The colors and shapes of many objects are changed.

Figure 25: Examples for the task: Interleaved Image Editing - Color Modifying

Image Editing - Text Editing

**Instruction**: Create an image displaying the result after inserting the word "clean" onto the white linen closest to the viewer, while leaving the rest of the image untouched.

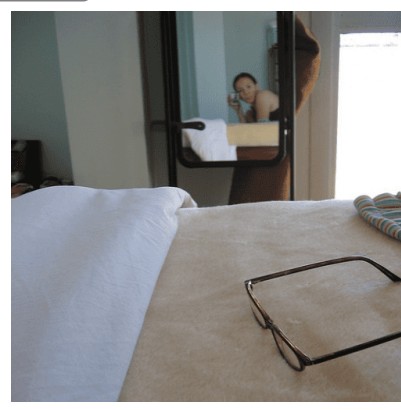

**Good Example:** GEMINI 2.5 + GPT IMAGE

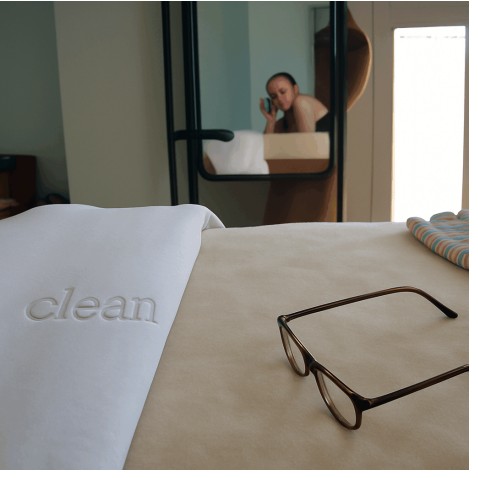

**Evaluation:** ✔, 0.786 (image similarity)

**Bad Example:** GEMINI IMAGE

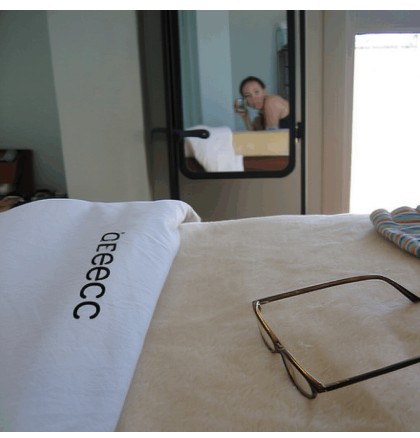

**Evaluation:** ✘
**Analysis**: The text rendered is wrong.
**Bad Example:** GEMINI 2.5 + IMAGEN 3

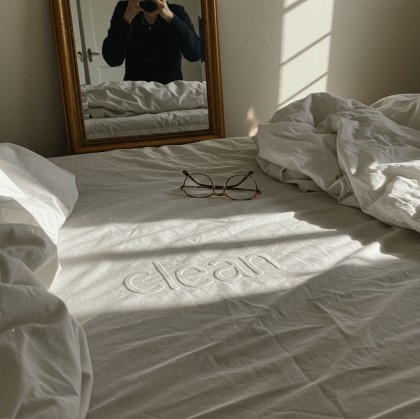

**Evaluation:** ✘
**Analysis**: The original image is changed. This is inevitable if using a text-to-image model incapable of image editing.

Figure 26: Examples for the task: Image Editing - Text Editing

Image Editing - Object Adding

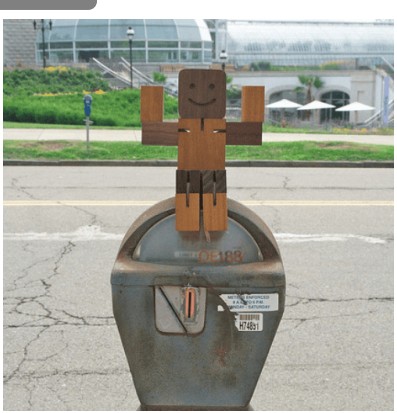

**Instruction**: Create an image displaying the result after placing a blue sticker on the meter, with all other elements in the image remaining unchanged.

**Good Example:** GEMINI IMAGE

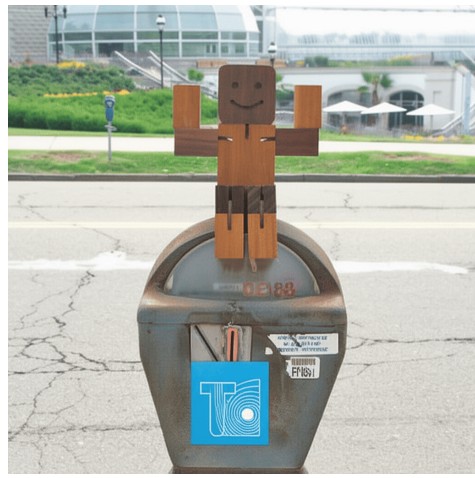

**Evaluation:** ✔, 0.911 (image similarity)

**Bad Example:** GPT-4O + GPT IMAGE

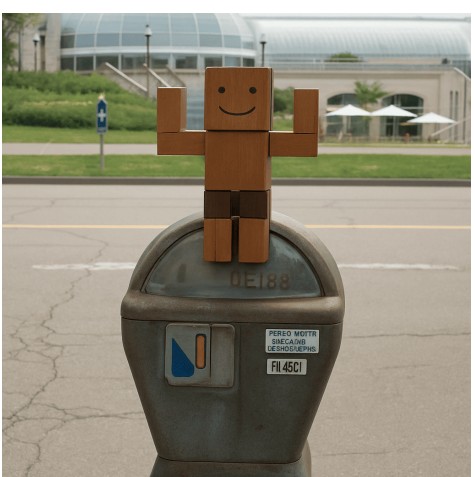

**Evaluation:** ✘
**Analysis**: No obviously visible blue sticker is added.

Figure 27: Examples for the task: Image Editing - Object Adding

Image Editing - Object Removing

**Instruction**: Create an image that displays the result after removing the man's wig, while leaving all other elements of the image unchanged.

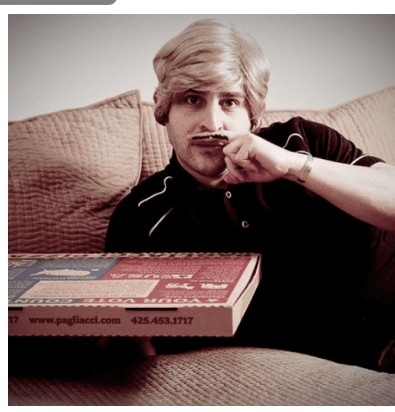

**Good Example:** GEMINI 2.5 + GPT IMAGE

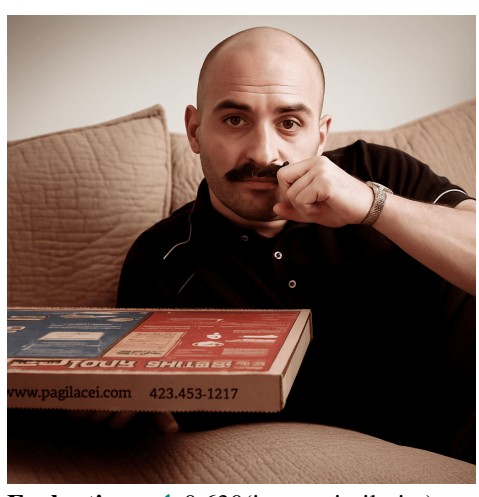

**Bad Example:** GEMINI IMAGE

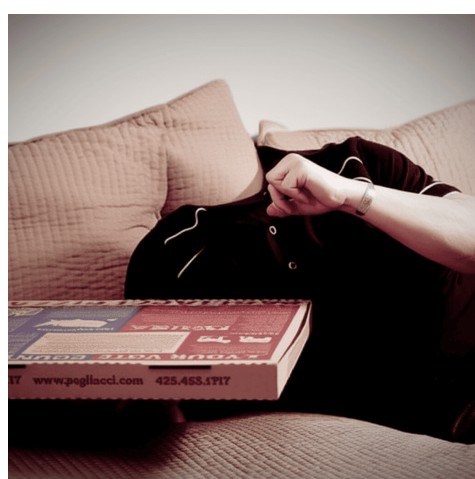

**Evaluation:** ✔, 0.630(image similarity)

**Evaluation:** ✘
**Analysis**: The wrong object (head) is removed.

Figure 28: Examples for the task: Image Editing - Object Removing

Image Editing - Object Modifying

**Instruction**: Generate an image displaying the result after replacing the nuts with blueberries, while keeping all other elements of the image unchanged.

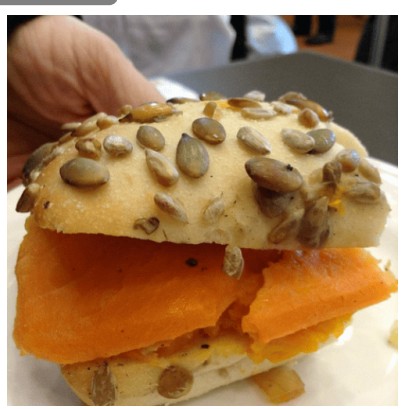

**Good Example:** GEMINI 2.5 + GPT IMAGE          **Bad Example:** GPT-4O + GPT IMAGE

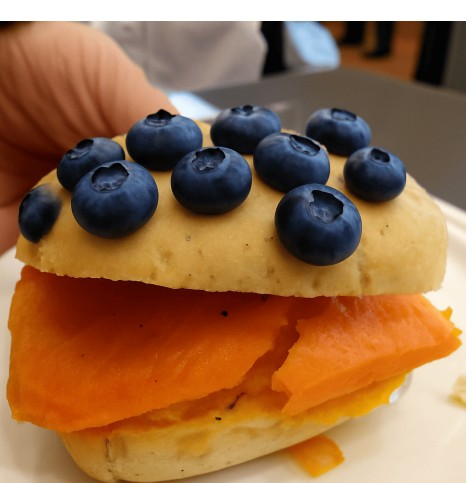
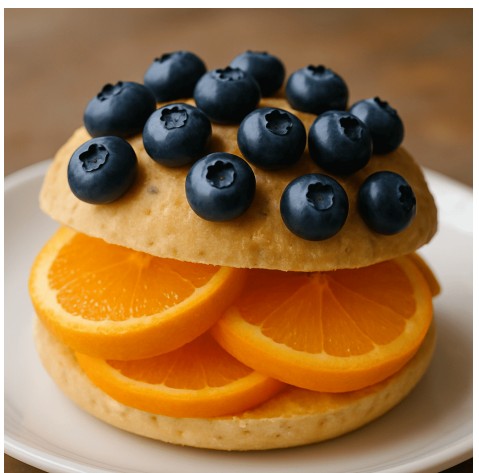

**Evaluation:** ✔, 0.853 (image similarity)

**Evaluation:** ✘
**Analysis**: Change elements (salmon to oranges) that should remain the unchanged.

Figure 29: Examples for the task: Image Editing - Object Modifying

Self Count

**Instruction**: Create an image of a kitchen counter displaying multiple apples—specifically, more than two but fewer than seven—that are clearly visible, not stacked, and not overlapping. The apples should be of standard size and typical colors. After generating the image, return ONLY the number of apples N in the following XML format: <count>N</count>.

**Good Example:** GEMINI 2.5 + GPT IMAGE
**Response**:

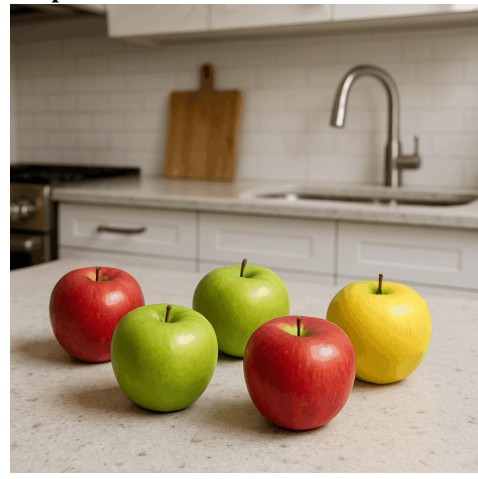

<count>5</count>
**Evaluation:** ✔

**Bad Example:** GEMINI 2.5 + IMAGEN 3
**Response**:

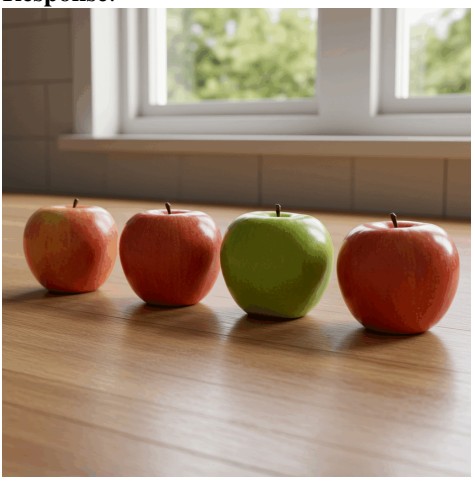

<count>5</count>
**Evaluation:** ✘
**Analysis**: The number of apples is four.

Figure 30: Examples for the task: Self Count

Self Color Recognition

**Instruction**: Create an image of a kitchen that includes a chair, a microwave, and a vase, ensuring that each object appears exactly once. Assign a unique color to each object, choosing from the following: blue, yellow, and red. Once the image is complete, return only a JSON object showing the color assigned to each item, formatted as: {"chair": "[color]", "microwave": "[color]", "vase": "[color]"}.

**Good Example:** GPT-4O + GPT IMAGE **Response**:

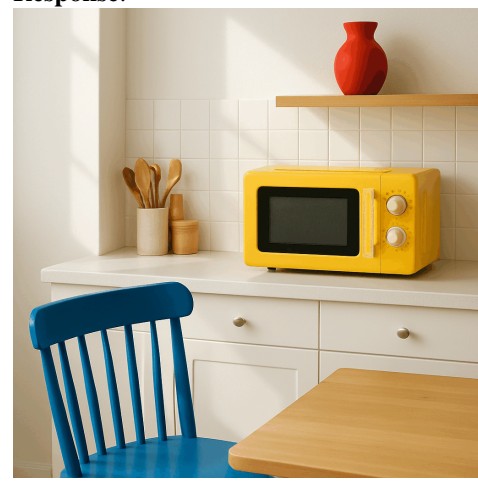

{"chair": "blue", "microwave": "yellow", "vase": "red"}
**Evaluation:** ✔

**Bad Example:** GEMINI IMAGE **Response**:

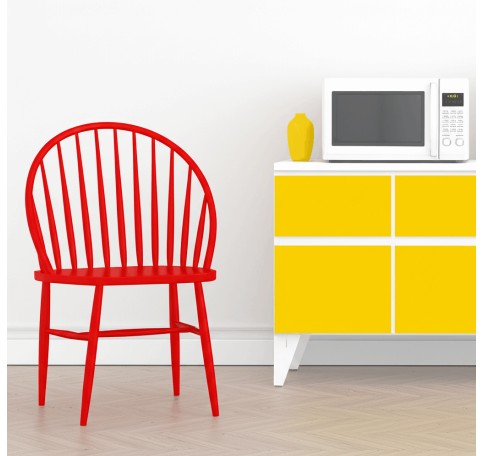

**Evaluation:** ✗
**Analysis**: No text response. Forgetting to output text is a common mistake for interleaved image-text generation.

Figure 31: Examples for the task: Self Color Recognition

---

Self Size Recognition

**Instruction**: Create an image of a living room that includes a carpet, a TV, and a poster, each with a different area and appearing only once in the scene. After generating the image, compare the areas of these three objects and provide ONLY a ranked list from smallest to largest area, such as ["poster", "TV", "carpet"].

- - - - - - - - - - - - - - - - - - - - - - - - - - - - - - - - - - - - - - - - - - - - -

**Good Example:** GPT-4O + GPT IMAGE
**Response**:

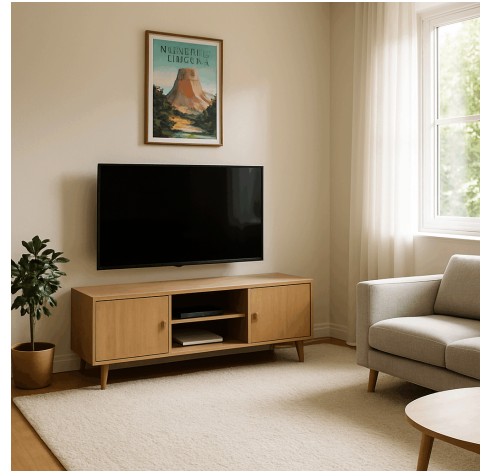

["poster", "TV", "carpet"]
**Evaluation:** ✔

**Bad Example:** GEMINI 2.5 + IMAGEN 3
**Response**:

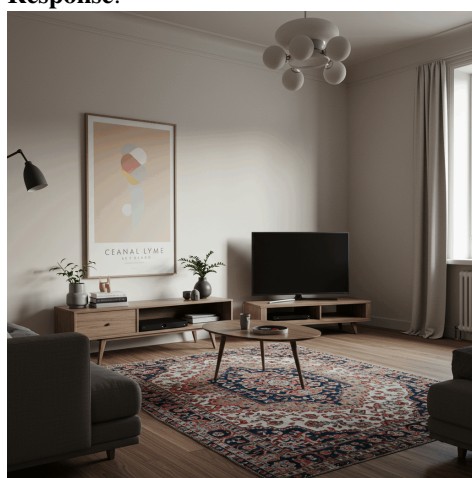

["poster", "TV", "carpet"]
**Evaluation:** ✘
**Analysis**: The poster is larger than then the TV

Figure 32: Examples for the task: Self Size Recognition

Self OCR

**Instruction**: Create an image of a whiteboard displaying a motivational quote. The quote must be between 5 and 10 words long, written in a clear, standard, and easily readable typeface without any artistic alterations. The quote should encourage people to follow their dreams. Once the image is generated, return ONLY the text from the whiteboard enclosed in XML format: <text>board text</text>.

**Good Example:** GEMINI 2.5+GPT IMAGE **Response**:

**Bad Example:** GEMINI IMAGE **Response**:

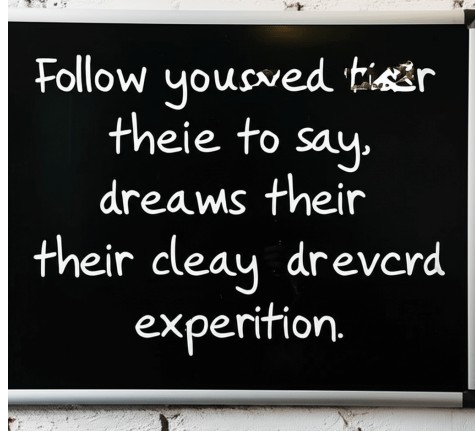

<text>chase your dreams, they know the way.</text>
**Evaluation:** ✔

<text>follow yousved theie to say, dreams their their cleay drevcrd experition.</text>
**Evaluation:** ✗
**Analysis**: The generated image has distorted text and does not align with model's response.

Figure 33: Examples for the task: Self OCR

---

**Self Relative Spatial Relation Recognition**

**Instruction**: Create an image showing a bathroom counter that includes a toothbrush, a bar of soap, and a tube of toothpaste. Each item should appear exactly once in the image. Once the image is complete, respond to these two questions: 1. The soap is to the ___ of the toothpaste tube: A. left B. right C. neither obviously left nor right. 2. The toothbrush is positioned ___ than the soap: A. higher B. lower C. neither obviously higher nor lower. Provide ONLY a list of letter answers (e.g., ["A", "B"]).

- - - - - - - - - - - - - - - - - - - - - - - - - - - - - - - - - - - - - - - - - - - - -

**Good Example:** GEMINI 2.5 + GPT IMAGE
**Response**:

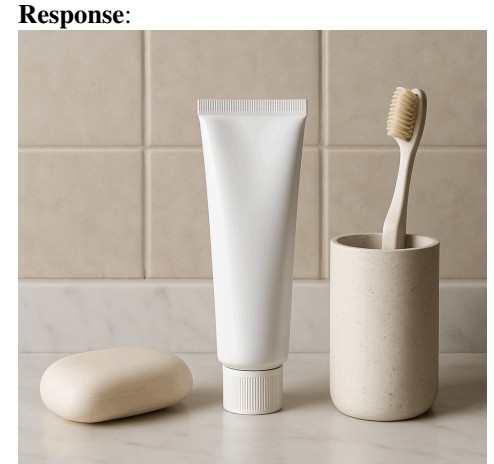

["A", "A"]
**Evaluation:** ✔

**Bad Example:** GEMINI IMAGE
**Response**:

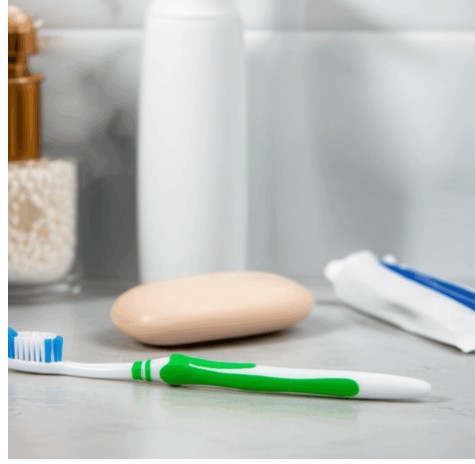

["A", "C"]
**Evaluation:** ✗
**Analysis**: The toothbrush is positioned lower than the soap.

Figure 34: Examples for the task: Self Relative Spatial Relation Recognition

---

**Self Absolute Spatial Relation Recognition**

**Instruction**: Create an image of a kitchen countertop featuring a blender, a bowl, and a cutting board, each positioned in a different quadrant of the image. Ensure that each object appears only once. Once the image is complete, answer the following two questions: 1. In which quadrant is the blender located? A. bottom left B. bottom right C. top left D. top right. 2. In which quadrant is the cutting board located? A. bottom left B. bottom right C. top left D. top right. Respond with ONLY a list of the corresponding letter choices (e.g., ["A", "D"])..

- - - - - - - - - - - - - - - - - - - - - - - - - - - - - - - - - - - - - - - -

**Good Example:** GEMINI 2.5 + IMAGEN 3
**Response**:

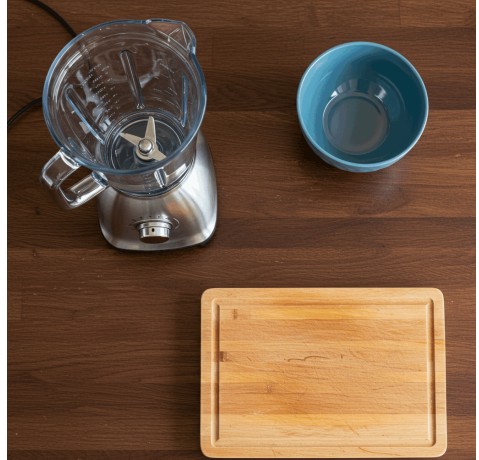

["C", "B"]
**Evaluation:** ✔

**Bad Example:** GEMINI 2.5 + GPT IMAGE
**Response**:

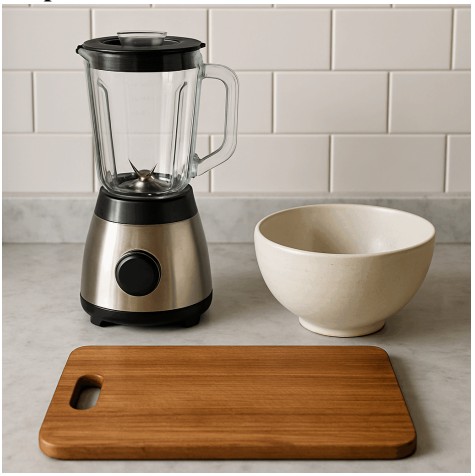

["C", "A"]
**Evaluation:** ✘
**Analysis**: The cutboard is at the bottom center of the image.
**Bad Example:** GPT-4O + GPT IMAGE
**Response**:

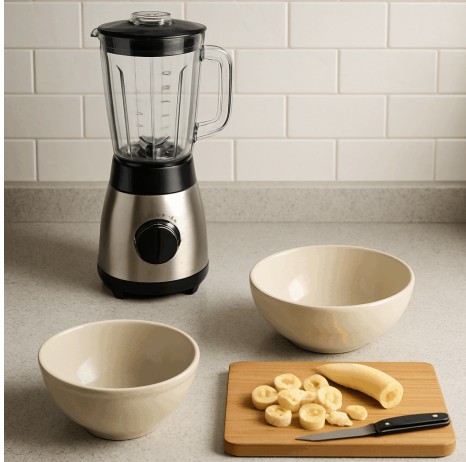

["C", "B"]
**Evaluation:** ✘
**Analysis**: There are multiple bowls. The wrong number of objects is a common mistake for all tasks.

Figure 35: Examples for the task: Self Absolute Spatial Relation Recognition

## Interleaved Math Reasoning

**Instruction**: Carefully examine the sequence and determine the most suitable pattern to replace the question mark. Analyze the sequence step by step and clearly explain your reasoning. Present your final answer enclosed in double angle brackets as follows: «a detailed text description of the pattern ONLY». Then, create an image representing the same pattern to fill in the question mark.

Unfortunately all the models we evaluate perform poorly on this task. Here we will demonstrate our gold reference images.
**Reference Answer:**

**Evaluation:** ✔

**Bad Example:** GEMINI 2.5 + GPT IMAGE **Response:**
To determine the pattern that replaces the question mark, let's analyze the positions of the white circle and the black square in each row of the sequence. We'll number the squares from 1 (leftmost) to 6 (rightmost).
Step 1: Analyze the position of the white circle.
Row 1: The white circle is in the 3rd square.
Row 2: The white circle is in the 4th square. (Movement: 3 → 4, which is +1)
Row 3: The white circle is in the 2nd square. (Movement: 4 → 2, which is -2)
. . .
A white circle in the 1st square.
A black square in the 6th square.
The squares in between (2nd, 3rd, 4th, and 5th) will be empty.
«A row of six squares with a white circle in the first square from the left and a black square in the sixth square from the left. The second, third, fourth, and fifth squares are empty. »

**Evaluation:** ✗
**Analysis**: All the moving patterns are correct, but the last step of calculating the position of the black square goes wrong.

Figure 36: Examples for the task: Interleaved Math Reasoning

Interleaved Code Reasoning

**Instruction**: [SVG Code] What does this SVG code represent? Analyze the elements step by step, then create a rendered image showing how it would appear in a browser. (reference image is not given)

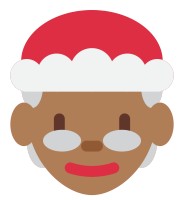

reference image

**Good Example:** GEMINI 2.5+GPT IMAGE **Response**:

This SVG code describes a cartoon image, likely an emoji or a simple illustration. Let's break down its components:

1.  **`<path fill="#AF7E57" d="M6 18.562..."/>`**: This is the main shape of the face, filled with a tan color (`#AF7E57`). It's a large, somewhat pear-shaped element forming the head.

2.  **`<path fill="#DD2E44" d="M30.961 10c..."/>`**: This path, filled with red (`#DD2E44`), forms the main part of a hat sitting on top of the head.

...

12.  **`<path fill="#D1D3D4" d="M10 25c0.../>` and `<path fill="#D1D3D4" d="M26.003 25c0..."/>`**: Two light gray shapes at the very bottom, possibly representing parts of a beard, chin, or stylized feet if it were a full figure. Given the context, they are likely part of the lower face or beard detail.

Collectively, these elements form an image of a **Santa Claus face emoji**. Here is a rendered image of how it would appear:

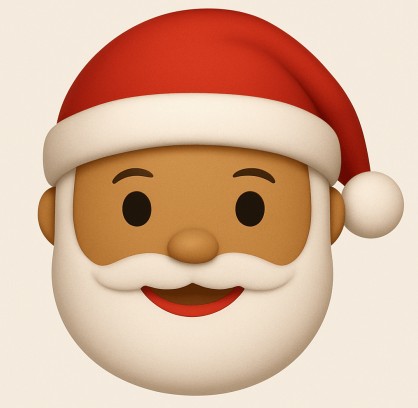

**Evaluation:** ✔, 0.781 (similarity score)

**Bad Example:** GPT-4O + GPT IMAGE **Response**:

The SVG code represents a stylized cartoon face, possibly a bear or a similar animal, with various decorative elements. Let's break down the elements step by step:

1.  **Main Face Shape**: The main face is created using a path with a fill color of `#AF7E57`, which is a brownish color. This path forms the central part of the face.

2.  **Ears**: Two ellipses with a fill color of `#D1D3D4` are used to create the ears. These are positioned at the top left and right of the face.

...

8.  **Mouth**: A path with a fill color of `#915A34` is used to create the mouth, adding detail to the expression.

9.  **Background Elements**: The SVG includes additional paths and ellipses to add depth and detail to the overall design.

Now, let's create a rendered image showing how this SVG would appear in a browser:

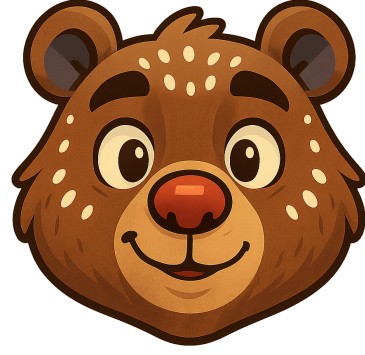

**Evaluation:** ✗
**Analysis**: Wrong final image.

Figure 37: Examples for the task: Interleaved Code Reasoning

## Image-Text Order

**Instruction**: Use visual storytelling to depict the four seasons. For each season, begin by creating an image of a landscape that represents that time of year, followed by a written description in natural language highlighting its key features. Provide only the requested content—do not include explanations, titles, or extra commentary.

**Good Example:** GEMINI 2.5+GPT IMAGE
**Response**: This spring scene showcases fresh green grass, trees bursting with new leaves and pink blossoms, and a variety of colorful flowers under a bright, clear sky.

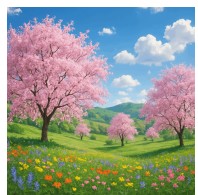

Summer is represented by a bright, sunny day with deep blue skies, lush green foliage at its peak, and often, warm, inviting waters.

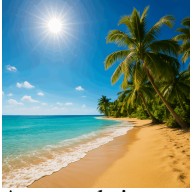

Autumn brings a palette of warm colors as leaves turn red, orange, and yellow; the air becomes crisp, and a sense of calm pervades the landscape.

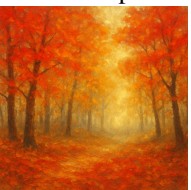

Winter is characterized by a blanket of snow covering the ground and trees, often with frozen bodies of water and a quiet, still atmosphere.

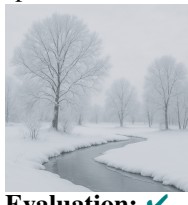

**Evaluation:** ✔

**Bad Example:** GEMINI IMAGE
**Response**: A vibrant green field stretches under a clear blue sky, bordered by a line of lush, dark green trees in the distance. The grass shows subtle stripes from recent mowing, indicating care and growth.

A stark white landscape is dominated by a snow-covered path leading into the distance, flanked by wooden posts connected by dark railings. The sky is a pale, overcast white, blending seamlessly with the snow-covered ground.

Patches of snow dust a field of dry, golden-brown grass, contrasting with the remaining green leaves on the distant trees under a bright blue sky with wispy clouds. A large tree branch with vibrant orange leaves extends into the frame from the upper left.

A winding path covered in fresh snow cuts through a forest where trees are heavily laden with snow, creating a monochrome scene in shades of white and dark gray. A wooden fence runs along the side of the path, also covered in snow.

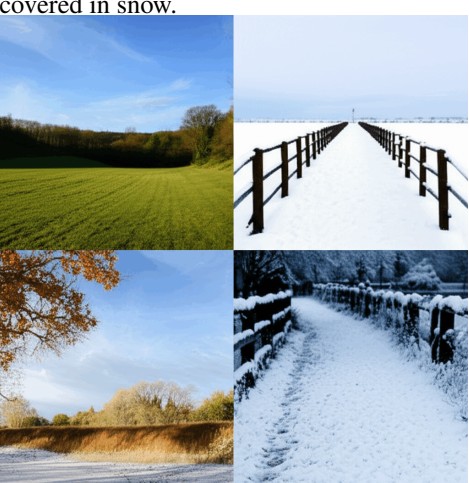

**Evaluation:** ✗
**Analysis**: Generate only one image.

Figure 38: Examples for the task: Image-Text Order

**Sound Begin-End**

**Instruction**: Create an audio that begins with the sound of chirping birds and concludes with the sound of a mouse click.

**Good Example:** MAKE-AN-AUDIO 2
**Audio Description:** an audio with the beginning 3 seconds of chirping birds and the last 1 sec of mouse click.
**Evaluation:** ✔

**Bad Example:** STABLE AUDIO
**Audio Description:** an audio with chirping birds all the time, and a mouth click only in the 2nd second, not the end.
**Evaluation:** ✘

Figure 39: Examples for the task: Sound Begin-End

**Sound Positional Inclusion**

**Instruction**: Create an audio of a city street, ensuring a police car siren is included in the first half.

**Good Example:** AUDIOGEN
**Audio Description:** an audio of a city street with the first 3 seconds including a police car siren. **Evaluation:** ✔

**Bad Example:** STABLE AUDIO
**Audio Description:** an audio of police car siren mixed with normal cars passing by street all the time.
**Evaluation:** ✘

Figure 40: Examples for the task: Sound Positional Inclusion

**Sound Reasoning**

**Instruction**: Produce the sound of a black bird commonly linked to death or misfortune.

**Good Example:** MAKE-AN-AUDIO 2
**Audio Description:** an audio of very typical crow cry.
**Evaluation:** ✔

**Bad Example:** STABLE AUDIO
**Audio Description:** bright and melodious chirp of a certain kind of bird that's obviously not crow.
**Evaluation:** ✘

Figure 41: Examples for the task: Sound Reasoning

**Sound Silence**

**Instruction**: Create an audio that begins with a loud car horn, followed by a long silence, and concludes with a distant siren.

**Good Example:** MakeAnAudio2
**Audio Description:** an audio with a loud car horn in the beginning 3 seconds, and then comes 4 seconds of silence, with the last 3 seconds of a distant siren.
**Evaluation:** ✔

**Bad Example:** AUDIOLDM 2
**Audio Description:** an audio without any silent time, and the two sounds are mixed together.
**Evaluation:** ✘

Figure 42: Examples for the task: Sound Silence

**Music Instrument Inclusion**

**Instruction**: Create a seamless saxophone improvisation.

**Good Example:** TANGO MUSIC
**Audio Description:** a casual piece of saxophone improvisation.
**Evaluation:** ✔

**Bad Example:** YUE
**Audio Description:** an audio starting with 3 seconds of laughter and then 5 seconds of improvised jazz music including piano, drums and saxophone.
**Evaluation:** ✘

Figure 43: Examples for the task: Music Instrument Inclusion

**Music Instrument Exclusion**

**Instruction**: Create an audio of a city street, ensuring a police car siren is included in the first half.

**Good Example:** MUSICGEN
**Audio Description:** an audio of smooth jazz music featuring bass but without drums.
**Evaluation:** ✔

**Bad Example:** STABLE AUDIO
**Audio Description:** an audio of jazz music with rhythms played by drums.
**Evaluation:** ✘

Figure 44: Examples for the task: Music Instrument Exclusion

**Music Intensity**

**Instruction**: Compose a cinematic orchestral piece that gradually fades out at the end.

**Good Example:** STABLE AUDIO
**Audio Description:** an audio of orchestral piece featuring a cinematic build with rich instrumentation and gradually fading out, creating a smooth ending.
**Evaluation:** ✔

**Bad Example:** TANGO MUSIC
**Audio Description:** an audio quite the opposite, with a tranquil start and getting more intense.
**Evaluation:** ✘

Figure 45: Examples for the task: Music Intensity

**Music Tempo**

**Instruction**: Create a laid-back lo-fi hip-hop beat at 100 BPM.

**Good Example:** MusicGen
**Audio Description:** an audio of a hip-hop beat at approximately 102 BPM.
**Evaluation:** ✔

**Bad Example:** AUDIOLDM2
**Audio Description:** an audio of a hip-hop beat at approximately 64 BPM.
**Evaluation:** ✘

Figure 46: Examples for the task: Music Tempo

---

Speaker Voice Attribution

**Instruction**: Generate an audio of a man speaking rapidly in a low-pitched voice, saying, "The detective carefully examined the crime scene, noting every detail that could lead him to the truth, knowing that even the smallest clue might be the key to solving the mystery."

- - - - - - - - - - - - - - - - - - - - - - - - - - - - - - - - -

**Good Example:** GEMINI 2.5 + VOXIN-STRUCT
**Speech Transcript:** (low-pitched male voice talking rapidly) The detective carefully examined the crime scene, noting every detail that could lead him to the truth, knowing that even the smallest clue might be the key to solving the mystery.
**Evaluation:** ✔

**Bad Example:** GEMINI 2.5 + VOICELDM
**Speech Transcript:** (high-pitched male voice talking rapidly) The detective carefully examined the crime scene, noting every detail that could lead him to the truth, knowing that even the smallest clue might be the key to solving the mystery.
**Evaluation:** ✘

Figure 47: Examples for the task: Speaker Voice Attribution

---

Multi-Lingual Speech

**Instruction**: Generate an audio of a man slowly speaking: "窗外的雨滴敲打着玻璃，滴滴答答的声音仿佛一首温柔的旋律，让她的思绪飘回了那个久远而温暖的夏天。"

- - - - - - - - - - - - - - - - - - - - - - - - - - - - - - - - -

**Good Example:** GEMINI 2.5 + VOXIN-STRUCT
**Speech Transcript:** (an audio of a man gently and slowly speaking Chinese) 窗外的雨滴敲打着玻璃，滴滴答答的声音仿佛一首温柔的旋律，让她的思绪飘回了那个久远而温暖的夏天。
**Evaluation:** ✔

**Bad Example:** GEMINI 2.5 + VOXIN-STRUCT
**Speech Transcript:** (an audio of a man speaking Chinese hastily) 窗外的雨滴敲打着玻璃，仿佛一首温柔的旋律，让她的思绪飘回了那个久远而温暖的夏天。
**Evaluation:** ✘

Figure 48: Examples for the task: Multi-Lingual Speech

---

Speaker Voice Replication

**Instruction**: Create an audio of reading the sentence, "The aroma of fresh coffee and warm pastries filled the air as she stepped into the café, instantly feeling a sense of comfort and familiarity in the cozy atmosphere," using the same voice as the reference speaker. (reference speech: a gentle male sound)

- - - - - - - - - - - - - - - - - - - - - - - - - - - - - - - - -

**Good Example:** GEMINI 2.5 + VOICELDM
**Speech Transcript:** (a warm, middle-pitched gentle male sound) The aroma of fresh coffee and warm pastries filled the air as she stepped into the café, instantly feeling a sense of comfort and familiarity in the cozy atmosphere.
**Evaluation:** ✔

**Bad Example:** GEMINI 2.5 + VOXIN-STRUCT
**Speech Transcript:** (a hoarse male sound) The aroma of fresh coffee and warm pastries filled the air as she stepped into the café.
**Evaluation:** ✘

Figure 49: Examples for the task: Speaker Voice Replication

---

**Transcript Generation**

**Instruction**: Create a speech on the internet revolution, making sure the word "digital" is used at least twice.

- - - - - - - - - - - - - - - - - - - - - - - - - - - - - - - - - - - - - - - - - - - -

**Good Example:** GEMINI 2.5 + VOXIN-STRUCT
**Speech Transcript:** As you just heard, the internet revolution has profoundly impacted our lives, marking a significant shift into a new digital era. This ongoing digital transformation continues to unfold, bringing new opportunities and challenges.
**Evaluation:** ✔

**Bad Example:** GEMINI 2.5 + VOICELDM
**Speech Transcript:** Friends, colleagues, we stand at the precipice of an era by connectivity. The internet revolution has reshaped our world, ushering in a new digital age with transformation connecting billions and democratizing information, and its story is still being written.
**Evaluation:** ✘
**Analysis**: keyword "digital" appears only once.

Figure 50: Examples for the task: Transcript Generation

---

**Transcript Editing**

**Instruction**: Revise the provided speech by replacing the third sentence with: "Join hands, break barriers, and transform the world." (Provided speech transcript: "We must act now. Every action counts. We can make a difference together if we remain united. Progress comes from shared responsibility.")

- - - - - - - - - - - - - - - - - - - - - - - - - - - - - - - - - - - - - - - - - - - -

**Good Example:** GEMINI 2.5 + VOICELDM
**Speech Transcript:** We must act now. Every action counts. We can make a difference together if we remain united. Join hands, break barriers, and transform the world.
**Evaluation:** ✔

**Bad Example:** GEMINI 2.5 + VOXIN-STRUCT
**Speech Transcript:** We must act now. Every action counts. Join hands, break barriers, and transform the world.
**Evaluation:** ✘
**Analysis**: The third sentence is missing.

Figure 51: Examples for the task: Transcript Editing



**Speech Conversation**

**Instruction**: Create five consecutive speeches forming a dialogue between James and Victoria about architectural innovations for sustainable urban living, with each speech delivered by a single speaker. Use the following speaking order: James, Victoria, Victoria, James, James. Constraints: 1. Victoria's second speech must begin with the phrase "Contrary to your point." 2. James's final speech must contain exactly two uses of the word "design".
Ensure each speaker maintains a consistent voice throughout the conversation.

- - - - - - - - - - - - - - - - - - - - - - - - - - - - - - -

**Good Example:** GEMINI 2.5 + VOICELDM
**Conversation Transcript:**
James: I've been fascinated by the potential of kinetic pavements in high-traffic urban areas to generate clean energy.
Victoria: That's an interesting concept, James! I also think smart water management systems like rainwater harvesting and grey-water recycling are vital.
Victoria: Contrary to your point about focusing on generation, I believe water conservation innovations offer more immediate and widespread benefits for sustainability.
James: Both are crucial, Victoria. Perhaps integrating biomimetic architecture that learns from nature's efficiency could address energy and water simultaneously.
James: The overall urban design must be rethought. We need a cohesive design approach that incorporates all these innovations seamlessly for truly sustainable cities.
**Evaluation:** ✔

**Bad Example:** GEMINI 2.5 + VOXIN-STRUCT
**Conversation Transcript:**
James: I believe modular construction is key to sustainable urban development, significantly reducing waste and build times.
Victoria: That's a valid point, James. Integrating green roofs and vertical gardens can also drastically improve air quality and biodiversity.
Victoria: Contrary to your point about modular construction being the key, I think retrofitting existing buildings with smart technologies offers a more immediate impact on energy consumption.
James: Retrofitting is important, Victoria, but new constructions set the standard. We need innovations in material science too, like self-healing concrete.
James: Ultimately, a holistic approach to urban design is crucial.
**Evaluation:** ✗
**Analysis**: James's final speech only contain one use of the word "design"



Figure 52: Examples for the task: Speech Conversation

---

**Audio-Text Order**

**Instruction**: Develop a 2-step guide for delivering a persuasive presentation. For each step, begin with detailed written instructions in natural language, followed by a speech from a communication expert demonstrating the technique. For the second step, also include an additional speech at the end that illustrates how the technique is applied in a formal business context. Generate only what is requested—no explanations, titles, or extra commentary.

- - - - - - - - - - - - - - - - - - - - - - - - - - - - - - - - - - - -

**Good Example:** GEMINI 2.5 + VOXIN-STRUCT
**Response:** Step 1: Thoroughly research your audience to understand their needs, motivations, existing knowledge, and potential objections. Adapt your language, examples, and level of detail to resonate specifically with them. Frame your core message in a way that directly addresses their interests and demonstrates clear value from their perspective, ensuring your arguments are supported by credible evidence and logical reasoning.
(speech) *To truly persuade, you must first connect—understand who you're speaking to, what their concerns are, their aspirations. Tailor your message to their world, not yours. Make it relevant, make it resonate.*
Step 2: Structure your presentation for maximum impact by creating a clear, logical, and compelling narrative. Begin with a strong opening that grabs attention and establishes relevance. Develop your core arguments with supporting evidence, data, and illustrative examples, addressing potential counter-arguments proactively. Conclude with a memorable summary and a clear call to action, guiding your audience on the next steps.
(speech)*Your delivery is as crucial as your content. Project confidence through your voice and body language. And when faced with questions or doubts, listen carefully, acknowledge their points, and respond thoughtfully. This builds trust.*
(speech) *While I understand concerns about the initial investment, consider the long-term ROI. Our projections show a 20% return within two years, significantly boosting our profit margins and market competitiveness. This strategic move positions us for success.*
**Evaluation:** ✔

**Bad Example:** GEMINI 2.5 + VOICELDM
**Response:**
Step 1: Thoroughly understand your audience and craft a compelling narrative. This involves researching their demographics, interests, pre-existing beliefs, and potential pain points. Use this understanding to build a story that resonates, typically following a structure that introduces a problem, presents your proposed solution, highlights the benefits, and concludes with a clear call to action. Employ vivid language, relatable examples, and appropriate emotional appeals to make your message memorable and impactful.
Step 2: Master your delivery and prepare to handle objections gracefully. Effective delivery encompasses strong body language—maintaining eye contact, adopting an open posture, and using purposeful gestures—as well as vocal variety, modulating your tone, pace, and volume to maintain engagement.
(speech) *To truly persuade, you must first connect—understand who you're speaking to, what their concerns are, their aspirations. Tailor your message to their world, not yours. Make it relevant, make it resonate.*
(speech) *A persuasive presentation flows like a good story. Hook your audience early, build your case with compelling evidence, and then guide them to action.*
(speech) *Good morning. Our analysis indicates a 15% market share increase is achievable by Q4 if we implement the proposed strategy. This directly addresses the growth targets set last quarter and positions us ahead of key competitors. We seek your approval to proceed.*

**Evaluation:** ✗
**Analysis**: Speeches and texts are not interleaved in the expected order.

Figure 53: Examples for the task: Audio-Text Order

