# OpenReview forum: "MMMG: A Comprehensive and Reliable Benchmark for Multitask Multimodal Generation"
_ICLR.cc/2026/Conference — Submitted to ICLR 2026_

### Official Review · Reviewer_R7P8 · 2025-10-30

**Soundness:** 3
**Presentation:** 2
**Contribution:** 2
**Rating:** 4
**Confidence:** 4

**Summary:**

This paper introduces MMMG, a comprehensive benchmark for evaluating multitask multimodal generation, designed to address the misalignment between automated metrics and human evaluation, as well as the limited coverage of interleaved modalities in existing benchmarks. MMMG spans 4 modality combinations (image, audio, interleaved text-image, interleaved text-audio) and includes 55 tasks (31 newly developed) with 1248 structured instructions, targeting key capabilities like reasoning, controllability, and cross-modal coherence.

**Strengths:**

1. The benchmark exhibits high quality through rigorous data curation (only 10% of generated instructions pass strict filtering), comprehensive evaluation validation (high agreement among multiple annotators, with an average inter-annotator agreement of 97.1%), and strong reproducibility (detailed documentation of data sources, model checkpoints, and evaluation protocols).

2. The paper is well-structured, with intuitive presentations of task definitions (e.g., detailed metadata in Table 1), experimental design (e.g., evaluation pipelines in Section 3.2), and failure analyses (e.g., model weaknesses in Figure 3). This clarity makes it easy for readers to understand the scope of tasks and interpret benchmark results.

3. MMMG establishes a reliable evaluation baseline for multimodal generation, guides future research by identifying critical gaps (e.g., limited performance in audio generation), and enables fine-grained analysis of model weaknesses (e.g., poor multimodal reasoning in top models). Collectively, these contributions advance the field of multimodal generation.

**Weaknesses:**

1. Although MMMG includes 55 tasks (31 newly developed), Appendix B.1 (Data Source) reveals that most task data relies on simple sampling or instruction modification from existing datasets (e.g., HotpotQA for object reasoning, EmuEdit for image editing, LibriSpeech for speech replication). This heavy dependence on pre-existing data undermines the novelty of the "newly developed" tasks, as they do not involve fully independent or original data curation.

2. The paper provides insufficient details about the data sources and construction processes for the 31 newly developed tasks. Unlike the detailed descriptions of data derived from existing datasets (e.g., COCO for object images, CLEVR for color modification), there is no clear documentation of how the newly designed tasks were sourced, how their instructions were synthesized, or how reference outputs (if any) were created—hindering reproducibility and assessment of task validity.

3. The analysis of multimodal reasoning failures is inadequate. Benchmark results show top models perform poorly on such tasks (e.g., 10.1% accuracy for math, 31.8% for 3D code transformations), but the paper fails to: (1) Diagnose root causes—whether failures originate from model architecture (unified autoregressive vs. agent-based), limited training data for math/code multimodal scenarios, or overly strict VLM evaluation prompts; (2) Propose mitigations—without identifying the source of weaknesses, MMMG offers limited guidance for researchers to prioritize improvements.

4. The claim that top models "fall short on multimodal reasoning" lacks sufficient rigor. For math interleaved reasoning tasks, the paper adapts data from MM-IQ (Appendix B.2) by converting original multiple-choice questions into free-form generation tasks. This modification not only increases task difficulty but also introduces ambiguity: it is impossible to disentangle whether poor model performance stems from genuine weaknesses in multimodal reasoning or limitations in image generation (e.g., failing to render clear visual elements for math problems).

**Questions:**

1. Regarding the rigor of the "models fall short on multimodal reasoning" claim: Given that math interleaved reasoning tasks were adapted from MM-IQ by converting multiple-choice questions to free-form (Appendix B.2), which may confound reasoning weaknesses with image generation limitations, do you have additional data or controlled experiments (e.g., testing models on the original multiple-choice format vs. free-form, or evaluating image generation quality independently for math tasks) to disentangle whether poor performance stems from multimodal reasoning deficits or inadequate visual rendering?

2. For the 31 newly developed tasks: Since Appendix B.1 details data sourcing for tasks derived from existing datasets (e.g., HotpotQA, EmuEdit) but lacks clarity on the 31 new tasks, could you supplement details on their specific data sources, instruction synthesis processes (e.g., how prompts were designed beyond GPT-4O generation), and creation of reference outputs (if applicable)? This would help assess the novelty and validity of these "new" tasks compared to modified existing ones.

3. On the insufficient analysis of multimodal reasoning failures: Given top models’ low accuracy on math (10.1%) and 3D code transformation (31.8%) tasks, have you conducted follow-up analyses to identify root causes—such as comparing performance across model architectures (unified autoregressive vs. agent-based), testing with augmented math/code multimodal training data, or adjusting VLM evaluation prompts to rule out strictness bias? Additionally, do you have preliminary insights or suggestions for mitigating these failures?

---

> ### Author Response · Authors · 2025-11-22
> **Rebuttal by Authors (1/7)**
>
> We would like to thank the reviewer for your thoughtful comments and feedback.
>
> > Although MMMG includes 55 tasks (31 newly developed), Appendix B.1 (Data Source) reveals that most task data relies on simple sampling or instruction modification from existing datasets (e.g., HotpotQA for object reasoning, EmuEdit for image editing, LibriSpeech for speech replication). This heavy dependence on pre-existing data undermines the novelty of the "newly developed" tasks, as they do not involve fully independent or original data curation.
>
> We would like to clarify that our main contribution and novelty lies in the task and evaluation design, not in collecting entirely new pixels or waveforms. Popular benchmarks such as MathVista [1] and BLINK [2] are more heavily reliable to tasks from previous corpora, and they made very important contributions to the field. For MMMG, its curation process requires many hard work:
>
> First, **selecting or creating verifiable subtasks itself is non-trivial**. Not all tasks or instructions from existing corpora can be reliably evaluated. Take image editing tasks for example. Many original instructions in EmuEdit have ambiguous editing regions. We rewrite them to specify the edited region and target content unambiguously, so that both models and human evaluators know exactly what change is required. This selection and rewriting is time-consuming but necessary to guarantee that automatic metrics match human judgments.
>
> Also, **Adapt instruction for multimodal generation setup requires substantial work.** As one concrete example, although we select questions from HotpotQA, the original answers are mostly dates or locations, which cannot be directly turned into image generation instructions. We therefore carefully rewrite these questions into generative prompts such as: “the national flag of [country]”and “the zodiac sign of [date]” , so that the “answer” can be realized as an image while remaining automatically checkable.
>
> We believe all these are important for future research in the multimodal world. As RL with verifiable rewards become increasingly important for the text models, the multimodal models will also advance a lot in this direction, which is exactly what MMMG designed for.
>
> [1] Lu P, Bansal H, Xia T, et al. Mathvista: Evaluating mathematical reasoning of foundation models in visual contexts[J]. arXiv preprint arXiv:2310.02255, 2023.
>
> [2] Fu X, Hu Y, Li B, et al. Blink: Multimodal large language models can see but not perceive[C]//European Conference on Computer Vision. Cham: Springer Nature Switzerland, 2024: 148-166.

---

> ### Author Response · Authors · 2025-11-22
> **Rebuttal by Authors (2/7)**
>
> > The paper provides insufficient details about the data sources and construction processes for the 31 newly developed tasks. Unlike the detailed descriptions of data derived from existing datasets (e.g., COCO for object images, CLEVR for color modification), there is no clear documentation of how the newly designed tasks were sourced, how their instructions were synthesized, or how reference outputs (if any) were created—hindering reproducibility and assessment of task validity. For the 31 newly developed tasks: Since Appendix B.1 details data sourcing for tasks derived from existing datasets (e.g., HotpotQA, EmuEdit) but lacks clarity on the 31 new tasks, could you supplement details on their specific data sources, instruction synthesis processes (e.g., how prompts were designed beyond GPT-4O generation), and creation of reference outputs (if applicable)? This would help assess the novelty and validity of these "new" tasks compared to modified existing ones.
>
> Thank you for pointing out that the construction process for the new tasks was under-specified. We have clarified our instruction-generation pipeline in Section 3.1 and Appendix B.1, and we will further expand these sections in the revision. For the 31 newly developed tasks, data is created via a shared four-step pipeline:
>
> 1. Expert-designed templates and seeds. For each new task, CS experts first write a task-specific template and 5 seed examples that illustrate the intended capability, edge cases, and verification criteria.
>
> 2. LLM-based instantiation. We then use an LLM (GPT-4o) to instantiate many candidate instructions from these templates, varying entities, attributes, and compositional structure.
>
> 3. Manual filtering. Human annotators remove ambiguous, low-quality, or unverifiable instructions, ensuring that each retained instance admits a clear notion of correctness.
>
> 4. Paraphrasing for diversity. The remaining instructions are paraphrased (again via LLM + human checks) to reduce surface-form overlap while preserving the underlying capability being tested.
>
> To support reproducibility:
>
> 1. Templates and examples. All task templates are shown in Figures 5–53 of the appendix. The initial 5 seed examples for each task are provided in our anonymous repository; they correspond to the first 5 lines of each task-specific JSONL file.
>
> 2. Additional templates (to be added). In the revised version, we will incorporate representative templates that were omitted due to space:
>
> - **Absolute Spatial Relation**
> “Generate an image of [scene], with [object] at the [position] quarter of the image.”
>
> - **Double Text Rendering**
> “Generate an image of exactly two [objects] on a [scene]. The [object] [position] should say [text] while [object] [position] should say [text]. The text should be written in a clear, standard, easily readable typeface without any artistic distortions, and the objects should be clearly visible and not overlapping.”
>
> - **Text Altering**
> (1) “Generate an image showing the editing result after removing [text] at [position].”
> (2) “Generate an image showing the editing result after changing [text] to [text] at [position].”
> (3) “Generate an image showing the editing result after translating [text] at [position] into [language].”
>
> - **Interleaved Adding / Interleaved Altering**
> The reference images are manually created with Photoshop to ensure pixel-level correctness.
>
> - **Speech Retrieval**
> “Listen to the given speech and answer the following question: [question]. Answer the question in speech, using the original speech verbatim, without any change.”
>
> - **Image-Text / Audio-Text Interleaving**
> (1) “If [condition A], generate [modality B], otherwise generate [modality C].”
> (2) “Generate [modality] first and then [modality] according to [order constraint].”

---

> ### Author Response · Authors · 2025-11-22
> **Rebuttal by Authors (3/7)**
>
> > On the insufficient analysis of multimodal reasoning failures: Given top models’ low accuracy on math (10.1%) and 3D code transformation (31.8%) tasks, have you conducted follow-up analyses to identify root causes—such as comparing performance across model architectures (unified autoregressive vs. agent-based), testing with augmented math/code multimodal training data, or adjusting VLM evaluation prompts to rule out strictness bias? Additionally, do you have preliminary insights or suggestions for mitigating these failures?
>
> We agree that a more systematic analysis of modality-specific challenges can strengthen the work. In the revised version, we add a dedicated section on modality-specific error analysis, including (1) an explicit error taxonomy, (2) error-type distributions and statistics, and (3) analysis of likely causes behind these differences. We also present models' error distribution on top of our newly added error taxonomy. We investigate 246 image generation errors and 330 interleaved image-text generation errors on 4 models. The error anlysis for  interleaved image-text tasks is in the table below, please refer to the rebuttal to Reviewer cF2M for image generation error analysis.
>
> |Error Category|Example|Possible Reasons|Number (Percentage) of All errors|Number (Percentage) of Gemini 2.0 Errors|Number (Percentage) of Gemini 2.5 + GPT Image Agent Errors|
> |---|---|---|---|---|---|
> |Generate wrong number of images|Instruction: "Using the provided image as the reference angle, create four additional images of the object from these specific angles in the given order: 30 degrees to the right, 60 degrees to the left, 60 degrees to the right, and 30 degrees to the left." Error: The model generates only 3 images, missing the final "30 degrees to the left" view.|Difficulty in precise counting and planning; the model loses track of the count during the sequential generation loop.|40 (12.2%)|33 (13.9%)|7 (7.6%)|
> |Only image description provided but no image|Instruction: "Create an image depicting a musician's room that includes exactly one microphone, one guitar case, and one music stand, each of different sizes." Error: The model provides the text evaluation ("The microphone is smaller than...") but does not generate the actual image of the room.|Tool-use failure; the model prioritizes the text reasoning task actually calling the image generation tool.|2 (0.6%)|2 (0.8%)|0 (0.0%)|
> |Low image quality or distorted, unrealistic object|Instruction: "Based on the provided image showing a frontal view, create four more images depicting the scene from these specific angles in order:90 degrees to the left, 45 degrees to the right, 45 degrees to the left, and 90 degrees to the right." Error: The first image is okay, but in the "90 degrees to the left" view, the object becomes severely distorted and blurry.|This may be because of under-training. Models are not trained on the target objects enough to reproduce full details.|4 (1.2%)|4 (1.7%)|0 (0.0%)|
> |Combine multiple images together instead of generating multiple images|Instruction: "Based on the provided image showing the frontal view, create four additional images depicting the scene from the following angles, in order: 30 degrees to the right, 45 degrees to the left, 75 degrees to the right, and 30 degrees to the left." Error: Instead of 4 separate image files, the model generates a single image grid containing 4 panels.|Models, especially modality-unified ARMs are likely to tangle multiple images in output due to continuous latent representations. This is likely because of the strong bias in the training data for single-turn unified outputs.|3 (0.9%)|3 (1.3%)|0 (0.0%)|
> |Inconsistent image sequence: objects, style or scene change during generation|Instruction: "Create four images that sequentially show the addition of a passport, a map, a camera, and a pair of sunglasses into an initially empty open suitcase. Do not remove any previously added items." Error: In the 3rd image (adding camera), the passport added in the 1st image disappears, and the suitcase style changes from vintage to modern.|Agent-based models don't show good planning capabilities on image generation tasks. ARMs also lack cross-image consistency mechanisms: state is not maintained across steps.|16 (4.9%)|8 (3.4%)|8 (8.7%)|
> |Image sequence show no variance: image repetition|Instruction: "Using the provided image as the reference angle, create four additional images of the object from these specific angles in order: 60 degrees to the right, 60 degrees to the left, 30 degrees to the right, and 30 degrees to the left." Error: All 4 generated images appear identical to the original reference angle, ignoring the rotation instructions.|Mode output collapse; This is likely caused by the same reason as LLMs output repetitive contents, indicating lack of post-training.|11 (3.3%)|8 (3.4%)|3 (3.3%)|

---

> ### Author Response · Authors · 2025-11-22
> **Rebuttal by Authors (4/7)**
>
> |Error Category|Example|Possible Reasons|Number (Percentage) of All errors|Number (Percentage) of Gemini 2.0 Errors|Number (Percentage) of Gemini 2.5 + GPT Image Agent Errors|
> |---|---|---|---|---|---|
> |Fail to render objects of the required number|Instruction: "Create an image of an office desk that includes a stapler, a mouse, and a pen..." Error: The generated image contains two pens, violating the "exactly once" constraint.|The model demonstrates limited numeracy and counting capabilities. This may be due to the attention mechanism which often fails to disentangle identical object instances, leading to the generation of superfluous objects.|12 (3.6%)|10 (4.2%)|2 (2.2%)|
> |Failed spatial generation, image are not in the position / angle asked|Instruction:"Using the provided image as the reference angle, create four additional images of the object from these specific angles: 30 degrees to the right..." Error: Generate 2 images about 30 degrees to the right and 2 images about 30 degrees to the left.|Poor understanding of 3D position. This could be because models are trained on image-caption pairs which stresses the semantic consistency and overlook other perception abilities.|4 (1.2%)|2 (0.8%)|2 (2.2%)|
> |Failed temporal generation, images don't show the adding / removing order as required|Instruction: "Create three images that sequentially show the addition of a coffee mug, a notebook, and a pen to an initially empty desk. Error: The first image shows the scene of the final result and the second image shows the right scene of a coffee mug and a notebook.|Agent-based models don't show good planning capabilities on image generation tasks. ARMs also lack cross-image consistency mechanisms: state is not maintained across steps.|21 (6.4%)|20 (8.4%)|1 (1.1%)|
> |Failed logic generation, generated images don't show the logical order as required|Instruction: "Create three images, each featuring a single balloon of a different color from: red, yellow, and purple. Arrange the images in alphabetical order based on the color names." Error: The generated images are still the order of red, yellow and purple. Don't show the right logical reasoning process.|Agent-based models don't show good planning capabilities on image generation tasks. ARMs also lack cross-image consistency mechanisms: state is not maintained across steps.|11 (3.3%)|7 (3.0%)|4 (4.3%)|
> |Wrong editing region|Instruction: "Create an image after the utensils have been removed from the photo." Error: The model removes the food on the plate, but leaves the utensils (fork and knife) untouched.|Current training for image editing lacks verifiable, accurate reward, which makes the editing region usually not accurate.|5 (1.5%)|3 (1.3%)|2 (2.2%)|
> |Oversize / undersized editing region|Instruction: "Create an image that displays the result after removing the man's wig, while leaving all other elements of the image unchanged." Error: The edit removes the wig but also unintentionally removes part of the man's forehead and the background curtain behind him.|Current training for image editing lacks verifiable, accurate reward, which makes the editing region usually not accurate.|2 (0.6%)|1 (0.4%)|1 (1.1%)|
> |Invalid edit: edit not applied to text / object as required|Instruction: "Create an image that shows the result after replacing the cop with a smiling clown, while keeping everything else in the image unchanged." Error: The output image is identical to the input image; the cop remains and no clown appears.|Under-training for image editing tasks. Models can't understand instruction very well.|20 (6.1%)|16 (6.8%)|4 (4.3%)|
> |Partial edit: edit only apply to single object / text|Instruction: "Generate an image showing the editing result after making all the sprinkles on the cupcakes blue..." Error: Only the sprinkles on the front cupcake turn blue; the sprinkles on the background cupcakes remain multi-colored.|Under-training for image editing tasks. Models can't understand instruction very well.|4 (1.2%)|3 (1.3%)|1 (1.1%)|
> |Global changes outside the editing region|Instruction: "Create an image displaying the result after replacing the highest kite with an eagle.." Error: The eagle is added correctly, but the sky changes from blue to sunset orange, altering the global scene.|Current training for image editing lacks verifiable, accurate reward, which makes the editing region usually not accurate.|11 (3.3%)|4 (1.7%)|7 (7.6%)|
> |Generated image-text interleaved content are not in required modality order|Instruction: "For each phase, start with an image that illustrates the phase, followed by a written explanation in natural, conversational language." Error: The model outputs all the text explanations first, and then dumps all the images at the very end of the response.|Lack of Instruction following training for interleaved image-text content. Models simply can't have correct output structure and modality sequencing.|6 (1.8%)|5 (2.1%)|1 (1.1%)|

---

> ### Author Response · Authors · 2025-11-22
> **Rebuttal by Authors (5/7)**
>
> |Error Category|Example|Possible Reasons|Number (Percentage) of All errors|Number (Percentage) of Gemini 2.0 Errors|Number (Percentage) of Gemini 2.5 + GPT Image Agent Errors|
> |---|---|---|---|---|---|
> |Parsing error: Wrong function calling format for image generation|Instruction: "Develop a 3-step guide ..." Error: The model generates the text steps correctly but outputs the raw tool code [IMG_GEN: "coffee pouring"] or no image at all.|Lack of training on agentic data. Models can't output formatted text, which is a common requirement for tool agents.|0 (0.0%)|0 (0.0%)|0 (0.0%)|
> |Parsing error: failed to generate text in the required format|Instruction: "...Once the image is complete, return only a JSON object that maps each item to its corresponding color, using the format: {"passport": "[color]", "hat": "[color]", "camera": "[color]"}." Error: The model returns a bulleted list or a sentence like "The passport is red, the hat is beige..." instead of valid JSON.|Lack of training on agentic data. Models can't output formatted text, which is a common requirement for tool agents.|20 (6.1%)|20 (8.4%)|0 (0.0%)|
> |Image inconsistent with the self-generated object description in text|Instruction: "Create four images that sequentially show the result after removing the sunglasses, the camera, the map, and the passport from the original image." Error: The text response correctly states "Removing the passport...", but the final generated image still clearly shows the passport sitting in the suitcase.|Lack of training on interleaved image-text tasks. Models can't output a self-consistent image-text pair. For agentic models, this is usually due to the image generation tool not strictly following the instruction, since the planning model doesn't correct the plan with generation feedback. A debugging or mulit-turn system would be the direct solution.|23 (7.0%)|15 (6.3%)|8 (8.7%)|
> |Image inconsistent with the self-generated relation description in text|Instruction: "...Once the image is complete, answer the following two questions: 1. The measuring tape is to the ___ of the screwdriver: A. left B. right C. neither obviously left nor right..." Error: The model answers "A" (Left) based on its text reasoning, but in the generated image, the measuring tape is actually on the right.|Lack of training on interleaved image-text tasks. Models can't output a self-consistent image-text pair. For agentic models, this is usually due to the image generation tool not strictly following the instruction, since the planning model doesn't correct the plan with generation feedback. A debugging or mulit-turn system would be the direct solution.|45 (13.7%)|31 (13.1%)|14 (15.2%)|
> |Image inconsistent with the self-generated OCR recognition results in text|Instruction: "...After generating the image, output ONLY the announcement text in XML format: <text>announcement</text>..." Error: The XML output reads <text>Science Fair</text>, but the text actually rendered in the image says gibberish.|Lack of training on interleaved image-text tasks. Models can't output a self-consistent image-text pair. For agentic models, this is usually due to the image generation tool not strictly following the instruction, since the planning model doesn't correct the plan with generation feedback. A debugging or mulit-turn system would be the direct solution.|16 (4.9%)|14 (5.9%)|2 (2.2%)|
> |No step by step reasoning analysis|Instruction: "Analyze the following SVG code step-by-step, explaining what each tag draws." Error: The model provides a single summary sentence ("This draws a house") without the requested line-by-line or step-by-step analysis.|Under-training for multimodal reasoning tasks involving image generation. Models present reasoning traces in text, yet this may not be the most effective method for multimodal reasoning. RLVR may improve multimodal reasoning capabilities if more verifiable data like MMMG is released.|2 (0.6%)|2 (0.8%)|0 (0.0%)|
> |Wrong reasoning analysis|Instruction: "What geometric shape does this SVG code describe? ... \<rect x='10' y='10' width='100' height='100' /\> ..." Error: The model incorrectly reasons to get "This code describes a circle."|The model hallucinates the function of the code or cannot mentally simulate the SVG syntax geometry.|46 (14.0%)|22 (9.3%)|24 (26.1%)|
> |Failed generation|Instruction: "..." Error: The output is a completely black image or a corrupted file that cannot be opened.|Activation of safety filters or internal system generation failures.|5 (1.5%)|4 (1.7%)|1 (1.1%)|

---

> ### Author Response · Authors · 2025-11-22
> **Rebuttal by Authors (6/7)**
>
> The overall takeaways are: (1) **multimodal reasoning failure fundamentally rooted in insufficient training data** and training data bias (under-training, coarse captions, distributional bias, spurious correlations). (2) This reveals that model architecture advantages primarily come from scalability - the ability to effectively consume larger amounts of data. This is why autoregressive models outperform diffusion models when scaling [1]. **For multimodal reasoning tasks, agent models may be preferable** as they can leverage powerful LLM capabilities, and these tasks typically do not impose strict consistency requirements on the generation tool. However, for general multimodal interleaved tasks, each architecture presents distinct trade-offs:
>
> + **(ARMs) Poor sequential generation control**. generate wrong number of images (13.9%), failed temporal consistency (8.4%): unified models struggle to maintain state and counting during long-sequence generation.
> + **(ARMs) Poor output format control**. parsing error: failed to generate text in required format (8.4%), combine multiple images together (1.3%): tendency to output entangled images, difficulty generating structured text outputs.
> + **(agent) Poor cross-image consistency**. inconsistent image sequence (8.7%), image inconsistent with relation description (15.2%): due to using independent image generation tools, lack of cross-image state maintenance and self-correction mechanisms.
> + **(agent) Poor global editing control**. global changes outside editing region (7.6%): planning model cannot effectively constrain the output scope of generation tools.
>
> A core design principle of MMMG is to make multimodal generation verifiable: for each task, we design automatic metrics or programs that strongly align with human judgments. This makes MMMG a natural candidate for **RLVR in multimodal generation, a possible solution to imporve multimodal reasoning**. Specifically, using MMMG tasks as reward functions for RL or iterative refinement, allowing models to optimize directly for verifiable multimodal behavior.
>
> We would also like to **clarify the concern on “overly strict VLM evaluation prompts”**. For each task family, we experimented with multiple prompts (e.g., different rubrics, with/without chain-of-thought, response format) and multiple evaluators. We then selected the prompt–evaluator combination with the highest agreement with human judgments, and this is what we report in the paper in Table 8. We will make this procedure and the final chosen prompts clearer in the revision.
>
> [1] Tian K, Jiang Y, Yuan Z, et al. Visual autoregressive modeling: Scalable image generation via next-scale prediction[J]. Advances in neural information processing systems, 2024, 37: 84839-84865.

---

> ### Author Response · Authors · 2025-11-22
> **Rebuttal by Authors (7/7)**
>
> > Regarding the rigor of the "models fall short on multimodal reasoning" claim: Given that math interleaved reasoning tasks were adapted from MM-IQ by converting multiple-choice questions to free-form (Appendix B.2), which may confound reasoning weaknesses with image generation limitations, do you have additional data or controlled experiments (e.g., testing models on the original multiple-choice format vs. free-form, or evaluating image generation quality independently for math tasks) to disentangle whether poor performance stems from multimodal reasoning deficits or inadequate visual rendering?
>
> We appreciate this point and agree that the current wording may sound stronger than what the data strictly supports. In the revision we will soften and refine the claim to: “Current models struggle on MMMG’s multimodal math puzzle tasks, which require both solving visual puzzles and faithfully realizing the solution in the generated image.”
>
> As reported in Figure 36, we **indeed evaluate results in both text and image dimension for interleaved math tasks**. The instructions ask models to generate both the next pattern in language description and pattern in image. The detailed evaluation is as follows.
>
> |Seed|LlamaAnole|GPT-4o + GPT Image|Gemini 2.5 + GPT Image|Gemini 2 Image|GPT Image|
> |-|-|-|-|-|-|
> |text right image right|0.000|0.000|0.038|0.050|0.000|0.000|
> |text right image wrong|0.000|0.000|0.000|0.000|0.025|0.013|
> |text wrong image right|0.000|0.000|0.050|0.050|0.013|0.050|
> |text wrong image wrong|1.000|1.000|0.913|0.900|0.963|0.938|
>
> We find that on average over all models, 95.2% errors are **indeed caused by failed multimodal reasoning** (wrong text wrong image) instead of image generation limitations. Only on average 0.6% errors are caused by image rendering limitations (right text wrong image). We also look into the “impossible” wrong-text-and-right-image category and it turns out that most errors in this category come from overly strict language pattern judgement and loose image pattern judgement. The average error rate 2.7% falls in our reported human agreement rate. This may be because VLM has weaker image capabilities compared to text such that the judgement will be loose for images under the same prompt. However, we indeed find 3 examples that have clearly wrong language pattern image description and clearly right image pattern. This reveals reasoning within the same modality [1] may be more powerful than reasoning with text. For more in-detailed analysis of the reasoning trace errors, please refer to the error table of the rebuttal to the previous question.
>
> We would also like to clarify why we do not focus on multiple-choice questions.Our goal is specifically to evaluate multimodal generation instead of understanding capabilities. Multiple-choice MM-IQ questions merely test visual understanding and math reasoning. MMMG’s math tasks ask models to generate a new image that instantiates the correct pattern, which requires not only understanding but also constructing a high-fidelity solution in the pixel space. These two formats therefore probe different capabilities and are not directly comparable. A model might choose the correct option without being able to generate a corresponding correct image.
>
> [1] Hu Y, Shi W, Fu X, et al. Visual sketchpad: Sketching as a visual chain of thought for multimodal language models[J]. Advances in Neural Information Processing Systems, 2024, 37: 139348-139379.

---

> > ### Comment · Reviewer_R7P8 · 2025-11-27
> >
> > Thank you for your detailed response. Most of my concerns have been addressed, and I will raise my score.

---

> > > ### Author Response · Authors · 2025-11-27
> > > **Thank you!**
> > >
> > > Thank you! We appreciate your thoughtful review comments again - they have greatly improved our paper.

---

### Official Review · Reviewer_cF2M · 2025-11-01

**Soundness:** 3
**Presentation:** 3
**Contribution:** 3
**Rating:** 6
**Confidence:** 4

**Summary:**

The paper introduces MMMG, a benchmark designed to evaluate multitask multimodal generation across text, image, audio, and mixed modalities. It contains 1,248 instructions covering 55 tasks and uses automatic human-aligned scoring to assess model outputs. The benchmark is validated against human preference data and achieves high agreement and consistency across modalities. It provides a unified framework for measuring the performance of multimodal generative models and comparing their strengths and weaknesses across different modalities.

**Strengths:**

1. Proposes a unified benchmark that evaluates multimodal generation across text, image, audio, and interleaved modalities within a single framework.
2. Uses automatic human-aligned scoring verified against human preference data, reducing the reliance on manual evaluation.
3. Provides detailed cross-modal comparison results that reveal modality-specific weaknesses, especially in audio and multi-image generation.

**Weaknesses:**

1. Some task have limited coverage, for example, the code tasks involve only a single programming language, which restricts representativeness.
2. The analysis of modality-specific challenges is not sufficiently in-depth, lacking systematic error-type distributions, cross-model comparisons, and causal interpretation that could provide more actionable insights.

**Questions:**

1. Could you consider expanding the diversity of certain tasks, such as including multiple programming languages in the code-related tasks, to avoid narrow coverage?
2. Could you provide a more in-depth analysis of modality-specific challenges, including systematic error-type statistics, cross-model comparisons, and possible causes behind these differences?

---

> ### Author Response · Authors · 2025-11-22
> **Rebuttal by Authors (1/4)**
>
> We would like to thank the reviewer for your thoughtful comments and feedback.
>
> > Some task have limited coverage, for example, the code tasks involve only a single programming language, which restricts representativeness. Could you consider expanding the diversity of certain tasks, such as including multiple programming languages in the code-related tasks, to avoid narrow coverage?
>
> We appreciate this suggestion. Our goal with MMMG is not to build a benchmark for any specific task, but to design a **principled, modular, and extensible pipeline for verifiable multimodal generation**. The current “code” task on SVG code understanding just instantiates this pipeline in a **fronten-code-to-rendered-image** setting, serving to test a core capability that if models can understand the visual codes they generated in image modality.
>
> At the same time, we agree that showing concrete extensions demonstrates the generality of our pipeline and alleviates concerns about narrow coverage. Therefore, in the revision we extend the code-related tasks along two axes as an example of this extensibility. (1) Multi-language coverage: we collect 20 additional instructions of another widely-used front-end language TikZ (LaTex) apart from SVG (XML). The data is filtered from DaTikZ [1] (2) A new debugging task: we collect 20 additional instructions asking the model to understand the front-end code after adding a patch. The aim of this task is to test the model's debugging instead of generation understanding capability. The data is filtered from SVGEditBench [2]. The 4 model performance and human agreement are as follows:
>
> |Task|Gemini 2.5 + GPT Image|GPT4o + GPT Image|Gemini 2.5 Image|GPT Image|Agreement with human judge|Pearson correlation with human judge|
> |---|---|---|---|---|---|---|
> |Code (LaTeX)|0.054|0.038|0.048|0.094|1.000|1.000|
> |Code (Debug)|0.383|0.185|0.142|0.074|0.925|0.860|
>
> The results demonstrate that models are still consistently struggling with multimodal code understanding without image reference, with average model performance on Code (LateX) being only 0.059. It is inspiring that agent-based models are able to understand multimodal code patches when presenting the original image as reference, where Gemini 2.5 + GPT Image achieves 0.383 accuracy. This indicates: (1) Agent-based models are still better than modality-unified models in multimodal code tasks due to their incomparable text-based reasoning capabilities. Future work on improving reasoning with images would be the key to improve modality-unified models’ performance. (2) Being able to see the visualized multimodal codes is important for models to understand multimodal codes. This reveals a core difference in the multimodal code debugging system than the common debugging system, where image signals (like screenshots) apart from unit tests should be included.
>
> We would also like to add HTML Canvas (JavaScript) subtasks. However, there is no open Canvas dataset which means we need to crawl from the raw github repository. Due to the time limitation of the rebuttal period, we will add this subtask in the revised version.
>
> [1] Belouadi J, Lauscher A, Eger S. Automatikz: Text-guided synthesis of scientific vector graphics with tikz[J]. arXiv preprint arXiv:2310.00367, 2023.
>
> [2] Nishina K, Matsui Y. Svgeditbench v2: A benchmark for instruction-based svg editing[J]. arXiv preprint arXiv:2502.19453, 2025.

---

> ### Author Response · Authors · 2025-11-22
> **Rebuttal by Authors (2/4)**
>
> > The analysis of modality-specific challenges is not sufficiently in-depth, lacking systematic error-type distributions, cross-model comparisons, and causal interpretation that could provide more actionable insights. Could you provide a more in-depth analysis of modality-specific challenges, including systematic error-type statistics, cross-model comparisons, and possible causes behind these differences?
>
> |Error Category|Example|Possible Reasons|Number (Percentage) of All errors|Number (Percentage) of Gemini 2.0 Errors|Number (Percentage) of GPT Image errors|
> |---|---|---|---|---|---|
> |Fail to render required scene properly|Instruction:"Create an image of a waterfall flowing into the sky, containing one pumpkin and one suitcase." Error: Rendered as normal waterfall, not flowing into the sky.|The model relies heavily on physical laws and realistic priors (e.g., gravity) embedded in the training data, struggling to synthesize scenes that violate common sense physics.|3 (1.2%)|0 (0.0%)|3 (3.0%)|
> |Fail to render objects of the required number|Instruction:"Generate an image of a historical site. Please include a single spaceship and a giraffe in the image." Error:Two spaceship even if instruction asked for one|The model demonstrates limited numeracy and counting capabilities. This may be due to the attention mechanism which often fails to disentangle identical object instances, leading to the generation of superfluous objects.|49 (19.9%)|27 (18.5%)|22 (22.0%)|
> |Object rendered with distortion or unrealistic|"instruction": "Create an image of an endless mirrored hallway, with one cactus and one saxophone." Error: The generated mirrored hallway is distorted.|This may be because of under-training. Models are not trained on the target objects enough to reproduce full details.|15 (6.1%)|6 (4.1%)|9 (9.0%)|
> |Required object not present, partial present or the wrong object present|Instruction: "Create an image of a whale flying through a sunset sky, featuring one suitcase and one mailbox." Error: The image doesn't include a suitcase.|Failed instruction following may indicate under post-training. Instead of generating the required image, models generate the most plausible image.|1 (0.4%)|0 (0.0%)|1 (1.0%)|
> |Unable to remove the correlated object|Instruction:"Generate an image of people camping. Do not include tents in the image." Error: The generated image still has tents.|Negative constraints paradoxically activate associated concepts within the latent space. Furthermore, strong semantic co-occurrence between scenes and objects makes it difficult to decouple these associations (spurious correlation).|31 (12.6%)|24 (16.4%)|7 (7.0%)|
> |Fail to render required attribute (single object)|Instruction:"Generate an image of a single chair with 5 legs." Error: The generated chair has 4 legs.|Strong visual priors (such as the normative four-legged chair) override textual prompts, creating a conflict between the model's internal knowledge and counterfactual instructions.|28 (11.4%)|20 (13.7%)|8 (8.0%)|
> |Fail to bind object attributes correctly (multiple objects)|Instruction:"Orange elephant with purple polka dots" Error: colors mixed on elephant's body.|Features from one entity (e.g., color, texture) erroneously propagate to adjacent objects. This is likely because of coarse captions used for training, which omit detailed object attribution.|17 (6.9%)|9 (6.2%)|8 (8.0%)|
> |Fail to apply attribute requirements to all objects (multiple objects)|Instruction:"Generate an image of birds on a wire where all birds are facing the same direction."Error: Not all birds facing the same direction.|Difficulty in maintaining attribute consistency across multiple entities. Stochastic generation processes (diffusion models) often fail to enforce constraints uniformly on every individual instance.|6 (2.4%)|4 (2.7%)|2 (2.0%)|
> |Fail to apply attribute requirements to only the required objects (multiple objects)|Instruction:"Generate an image of a bookshelf where all books are standing vertically except one lying horizontally." Error: "More than one book is lying horizontally."|High logical complexity hampers the execution of "exception" logic, causing the model to over-generalize rules to excluded entities.|6 (2.4%)|5 (3.4%)|1 (1.0%)|
> |Wrong / Insufficient knowledge to reason the target object, wrong object generated|Instruction: "Create an image featuring only the national flag of a country that has a major city located on both the European and Asian continents." Error: The model was expected to generate the Turkish flag, but it mistakenly generated the French flag.|Unlike LLMs, image generation models usually are not trained on extensive world knowledge such that they can only follow the instructions the user gives and can't predict the user's intent.|20 (8.1%)|8 (5.5%)|12 (12.0%)|

---

> ### Author Response · Authors · 2025-11-22
> **Rebuttal by Authors (3/4)**
>
> |Error Category|Example|Possible Reasons|Number (Percentage) of All errors|Number (Percentage) of Gemini 2.0 Errors|Number (Percentage) of GPT Image errors|
> |---|---|---|---|---|---|
> |Wrong / Insufficient knowledge to reason the target object, multiple objects are generated|Instruction: "Generate an image of a small, handheld stringed instrument, commonly used in folk and country music and important in black American music." Error: Generate multiple instruments.|This shows signs of reward hacking in post-training. When the reward model doesn't punish the behavior of the guessing. The model naturally generates multiple objects instead of following instructions.|1 (0.4%)|0 (0.0%)|1 (1.0%)|
> |Fail to render the required logical relation between objects|Instruction: "Create an image featuring a single nail and a single snake, and the nail is longer than the snake." Error: The nail is still shorter than the snake.|The unrealistic logical relation in the prompt conflicts with the model's internal semantic priors, with the model favoring the prior.|25 (10.2%)|17 (11.6%)|8 (8.0%)|
> |Fail to render the required spatial relation between objects|Instruction: "Generate an image of a peaceful garden, with a single watering can at the up right quarter of the image. Error: The watering can is at the right.|Models are trained on image-caption pairs which stresses the semantic consistency and overlook the perception accuracy.|12 (4.9%)|6 (4.1%)|6 (6.0%)|
> |Fail to render accurate image format|Instruction: "Generate a waterfall in a lush rainforest. The entire image should be surrounded by a simple and flat, solid and cyan border and approximately 10% of the image width on all sides." Error: Generate about 30% of the image width on all sides.|The continuous nature of image generation makes precise geometric or color thresholds difficult to enforce. Models tend to prioritize semantic coherence over strict adherence to low-level geometric constraints.|9 (3.7%)|7 (4.8%)|2 (2.0%)|
> |Fail to render image format at all|Instruction: "Generate a space scene with planets and nebulae. The entire image should be surrounded by a simple and flat, solid and black border and approximately 10% of the image width on all sides." Error: Generate the scene without the border.|Formatting constraints are ignored as the model interprets them as soft suggestions rather than hard rules. This is likely a OOD problem, where image formats are not asked in the training data so that model doesn't understand the format.|1 (0.4%)|0 (0.0%)|1 (1.0%)|
> |Rendered text with misspelling|Instruction:"Generate the text 'always move forward'." Error: Generate wrong text because spelling error 'always move forwad'.|Text rendering is a known problem for image generation. Image generating models operate at the pixel level rather than the character level. Lacking explicit orthographic verification, they tend to generate "pseudo-text" that is visually similar but orthographically incorrect.|10 (4.1%)|6 (4.1%)|4 (4.0%)|
> |Rendered text with duplication|Instruction:"Generate the text "capture the moment"." Error: Generate wrong text because duplicate text"capture the the moment".|Repetition artifacts arising from loops or redundancies within the text-generation attention mechanisms.|3 (1.2%)|2 (1.4%)|1 (1.0%)|
> |Required text not rendered|Instruction:"Generate the text "Washington WA 98105 Evergreen State". Error: No text is generated.|Lack of text rendering training makes the model neglects text requirements.|0 (0.0%)|0 (0.0%)|0 (0.0%)|
> |Required text rendered in distortion|Instruction:"Generate the text "Enjoy The Little Things Always". Error:'Enjoy' has distorted glyphs.|Lack of text rendering training makes the model unable to generate high-quality text.|1 (0.4%)|1 (0.7%)|0 (0.0%)|
> |Required text in the wrong place or overflow|Instruction: "Generate an image of exactly two posters on an office wall. The poster on the left side should have \"Plan wisely, embrace each day's potential\" printed on it while the poster on the right side should have \"Persistence transforms obstacles into stepping stones\" printed on it. Error: Generate 3 posters and the text \"Persistence transforms obstacles into stepping stones\" is in the middle poster.|Lack of text rendering training makes the model unable to put text in the right place. Adding a layout controller or train on such data could be helpful.|5 (2.0%)|2 (1.4%)|3 (3.0%)|
> |Required text in wrong language|Instruction: "Generate an image of a notebook and the only text on it is "不要忘记". Error: Generate Chinese characters like symbols but unrecognizable.|Distributional bias in training data; visual similarities between character sets of different languages misclassification during generation.|3 (1.2%)|2 (1.4%)|1 (1.0%)|

---

> ### Author Response · Authors · 2025-11-22
> **Rebuttal by Authors (4/4)**
>
> We agree that a more systematic analysis of modality-specific challenges can strengthen the work. In the revised version, we add a dedicated section on modality-specific error analysis, including (1) an explicit error taxonomy, (2) error-type distributions and statistics, and (3) analysis of likely causes behind these differences. We also present models' error distribution on top of our newly added error taxonomy. We investigate 246 image generation errors and 330 interleaved image-text generation errors on 4 models. The error anlysis for image tasks is in the table above, please refer to the rebuttal to Reviewer R7P8 for interleaved image-text error analysis.
>
> The overall takeaway from the table is: (1) Both models show similar error patterns, with all failures fundamentally rooted in insufficient training data and training data bias (under-training, coarse captions, distributional bias, spurious correlations). (2) This reveals that model architecture advantages primarily come from scalability - the ability to effectively consume larger amounts of data. This is why autoregressive models outperform diffusion models when scaling [1].
>
> [1] Tian K, Jiang Y, Yuan Z, et al. Visual autoregressive modeling: Scalable image generation via next-scale prediction[J]. Advances in neural information processing systems, 2024, 37: 84839-84865.

---

> ### Comment · Reviewer_cF2M · 2025-11-23
>
> Thank you for your detailed response. I think the paper is acceptable, most of my concerns have been addressed, and I will raise my score.

---

> > ### Author Response · Authors · 2025-11-25
> > **Thank you!**
> >
> > Thank you! We appreciate your thoughtful review comments again - they have greatly improved our paper.

---

### Official Review · Reviewer_4AuQ · 2025-11-01

**Soundness:** 3
**Presentation:** 3
**Contribution:** 3
**Rating:** 6
**Confidence:** 3

**Summary:**

The authors propose MMMG, a large-scale benchmark for multitask multimodal generation, including text, image, and audio modalities. Specifically, it involves 55 tasks, with 31 newly designed, and the evaluation pipeline is fully automatic. In this paper, 29 generative models are evaluated.

**Strengths:**

- the paper extand the scope of the previous benchmarks, with more comprehensive tasks and generative models
- the evaulation process is fully automatic, without human evaulation

**Weaknesses:**

- reproducibility: while the paper claims that “all tasks and evaluation scripts will be released,” no code, data, or even minimal examples are available to reviewers
- reliability: the reported high human–judge agreement lacks sufficient methodological details: the scale of human annotations, sampling strategy, agreement metric, and per-task breakdown are all missing
- task design: although 31 new tasks are claimed, many appear to be prompt variants or loosely defined categories. the authors provide limited evidence that these tasks **systematically** capture **distinct** generation abilities or correlate with real-world applications

**Questions:**

Please refer to the weaknesses section.

---

> ### Author Response · Authors · 2025-11-22
> **Rebuttal by Authors (1/2)**
>
> We would like to thank the reviewer for your thoughtful comments and feedback.
>
> > reproducibility: while the paper claims that “all tasks and evaluation scripts will be released,” no code, data, or even minimal examples are available to reviewers
>
> Appendix A.3 (“Reproducibility Statement”) already describes our reproducibility: it points to an anonymous repository which contains all benchmark data, evaluation scripts and protocol (VLM prompts, CLAP / Whisper / DreamSim / SSIM wrappers, and programmatic verifiers, etc), human-annotation procedures and interfaces and examples in readme to replicate the experiments results. The link is at Line 854.
>
> > reliability: the reported high human–judge agreement lacks sufficient methodological details: the scale of human annotations, sampling strategy, agreement metric, and per-task breakdown are all missing
>
> The requested methodological details are already in Section 4.1, Table 8, and App. C.5, but we agree they are too scattered and will make them more prominent. Concretely:
>
> - **Scale of human annotations.** Section 4.1 states that we run human studies on 41 tasks, covering 938 instructions and 2556 evaluation questions each judged by two independent annotators, with a third annotator resolving disagreements, yielding 5412 total annotations.
>
> - **Sampling strategy.**  Appendix C.2 states for each instruction we randomly sample two models from all models evaluated on that instruction and obtain one generation per model. Annotators are randomly drawn from a pool of 20 CS-related graduate students, and each generation is assigned to two distinct annotators. We also state sampling parameter details in this section.
>
> - **Agreement metrics.**  Section 4.1 and Table 8 specify what we report: agreement (for discrete labels, simple accuracy; for continuous scores, agreement after binning into discrete categories); Pearson correlation; Word Accuracy for text rendering, text editing, OCR, and some speech tasks. We also report inter-annotator agreement per task in Table 8, with an average of 0.971 and a worst case of 0.917.
>
> - **Per-task breakdown.**  Table 8 already contains the **per-task** human–evaluator and inter-annotator agreement for all tasks evaluated by humans (image, audio, and speech).
>
> We will add a short paragraph in Section 4.1 summarizing the above bullet points in one place, so that the scale, sampling, metrics, and per-task breakdown are easy to find and refer to corresponding tables and sections in the Appendix.

---

> ### Author Response · Authors · 2025-11-22
> **Rebuttal by Authors (2/2)**
>
> > task design: although 31 new tasks are claimed, many appear to be prompt variants or loosely defined categories. the authors provide limited evidence that these tasks systematically capture distinct generation abilities or correlate with real-world applications.
>
> We would like to clarify that task templates
> and instructions are not loosely defined. In Section 3.1 we describe our task creation pipeline as:  we first manually design an initial pool of 78 tasks with each task containing 1-5 task templates written by our graduate researchers and is intended to test one specific capability of models. We then go through rigorous filtering to avoid unrealistic, ambiguous and redundant tasks or tasks without real-world application.
>
> To address the reviewer’s concern more directly, we add a new analysis that studies **how redundant or complementary our tasks are from the perspective of model performance**, a standard approach in recent multi-task benchmark work [1, 2]. For each MMMG task $t$, we form a vector $a_t\$ of per-model accuracies across all evaluated models (for image tasks, the 10 models in Table 9). For each pair of tasks $\(t_i, t_j)$ in the same modality, we compute the Pearson correlation between $a_{t_i}\$ and $a_{t_j}\$. Intuitively, if two tasks always rank models in almost the same way, they behave like a single task; if correlations are lower, they capture complementary signals. Using the nine image generation models shared with Table 3, we first compute cross-benchmark Spearman correlations between GenEval, DrawBench, and GenAI-Bench:
>   - ρ(GenEval, GenAI-Bench) = **0.778**
>   - ρ(DrawBench, GenAI-Bench) = **0.754**
>   - ρ(GenEval, DrawBench) = **0.569**
>
>   |Task|Object Include|Object Exclude|Object Count|Object Knowl.|Object Common.|Object Attribute|Compare Relation|Universal Relation|Relative Spatial|Absolute Spatial|Region Fill|Border Fill|Single TR|Double TR|Multi-lingual TR|
>   |---|---|---|---|---|---|---|---|---|---|---|---|---|---|---|---|
>   |Object Include|-|0.677|0.464|0.712|0.441|0.557|0.734|0.654|0.720|0.501|0.473|0.096|0.570|0.611|0.627|
>   |Object Exclude|0.677|-|0.859|0.671|0.232|0.742|0.884|0.930|0.889|0.540|0.691|0.389|0.601|0.799|0.656|
>   |Object Count|0.464|0.859|-|0.609|0.198|0.623|0.673|0.919|0.857|0.680|0.696|0.378|0.767|0.894|0.838|
>   |Object Knowl.|0.712|0.671|0.609|-|0.466|0.363|0.764|0.671|0.700|0.529|0.530|0.309|0.634|0.772|0.663|
>   |Object Common.|0.441|0.232|0.198|0.466|-|-0.120|0.304|0.298|0.196|-0.068|0.087|-0.458|0.477|0.453|0.220|
>   |Object Attribute|0.557|0.742|0.623|0.363|-0.120|-|0.664|0.664|0.740|0.615|0.823|0.612|0.188|0.449|0.550|
>   |Compare Relation|0.734|0.884|0.673|0.764|0.304|0.664|-|0.735|0.879|0.518|0.729|0.264|0.465|0.695|0.525|
>   |Universal Relation|0.654|0.930|0.919|0.671|0.298|0.664|0.735|-|0.881|0.621|0.637|0.395|0.744|0.844|0.750|
>   |Relative Spatial|0.720|0.889|0.857|0.700|0.196|0.740|0.879|0.881|-|0.763|0.807|0.400|0.619|0.789|0.729|
>   |Absolute Spatial|0.501|0.540|0.680|0.529|-0.068|0.615|0.518|0.621|0.763|-|0.802|0.515|0.468|0.633|0.745|
>   |Region Fill|0.473|0.691|0.696|0.530|0.087|0.823|0.729|0.637|0.807|0.802|-|0.557|0.252|0.611|0.593|
>   |Border Fill|0.096|0.389|0.378|0.309|-0.458|0.612|0.264|0.395|0.400|0.515|0.557|-|-0.113|0.201|0.281|
>   |Single TR|0.570|0.601|0.767|0.634|0.477|0.188|0.465|0.744|0.619|0.468|0.252|-0.113|-|0.859|0.817|
>   |Double TR|0.611|0.799|0.894|0.772|0.453|0.449|0.695|0.844|0.789|0.633|0.611|0.201|0.859|-|0.851|
>   |Multi-lingual TR|0.627|0.656|0.838|0.663|0.220|0.550|0.525|0.750|0.729|0.745|0.593|0.281|0.817|0.851|-|
>
> We also compute the correlation of all image task pairs in MMMG. Across all such task pairs, we find that only 38 pairs (18.01% of all pairs) have correlation higher than 0.778. In other words, more than 80% of MMMG task pairs are less correlated with each other than two popular image benchmarks are with each other. This indicates that our tasks are not a collection of near-duplicate prompts: they provide more diverse behavioral signals than entire existing benchmarks do. Since there is no baseline benchamrks for reference in other modalities, we only present the correlation between image tasks.
>
> Also, the experiments results in Section 4.3 and Table 3 states that even though MMMG may not align with real-world user instruction distribution, its unpreceding reliability make it more correlated with real-world application like Chatbot Arena.
>
> [1] Ni J, Xue F, Yue X, et al. Mixeval: Deriving wisdom of the crowd from llm benchmark mixtures[J]. Advances in Neural Information Processing Systems, 2024, 37: 98180-98212.
>
> [2] Zhang G, Hardt M. Inherent trade-offs between diversity and stability in multi-task benchmarks[J]. arXiv preprint arXiv:2405.01719, 2024.

---

### Official Review · Reviewer_RE41 · 2025-11-03

**Soundness:** 3
**Presentation:** 3
**Contribution:** 3
**Rating:** 6
**Confidence:** 4

**Summary:**

The paper introduces MMMG, a benchmark framework aiming to evaluate multitask multimodal generation models across diverse modality combinations, including image, audio, text-image, and audio-text tasks. MMMG features 55 tasks (with 31 newly developed), 1248 systematically curated instructions, and task-specific automated evaluation pipelines calibrated to human judgment. The work validates its evaluation methodology with extensive human studies, demonstrates high human alignment (average 94.4%), and provides detailed benchmarking results over 29 multimodal models, highlighting current strengths and limitations, particularly in audio and interleaved modalities

**Strengths:**

1. MMMG spans a much broader set of modalities and tasks than previous benchmarks, which contrasts task coverage and evaluation methods with prior works. This breadth not only increases utility for the community but also allows for detailed capability analysis of models across modalities.

2.  The evaluation section describes the tailored combination of automatic and programmatic metrics (VLM, CLAPScore, SSIM, DreamSim, etc.), extensively calibrated to maximize human agreement per modality. Figure 1 provides transparent examples showing task definitions, pseudo-code, and sample outputs, illustrating both the richness and the reliability of the evaluation protocols.

3.  Benchmarking is conducted over 29 diverse, open, and proprietary models. The results, broken down by subtask as seen in Figure 2 and surrounding text, illuminate where leading models excel and fail, and highlight meaningful gaps in current generation capabilities.

**Weaknesses:**

1.  While Table 2 provides a broad comparative overview, the paper omits direct citation and discussion of several recent, highly related benchmarks (see "Potentially Missing Related Work" below), such as MMIG-Bench[1]  and GEM[2].

2. While the Spearman correlation with Chatbot Arena is reported, the issue of overfitting to benchmark-specific instruction distributions versus true generalization remains open. The impact of this gap is only partially discussed and not probed empirically, leaving open the possibility that MMO's human alignment may not extend to long-tail, natural instructions.

3. While the paper boasts coverage of subjective reasoning (e.g., commonsense, multi-hop, creative attribute manipulation), it does not always specify rigorous, repeatable scoring formulas for these tasks, especially where "human alignment" cannot be boiled down to a single automatic test.

4. Although the paper highlights the use of programmatic and model-based evaluation (CLAPScore, Whisper, WavLM, DreamSim), there is limited concrete discussion or evidence regarding the limitations of these metrics in more complex settings (e.g., music genre, instrument separation, conversation consistency for speech-audio tasks). For example, the sensitivity of CLAPScore to reference selection (Section 4.1) is acknowledged, but quantitative ablation results or detailed error analysis are missing. This leaves ambiguity about how robust the metrics truly are, especially for subjective or creative generation tasks that resist narrow programmatic evaluation.


[1] Hua, H., Zeng, Z., Song, Y. (2025): MMIG-Bench: Towards Comprehensive and Explainable Evaluation of Multi-Modal Image Generation Models.

[2] Su, Y., et al. (2024): GEM: A General Evaluation Benchmark for Multimodal Tasks.

**Questions:**

As shown in weaknesses.

---

> ### Author Response · Authors · 2025-11-21
> **Rebuttal by Authors (1/3)**
>
> We would like to thank the reviewer for your thoughtful comments and feedback.
>
> > While Table 2 provides a broad comparative overview, the paper omits direct citation and discussion of several recent, highly related benchmarks (see "Potentially Missing Related Work" below), such as MMIG-Bench[1] and GEM[2].
>
> Thank you for bringing up these two works! We will add them to the related work section and update Table 2 accordingly. GEM focused on image–text and video–text **understanding and caption generation**. It does not evaluate multimodal generation capability, so we will cite it as a multimodal understanding benchmark rather than include it in Table 2.
>
> To clarify the relationship between MMIG-Bench and MMMG, we will add a row to Table 2 (simplified here):
>
> |Dataset|#Inst.|#Tasks|Image|Audio|Text+Image|Text+Audio|Human|MLM|Score|Code|Gen|Edit|Reason|
> |---|---|---|---|---|---|---|---|---|---|---|---|---|---|
> |MMIG-Bench|4850|12|✔|✘|✔|✘|✔|？|✔|✘|✔|✔|✘|
> |MMMG(ours)|1248|55|✔|✔|✔|✔|✘|✔|✔|✔|✔|✔|✔|
>
> > While the Spearman correlation with Chatbot Arena is reported, the issue of overfitting to benchmark-specific instruction distributions versus true generalization remains open. The impact of this gap is only partially discussed and not probed empirically, leaving open the possibility that MMMG's human alignment may not extend to long-tail, natural instructions
>
> We appreciate the reviewer's concern and agree that correlation with Chatbot Arena alone does not fully resolve the question of generalization to the long tail nature instructions. Our intention is just to provide external validity for MMMG's rankings, not to claim that MMMG matches real-world prompt distributions.
>
> Specifically, we show that, despite being built from synthetic instructions, MMMG yields model rankings that correlate more strongly with Chatbot Arena than prior image benchmarks (Spearman 0.561 vs. 0.418–0.460), and also exhibits a larger performance spread between top and bottom models. This suggests that **improving evaluator reliability (our main focus) substantially increases agreement with human preferences even when the benchmark instruction distribution may not match Arena**.
>
> Also, to our best knowledge, Chatbot Arena is the only large-scale human ranking leaderboard for multimodal models. If there is, we are happy to add correlation analysis on that to further strengthen our claims.
>
> Finally, **MMMG already includes many long-tail instructions**. For example: generate objects with atypical shapes and attributes (e.g., “a bike with a square front wheel and a circular rear wheel”) or exclude related objects from the scene (e.g., “a birthday party without cakes”). as listed in Table 1 and examples from Figure 5 to Figure 53.

---

> ### Author Response · Authors · 2025-11-22
> **Rebuttal by Authors (2/3)**
>
> > While the paper boasts coverage of subjective reasoning (e.g., commonsense, multi-hop, creative attribute manipulation), it does not always specify rigorous, repeatable scoring formulas for these tasks, especially where "human alignment" cannot be boiled down to a single automatic test.
>
> We would like to clarify a possible misunderstanding. **We explicitly exclude subjective tasks In MMMG.** Tasks that require open-ended aesthetic or emotional judgment (e.g., music emotion, lyrics quality, style preferences) were *removed* during benchmark post-processing and are listed in Table 4 as excluded tasks. Only tasks for which we could define clear, objective criteria (or programmatic checks) were kept.
>
> We respectfully disagree that the commonsense, multi-hop, and attribute-based tasks in MMMG are inherently “subjective”. Our multi-hop and commonsense tasks follow the same paradigm as widely used LLM QA benchmarks such as HotpotQA and HellaSwag: each question has a single correct answer supported by widely acknowledged facts or commonsense relations, and models are evaluated by exact match to that answer. We suppose the “creative attribute manipulation” to be the object attribution tasks which specify concrete, unambiguous constraints (e.g., “a red cat with blue circular spots”). Evaluation checks whether these precise attributes hold in the generated output; there is no scoring based on aesthetic quality or “how creative” the result looks.
>
>
> We present the human annotation interface in Figure 4. Each instruction is associated with a task-specific multiple-choice (or similarly structured) question and a detailed set of criteria that can be objectively judged. A model output is counted as correct only if it satisfies all criteria simultaneously. The grading questions and criteria for each task are included in our released codebase with an anonymous link in Appendix A.3.
>
> As for **Inter-annotator agreement**, we already report inter-annotator agreement (IAA) in Section 4.1 and Table 8, showing an average IAA of 0.971 across 41 tasks, with the worst case still above 0.91. This indicates the evaluation is really objective and unambiguous.
>
> To avoid further misunderstanding, we will make this explicit in the revision by (1i) adding a short clarification that all tasks are formulated with single ground-truth answers and can be objectively evaluated, and (2) pointing readers to the released annotation process in Appendix C.4 and C.5.

---

> ### Author Response · Authors · 2025-11-22
> **Rebuttal by Authors (3/3)**
>
> > Although the paper highlights the use of programmatic and model-based evaluation (CLAPScore, Whisper, WavLM, DreamSim), there is limited concrete discussion or evidence regarding the limitations of these metrics in more complex settings (e.g., music genre, instrument separation, conversation consistency for speech-audio tasks). For example, the sensitivity of CLAPScore to reference selection (Section 4.1) is acknowledged, but quantitative ablation results or detailed error analysis are missing. This leaves ambiguity about how robust the metrics truly are, especially for subjective or creative generation tasks that resist narrow programmatic evaluation.
>
> We agree that understanding the failure modes of our automatic metrics is important. Our current focus in Section 4.1 and Table 8 demonstrates that, after careful calibration, these metrics achieve high agreement with human judgments. Because **residual disagreement is relatively small**, our qualitative discussion of limitations focused on the most salient issues (e.g., CLAPScore’s dependence on reference quality; VLM's sensitivity to out-of-distribution scenes) rather than on per-task error catalogs. We will expand this discussion in the revised version with quantitative analysis of all the protocols we use. Due to the time limitation of the rebuttal period, we only present the error analysis of CLAPScore and WavLM. We will present the rest of evaluation model analysis in the revision:
>
> CLAPScore (7.7% error in total):
>
> | Error Category | Percentage in All Errors | Observation | Plausible Causes |
> |---|---|---|---|
> | Audio generation failure | 20.0% | Generated audio consists of pure noise with no meaningful structure. | CLAPScore (audio) is not OOD generalizable. When generation quality is low, they can't make reliable judgments. |
> | Audio generation quality unsatisfactory | 26.7% | The generated audio contains recognizable and reasonable sounds but overall of low quality with distortion. For example, the models repeatedly failed to generate proper reggae music, often producing tracks dominated by drums. They also struggled to generate jazz music lacking essential jazz elements (e.g., saxophone). | CLAPScore (audio) is not OOD generalizable. When generation quality is low, they can't make reliable judgments. |
> | CLAP lack of fine-grained understanding | 26.7% | Some generated audio includes fine-grained details that are easily recognizable by humans, such as brief dog barks, or brief sneezes, but their duration is too short. Others contain noisy background such as sirens in loud traffic, which is treated as meaningless piece by CLAP model. | Since CLAPScore computes the similarity of the whole generated audio with the reference audio, short duration or noisy background will underestimate such similarity. Future methods could attempt to locate the target sound first before computing similarity. |
> | Underrepresented reference audio | 26.7% | Some generated audio is not representative in the reference audio. A peaceful drum sequence is not considered as containing drums by CLAP model since all the reference drum music are metallic-like. | The reference dataset we use are standard and may not fully capture a sound, an instrument or a music genre. Enriching the dataset diversity is important for more reliable evaluation. |
>
> WavLM (13.7% error in total):
>
> |Error Category|Percentage in All Errors|Observation|Plausible Causes|
> |---|---|---|---|
> |Generated speech with noise or distortion|21.4%|Some generated speeches contain noticeable background white noise. Sometimes, the noise is the intermittent volume fluctuations (sometimes slightly lower, sometimes higher) or long pauses, which may interfere with the model's judgment, thinking the speakers are different.|WavLM is not OOD generalizable. When generation quality is low, it can't make reliable judgement.|
> |Hard to decide speaker indentity|57.1%|Some generated speeches can be considered from the same speaker, but with clear synthetic pattern, making it hard even for human judges to decide if the generated speeches are from the same speaker.|WavLM is not OOD generalizable. It is trained on real-world human voice and may not effectively judge synthetic speeches.|
> |Computed similarity score close to threshold|21.4%|The generated speeches are very similar to the references, aside from slight white noise. This likely caused the model's judgment to be only minimally affected, placing the scores near the decision threshold (borderline).|WavLM is not well-calibrated. It will only give high scores to the same speaker and low scores to different speakers. When the score is close to the threshold, it is likely to make mistakes.|
>
> We hope this makes clear that we are aware of these metrics’ limitations, already calibrate them carefully against human judgments, and are committed to adding more quantitative evidence and examples in the final version.

---

> ### Comment · Reviewer_RE41 · 2025-11-26
>
> Thank you for the clear and detailed rebuttal. All my concerns have been well addressed. I am satisfied with the responses and will keep my original positive score.

---

### Author Response · Authors · 2025-11-30
**Global Responses by Authors**

**Dear Reviewers and Area Chair**,

We thank all reviewers for their valuable comments. Below are a conclusion of the contribution of MMMG and general responses to the common concerns raised during the rebuttal period. We are currently polishing the paper and incorporating the new results.

**Summary of Contributions**

MMMG introduces a comprehensive and reliable benchmark for multitask multimodal generation, spanning 55 tasks (31 newly developed), 1,248 instructions, and 4 modality combinations. It is the first benchmark that, while ensuring reliability, provides the most comprehensive multimodal model ranking and fine-grained capability analysis. This work enables reliable capability ranking across modalities (achieving 94.4% human agreement) and offers foundational insights for designing verifiable reward signals in future multimodal generation training.

**Summary of Reviewer Concerns and Responses**

We addressed reviewer concerns across four main categories:
1. Reproducibility and Methodology (4AuQ, RE41): We clarified that all code, data, and evaluation scripts are available in our anonymous repository. We detailed where we clarify the human study scale (938 instructions, 2,556 questions, 5,412 annotations) and per-task agreement metrics in the original paper.
2. Task Novelty and Design (R7P8, 4AuQ): We clarified that our contribution lies in principled task and evaluation design, not raw data collection. We provided task correlation analysis showing >80% of MMMG task pairs are less correlated than entire existing benchmarks, demonstrating complementary evaluation signals.
3. Error Analysis on Evaluation and Generation Models (RE41, cF2M, R7P8): We provided comprehensive error taxonomies for both evaluation models (34 CLAPScore, WavLM errors) and generation models (246 image errors and 330 interleaved errors across Gemini and GPT models), with error categorization, quantitative breakdowns and systematic root cause analysis.
4. Task Coverage and Granularity (cF2M, R7P8): We extended code tasks to include LaTeX/TikZ and debugging subtasks (per cF2M), and broke down the math tasks scoring: 95.2% of math task errors stem from reasoning failures rather than generation limitations, validating our multimodal reasoning claims (per R7P8).

**Acknowledgments**

We are honored that all reviewers who responded to our rebuttal (RE41, cF2M, R7P8) confirmed that we have addressed all or most of their concerns. We believe our detailed rebuttal has also thoroughly addressed all questions raised by reviewer 4AuQ. Prior to the score lock, reviewers cF2M and R7P8 raised their scores to 6 and 8 respectively, bringing our total score to 8666 (the remaining unresponded score is 6 from reviewer 4AuQ). We are grateful for the constructive engagement from all reviewers, which has substantially strengthened our work.

---

### Meta-Review · Area_Chair_RAv3 · 2026-01-04

**Summary:**

The paper presents MMMG, a benchmark for multitask multimodal generation covering 4 modality combinations, 55 tasks, and 1248 instructions. The reviewers initially raised concerns about reproducibility, originality, error analysis, and task coverage. Despite the authors’ efforts on supplementary experiments to address these concerns in the rebuttal, the work's core limitation—superficial analytical depth—remains unaddressed. Furthermore, the paper’s analysis of architecture differences among closed-source models using benchmark results is inherently not rigorous, given the unknown details of these models. This fundamental flaw diminishes MMMG's practicality and scientific impact.

**Reviewer Concerns:**

- **Solved Concerns**: The reproducibility, the originality of the dataset, the task coverage and granularity.

- **Outstanding Concerns**: The rebuttal’s error analysis, while structured, remains surface-level. It categorizes failures but fails to provide meaningful causal insights.

**Reviewer Scores:**

- Reviewer RE41: Remains at 6 – concerns addressed.
- Reviewer 4AuQ: Remains at 6 or decreases to 5 – reproducibility clarified, but the core issue of the rationality of the task design has not been substantially resolved.
- Reviewer cF2M: Remains at 6 or decreases to 5 – while code task expansion and error analysis are mentioned, no in-depth analysis or insights are actually provided.
- Reviewer R7P8: Remains at 4 - although an error analysis was provided, the depth of the analysis was limited, and the conclusions were not rigorous enough.

---

### Decision · Program_Chairs · 2026-01-26

Reject